# Span-Based Optimal Sample Complexity for Weakly Communicating and General Average Reward MDPs

**Matthew Zurek**
Department of Computer Sciences
University of Wisconsin-Madison
`matthew.zurek@wisc.edu`

**Yudong Chen**
Department of Computer Sciences
University of Wisconsin-Madison
`yudong.chen@wisc.edu`

## Abstract

We study the sample complexity of learning an $\varepsilon$-optimal policy in an average-reward Markov decision process (MDP) under a generative model. For weakly communicating MDPs, we establish the complexity bound $\widetilde{O}\left(SA\frac{\mathsf{H}}{\varepsilon^2}\right)$, where $\mathsf{H}$ is the span of the bias function of the optimal policy and $SA$ is the cardinality of the state-action space. Our result is the first that is minimax optimal (up to log factors) in all parameters $S, A, \mathsf{H}$, and $\varepsilon$, improving on existing work that either assumes uniformly bounded mixing times for all policies or has suboptimal dependence on the parameters. We also initiate the study of sample complexity in general (multichain) average-reward MDPs. We argue a new transient time parameter $\mathsf{B}$ is necessary, establish an $\widetilde{O}\left(SA\frac{\mathsf{B}+\mathsf{H}}{\varepsilon^2}\right)$ complexity bound, and prove a matching (up to log factors) minimax lower bound. Both results are based on reducing the average-reward MDP to a discounted MDP, which requires new ideas in the general setting. To optimally analyze this reduction, we develop improved bounds for $\gamma$-discounted MDPs, showing that $\widetilde{O}\left(SA\frac{\mathsf{H}}{(1-\gamma)^2\varepsilon^2}\right)$ and $\widetilde{O}\left(SA\frac{\mathsf{B}+\mathsf{H}}{(1-\gamma)^2\varepsilon^2}\right)$ samples suffice to learn $\varepsilon$-optimal policies in weakly communicating and in general MDPs, respectively. Both these results circumvent the well-known minimax lower bound of $\widetilde{\Omega}\left(SA\frac{1}{(1-\gamma)^3\varepsilon^2}\right)$ for $\gamma$-discounted MDPs, and establish a quadratic rather than cubic horizon dependence for a fixed MDP instance.

## 1 Introduction

The paradigm of Reinforcement learning (RL) has demonstrated remarkable successes in various sequential learning and decision-making problems. Empirical successes have motivated extensive theoretical study of RL algorithms and their fundamental limits. The RL environment is commonly modeled as a Markov decision process (MDP), where the objective is to find a policy $\pi$ that maximizes the expected cumulative rewards. Different reward criteria are considered, such as the finite horizon total reward $\mathbb{E}^\pi\left[\sum_{t=0}^T R_t\right]$ and the infinite horizon total discounted reward $\mathbb{E}^\pi\left[\sum_{t=0}^\infty \gamma^t R_t\right]$ with a discount factor $\gamma < 1$. The finite horizon criterion only measures performance for $T$ steps, and the discounted criterion is dominated by rewards from the first $\frac{1}{1-\gamma}$ time steps. In many situations where the long-term performance of the policy $\pi$ is of interest, we may prefer to evaluate policies by their long-run average reward $\lim_{T\to\infty}(1/T)\mathbb{E}^\pi\left[\sum_{t=0}^{T-1} R_t\right]$.

A foundational theoretical problem in RL is the sample complexity for learning a near-optimal policy using a generative model of the MDP [10], meaning the ability to obtain independent samples of the next state given any initial state and action. For the finite horizon and discounted reward criteria, the sample complexity of this task has been thoroughly studied (e.g., [2, 3, 15, 19, 1, 12]). However, despite significant effort (reviewed in Section 1.1), the sample complexity of the average reward setting is unresolved in existing literature.

38th Conference on Neural Information Processing Systems (NeurIPS 2024).

**Our contributions** In this paper, we resolve the sample complexity of weakly communicating Average-Reward MDPs (AMDP) in terms of $\mathsf{H} := \|h^\star\|_{\mathrm{span}}$, the span of the bias (a.k.a. relative value function) of the optimal policy. We show that $\widetilde{O}(SA\mathsf{H}/\varepsilon^2)$ samples suffice to find an $\varepsilon$-optimal policy of a weakly communicating MDP with $S$ states and $A$ actions. This bound, presented in Theorem 2, is the first that matches the minimax lower bound $\widetilde{\Omega}(SA\mathsf{H}/\varepsilon^2)$ up to log factors.

Furthermore, we initiate the study of sample complexity for average-reward *general MDPs*, which refers to the class of all finite-space MDPs without any restrictions [14]. General MDPs are not necessarily weakly communicating and all their optimal policies may be *multichain*. In this general setting, we demonstrate the span $\mathsf{H}$ *alone* cannot characterize the sample complexity, as the lower bound in Theorem 4 exhibits instances which require $\gg \mathsf{H}SA/\varepsilon^2$ samples. This observation motivates our introduction of a new *transient time bound* parameter $\mathsf{B}$, which in conjunction with $\mathsf{H}$ captures the sample complexity of general average-reward MDPs. Specifically, our Theorem 8 shows that $\widetilde{O}\left(SA\frac{\mathsf{B}+\mathsf{H}}{\varepsilon^2}\right)$ samples suffice to learn an $\varepsilon$-optimal policy, and Theorem 4 provides a matching minimax lower bound of $\Omega\left(SA\frac{\mathsf{B}+\mathsf{H}}{\varepsilon^2}\right)$. We remark that it is trivially impossible to achieve low regret in standard online settings of general MDPs, since the agent may become trapped in a closed class of low reward states [4]. The simulator setting is natural for studying general MDPs since it avoids this fatal issue, although the existence of multiple closed classes with different long-run rewards still plays a fundamental role in the minimax sample complexity, as reflected in the dependence on $\mathsf{B}$.

To establish the above upper bounds, we adopt the reduction-to-discounted-MDP approach [9, 20], and improve on prior work by developing enhanced sample complexity bounds for $\gamma$-discounted MDPs (DMDPs). We improve the analysis of variance parameters related to DMDPs using a new multistep variance Bellman equation, which is applied in a recursive manner to bound the variance of near-optimal policies. For general (multichain) MDPs, we further utilize law-of-total-variance ideas to bound the total variance contribution from transient states, which present new challenges significantly different to their behavior in the weakly communicating setting. Our average-to-discounted reduction also requires new techniques, because many structural properties used in earlier reduction arguments no longer hold for general MDPs. Our analysis leads to DMDP sample complexities of $\widetilde{O}\left(SA\frac{\mathsf{H}}{(1-\gamma)^2\varepsilon^2}\right)$ and $\widetilde{O}\left(SA\frac{\mathsf{B}+\mathsf{H}}{(1-\gamma)^2\varepsilon^2}\right)$ to learn $\varepsilon$-optimal policies in weakly communicating and general MDPs, respectively. Notably, the latter bound, valid for all MDPs, circumvents the existing lower bound $\widetilde{\Omega}\left(\frac{SA}{(1-\gamma)^3\varepsilon^2}\right)$ [3, 15]. Whereas this minimax lower bound allows the adversary to choose the transition matrix $P$ based on $\gamma$ with $\mathsf{B} \approx \frac{1}{1-\gamma}$ [3, Theorem 3], our result reflects the complexity of a *fixed* MDP $P$ through its parameters $\mathsf{H}, \mathsf{B}$ and a quadratic dependence on the effective horizon $\frac{1}{1-\gamma}$. This fixed-$P$ complexity is essential for our particular algorithmic approach, where the reduction discount $\gamma$ is chosen depending on $P$. It is also a more relevant framework in general for many RL problems where the discount factor is tuned for best performance on a particular instance.

## 1.1 Comparison with related work on average-reward MDPs

We summarize in Table 1 existing sample complexity results for average reward MDPs.

Various parameters have been used to characterize the sample complexity of average reward MDPs, including the diameter $D$ of the MDP, the uniform mixing time bound $\tau_{\mathrm{unif}}$ for all policies, and the span $\mathsf{H}$ of the optimal bias; formal definitions are provided in Section 2. All sample complexity upper bounds involving $\tau_{\mathrm{unif}}$ require the strong assumption that *all* stationary deterministic policies have finite mixing times. Otherwise, $\tau_{\mathrm{unif}} = \infty$ by definition, which for example occurs if some policy induces a periodic Markov chain. It is also possible to have $D = \infty$, while $\mathsf{H}$ and our newly introduced $\mathsf{B}$ are always finite for finite state-action spaces. As shown in [20], there is generally no relationship between $D$ and $\tau_{\mathrm{unif}}$; they can each be arbitrarily larger than the other. On the other hand, it has been shown that $\mathsf{H} \leq D$ [4] and that $\mathsf{H} \leq 8\tau_{\mathrm{unif}}$ [20]. Therefore, either of the first two minimax lower bounds in Table 1 (which both use hard instances that are weakly communicating) imply a lower bound of $\widetilde{\Omega}\left(SA\frac{\mathsf{H}}{\varepsilon^2}\right)$ and thus the minimax optimality of our Theorem 2.

To the best of our knowledge, no prior work has considered the average-reward sample complexity of general (potentially multichain) MDPs. Existing results make assumptions at least as strong as weakly communicating or uniformly bounded mixing times.

| Method | Sample Complexity | Reference | Comments |
|---|---|---|---|
| Primal-Dual SMD | $\widetilde{O}\left(SA\frac{\tau_{\text{unif}}^2}{\varepsilon^2}\right)$ | [8] | requires uniform mixing |
| Reduction to DMDP | $\widetilde{O}\left(SA\frac{\tau_{\text{unif}}}{\varepsilon^3}\right)$ | [9] | requires uniform mixing |
| Policy Mirror Descent | $\widetilde{O}\left(SA\frac{\tau_{\text{unif}}^3}{\varepsilon^2}\right)$ | [13] | requires uniform mixing |
| Reduction to DMDP | $\widetilde{O}\left(SA\frac{\tau_{\text{unif}}}{\varepsilon^2}\right)$ | [22] | requires uniform mixing |
| Reduction to DMDP | $\widetilde{O}\left(SA\frac{\mathsf{H}}{\varepsilon^3}\right)$ | [20] | weakly communicating |
| Refined Q-Learning | $\widetilde{O}\left(SA\frac{\mathsf{H}^2}{\varepsilon^2}\right)$ | [26] | weakly communicating |
| Reduction to DMDP | $\widetilde{O}\left(SA\frac{\mathsf{H}}{\varepsilon^2}\right)$ | Our Theorem 2 | weakly communicating |
| Reduction to DMDP | $\widetilde{O}\left(SA\frac{\mathsf{B}+\mathsf{H}}{\varepsilon^2}\right)$ | Our Theorem 8 | general MDPs |
| Lower Bound | $\widetilde{\Omega}\left(SA\frac{\tau_{\text{unif}}}{\varepsilon^2}\right)$ | [9] | implies $\widetilde{\Omega}\left(SA\frac{\mathsf{H}}{\varepsilon^2}\right)$ |
| Lower Bound | $\widetilde{\Omega}\left(SA\frac{D}{\varepsilon^2}\right)$ | [20] | implies $\widetilde{\Omega}\left(SA\frac{\mathsf{H}}{\varepsilon^2}\right)$ |
| Lower Bound | $\widetilde{\Omega}\left(SA\frac{\mathsf{B}+\mathsf{H}}{\varepsilon^2}\right)$ | Our Theorem 4 | general MDPs |

Table 1: **Algorithms and sample complexity bounds for average reward MDPs** with $S$ states and $A$ actions. The goal is finding an $\varepsilon$-optimal policy under a generative model. Here $\mathsf{H} := \|h^\star\|_{\text{span}}$ is the span of the optimal bias, $\tau_{\text{unif}}$ is a uniform upper bound on mixing times of all policies, and $D$ is the MDP diameter, with the relationships $\mathsf{H} \leq 8\tau_{\text{unif}}$ and $\mathsf{H} \leq D$. $\mathsf{B}$ is the transient time parameter.

The work [9] was the first to develop an algorithm based on reduction to a discounted MDP with a discount factor of $\gamma = 1 - \frac{\varepsilon}{\tau_{\text{unif}}}$. Their argument was improved in [20], which improved the uniform mixing assumption to only assuming a weakly communicating MDP, and used a smaller discount factor $\gamma = 1 - \frac{\varepsilon}{\mathsf{H}}$. These arguments both make essential use of the fact that the optimal gain is independent of the starting state, which does not hold for general MDPs. After analyzing the reductions, both [9] and [20] then solved the discounted MDPs by appealing to the algorithm from [12]. To the best of our knowledge, the algorithm of [12] is the only known algorithm for discounted MDPs which could work with either reduction, as the reductions each require a $\frac{\varepsilon}{1-\gamma}$-optimal policy from the discounted MDP, and other known algorithms for discounted MDPs do not permit such large suboptimality levels. (We discuss algorithms for discounted MDPs in more detail below.) Other algorithms for average-reward MDPs are considered in [9, 13, 26]. The above results fall short of matching the minimax lower bounds.

While preparing this manuscript, we became aware of [22], which considers the uniform mixing setting and obtains a minimax optimal sample complexity $\widetilde{O}\left(SA\frac{\tau_{\text{unif}}}{\varepsilon^2}\right)$ in terms of $\tau_{\text{unif}}$. Although developed independently, their work and ours have several similarities. We both utilize discounted reductions and observe that it is possible to improve the sample complexity of the resulting DMDP task by improving the analysis of variance parameters. They accomplish the improvement by leveraging the uniform mixing assumption, whereas we make use of the low span of the optimal policy. Note that $\mathsf{H} \leq 8\tau_{\text{unif}}$ holds in general and there exist MDPs with $\mathsf{H} \ll \tau_{\text{unif}} = \infty$, so our Theorem 2 is strictly stronger than the result of [22].

## 1.2 Comparison with related work on discounted MDPs

We discuss a subset of results for discounted MDPs in the generative setting. Several works [15, 19, 1, 12] obtain the minimax optimal sample complexity of $\widetilde{O}\left(SA\frac{1}{(1-\gamma)^3\varepsilon^2}\right)$ for finding an $\varepsilon$-optimal policy w.r.t. the discounted reward. However, only [12] is able to show this bound for the full range of $\varepsilon \in (0, \frac{1}{1-\gamma}]$. As mentioned, the reduction from average reward MDPs requires a large $\varepsilon$ in the resulting discounted MDP, making it unsurprising that all of [9, 20, 22] as well as our Algorithm 1 essentially use their algorithm. The matching lower bound is established in [15, 3].

As mentioned earlier, both we and the authors of [22, 21] independently observed that the $\widetilde{\Omega}\left(SA\frac{1}{(1-\gamma)^3\varepsilon^2}\right)$ sample complexity lower bound can be circumvented in the settings that arise

under the average-to-discounted reductions. The authors of [22, 21] assume uniform mixing and obtain a discounted MDP sample complexity of $\widetilde{O}\big(SA\frac{\tau_{\text{unif}}}{(1-\gamma)^2\varepsilon^2}\big)$, first in [21] by modifying the algorithm of [19], and then in [22] under a wider range of $\varepsilon$ by instead modifying the analysis of [12]. The work [21] also proves a matching lower bound. Our Theorem 1 for discounted MDPs attains a sample complexity of $\widetilde{O}\big(SA\frac{\mathsf{H}}{(1-\gamma)^2\varepsilon^2}\big)$ assuming only that the MDP is weakly communicating. Again, in light of the relationship that $\mathsf{H} \leq 8\tau_{\text{unif}}$, our results are strictly better (ignoring constants), and their lower bound also establishes the optimality of our Theorem 1.

## 2   Problem setup and preliminaries

A Markov decision process (MDP) is given by a tuple $(\mathcal{S}, \mathcal{A}, P, r)$, where $\mathcal{S}$ is the finite set of states, $\mathcal{A}$ is the finite set of actions, $P : \mathcal{S} \times \mathcal{A} \to \Delta(\mathcal{S})$ is the transition kernel with $\Delta(\mathcal{S})$ denoting the probability simplex over $\mathcal{S}$, and $r : \mathcal{S} \times \mathcal{A} \to [0, 1]$ is the reward function. Let $S := |\mathcal{S}|$ and $A := |\mathcal{A}|$ denote the cardinality of the state and action spaces, respectively. Unless otherwise noted, all policies considered are stationary Markovian policies of the form $\pi : \mathcal{S} \to \Delta(\mathcal{A})$. For any initial state $s_0 \in \mathcal{S}$ and policy $\pi$, we let $\mathbb{E}_{s_0}^\pi$ denote the expectation with respect to the probability distribution over trajectories $(S_0, A_0, S_1, A_1, \dots)$ where $S_0 = s_0$, $A_t \sim \pi(S_t)$, and $S_{t+1} \sim P(\cdot \mid S_t, A_t)$. Equivalently, this is the expectation with respect to the Markov chain induced by $\pi$ starting in state $s_0$, with the transition probability matrix $P_\pi$ given by $(P_\pi)_{s,s'} := \sum_{a \in \mathcal{A}} \pi(a|s)P(s' \mid s, a)$. We also define $(r_\pi)_s := \sum_{a \in \mathcal{A}} \pi(a|s)r(s, a)$. We occasionally treat $P$ as an $(\mathcal{S} \times \mathcal{A})$-by-$\mathcal{S}$ matrix where $P_{sa,s'} = P(s, a, s')$. We also let $P_{sa}$ denote the row vector such that $P_{sa}(s') = P(s, a, s')$. For any $s \in \mathcal{S}$ and any bounded function $X$ of the trajectory, we define the variance $\mathbb{V}_s^\pi[X] := \mathbb{E}_s^\pi(X - \mathbb{E}_s^\pi[X])^2$, with its vector version $\mathbb{V}^\pi[X] \in \mathbb{R}^\mathcal{S}$ given by $(\mathbb{V}^\pi[X])_s = \mathbb{V}_s^\pi[X]$. For $s \in \mathcal{S}$, let $e_s \in \mathbb{R}^\mathcal{S}$ be the vector that is all 0 except for a 1 in entry $s$. Let $\mathbf{1} \in \mathbb{R}^\mathcal{S}$ be the all-one vector. For each $v \in \mathbb{R}^\mathcal{S}$, define the span semi-norm $\|v\|_{\text{span}} := \max_{s \in \mathcal{S}} v(s) - \min_{s \in \mathcal{S}} v(s)$.

**Discounted reward criterion**  A discounted MDP is a tuple $(\mathcal{S}, \mathcal{A}, P, r, \gamma)$, where $\gamma \in (0, 1)$ is the discount factor. For a stationary policy $\pi$, the (discounted) value function $V_\gamma^\pi : \mathcal{S} \to [0, \infty)$ is defined, for each $s \in \mathcal{S}$, as $V_\gamma^\pi(s) := \mathbb{E}_s^\pi[\sum_{t=0}^\infty \gamma^t R_t]$, where $R_t = r(S_t, A_t)$ is the reward received at time $t$. It is well-known that there exists an optimal policy $\pi_\gamma^\star$ that is deterministic and satisfies $V_\gamma^{\pi_\gamma^\star}(s) = V_\gamma^\star(s) := \sup_\pi V_\gamma^\pi(s)$ for all $s \in \mathcal{S}$ [14]. In discounted MDPs the goal is to compute an $\varepsilon$-optimal policy, which we define as a policy $\pi$ satisfying $\|V_\gamma^\pi - V_\gamma^\star\|_\infty \leq \varepsilon$. We define one more variance parameter $\mathbb{V}_{P_\pi}[V_\gamma^\pi] \in \mathbb{R}^\mathcal{S}$, specific to a given policy $\pi$, by $(\mathbb{V}_{P_\pi}[V_\gamma^\pi])_s := \sum_{s' \in \mathcal{S}}(P_\pi)_{s,s'}[V_\gamma^\pi(s') - \sum_{s''}(P_\pi)_{s,s''}V_\gamma^\pi(s'')]^2$.

**Average-reward criterion**  In an MDP $(\mathcal{S}, \mathcal{A}, P, r)$, the average reward per stage or the *gain* of a policy $\pi$ starting from state $s$ is defined as $\rho^\pi(s) := \lim_{T \to \infty} \frac{1}{T}\mathbb{E}_s^\pi[\sum_{t=0}^{T-1} R_t]$. The *bias function* of any stationary policy $\pi$ is $h^\pi(s) := \text{C-lim}_{T \to \infty} \mathbb{E}_s^\pi[\sum_{t=0}^{T-1}(R_t - \rho^\pi(S_t))]$, where C-lim denotes the Cesaro limit. When the Markov chain induced by $P_\pi$ is aperiodic, C-lim can be replaced with the usual limit. For any policy $\pi$, its $\rho^\pi$ and $h^\pi$ satisfy $\rho^\pi = P_\pi \rho^\pi$ and $\rho^\pi + h^\pi = r_\pi + P_\pi h^\pi$.

A policy $\pi^\star$ is Blackwell-optimal if there exists some discount factor $\bar{\gamma} \in (0, 1)$ such that for all $\gamma \geq \bar{\gamma}$ we have $V_\gamma^{\pi^\star} \geq V_\gamma^\pi$ for all policies $\pi$. Henceforth we let $\pi^\star$ denote some fixed Blackwell-optimal policy, which is guaranteed to exist when $S$ and $A$ are finite [14]. We define the optimal gain $\rho^\star \in \mathbb{R}^\mathcal{S}$ by $\rho^\star(s) = \sup_\pi \rho^\pi(s)$ and note that we have $\rho^\star = \rho^{\pi^\star}$. For all $s \in \mathcal{S}$, $\rho^\star(s) \geq \max_{a \in \mathcal{A}} P_{sa}\rho^\star$, or equivalently $\rho^\star \geq P_\pi \rho^\star$ for all policies $\pi$ (and this maximum is achieved by $\pi^\star$). We also define $h^\star = h^{\pi^\star}$ (and we note that this definition does not depend on which Blackwell-optimal $\pi^\star$ is used, if there are multiple). For all $s \in \mathcal{S}$, $\rho^\star$ and $h^\star$ satisfy $\rho^\star(s) + h^\star(s) = \max_{a \in \mathcal{A}:P_{sa}\rho^\star=\rho^\star(s)} r_{sa} + P_{sa}h^\star$, known as the (unmodified) Bellman equation.

A weakly communicating MDP is such that the states can be partitioned into two disjoint subsets $\mathcal{S} = \mathcal{S}_1 \cup \mathcal{S}_2$ such that all states in $\mathcal{S}_1$ are transient under any stationary policy and within $\mathcal{S}_2$, any state is reachable from any other state under some stationary policy. In weakly communicating MDPs $\rho^\star$ is a constant vector (all entries are equal), and thus $(\rho^\star, h^\star)$ are also a solution to the modified Bellman equation $\rho^\star(s) + h^\star(s) = \max_{a \in \mathcal{A}} r_{sa} + P_{sa}h^\star$. When discussing weakly communicating MDPs we occasionally abuse notation and treat $\rho^\star$ as a scalar. A stationary policy is multichain if it

induces multiple closed irreducible recurrent classes, and an MDP is called multichain if it contains such a policy. Weakly-communicating MDPs always contain some gain-optimal policy which is unichain (not multichain), but in general MDPs, all gain-optimal policies may be multichain and $\rho^\star$ may not be a constant vector. All uniformly mixing MDPs are weakly communicating. In the average reward setting, our goal is find an $\varepsilon$-optimal policy, defined as a policy $\pi$ such that $\|\rho^\star - \rho^\pi\|_\infty \le \varepsilon$.

**Complexity parameters**  Our most important complexity parameter is the span of the optimal bias function $\mathsf{H} := \|h^\star\|_{\mathrm{span}}$. In addition, for general MDPs we introduce a new *transient time parameter* $\mathsf{B}$, defined as follows. Let $\Pi$ be the set of deterministic stationary policies. For each $\pi \in \Pi$, let $\mathcal{R}^\pi$ be the set of states which are recurrent in the Markov chain $P_\pi$, and let $\mathcal{T}^\pi = \mathcal{S} \setminus \mathcal{R}^\pi$ be the set of transient states. Let $T_{\mathcal{R}^\pi} = \inf\{t : S_t \in \mathcal{R}^\pi\}$ be the first hitting time of a state which is recurrent under $\pi$. We say an MDP satisfies the *bounded transient time property with parameter* $\mathsf{B}$ if for all policies $\pi$ and states $s \in \mathcal{S}$ we have $\mathbb{E}_s^\pi[T_{\mathcal{R}^\pi}] \le \mathsf{B}$, or in words, the expected time spent in transient states (with respect to the Markov chain induced by $\pi$) is bounded by $\mathsf{B}$.

We recall several other parameters used in the literature to characterize sample complexity. The diameter is defined as $D := \max_{s_1 \ne s_2} \inf_{\pi \in \Pi} \mathbb{E}_{s_1}^\pi[\eta_{s_2}]$, where $\eta_s$ denotes the hitting time of a state $s \in \mathcal{S}$. For each policy $\pi$, if the Markov chain induced by $P_\pi$ has a unique stationary distribution $\nu_\pi$, we define the mixing time of $\pi$ as $\tau_\pi := \inf\left\{t \ge 1 : \max_{s \in \mathcal{S}} \left\|e_s^\top (P_\pi)^t - \nu_\pi^\top\right\|_1 \le \frac{1}{2}\right\}$. If all policies $\pi \in \Pi$ satisfy this assumption, we define the uniform mixing time $\tau_{\mathrm{unif}} := \sup_{\pi \in \Pi} \tau_\pi$. Note that $D$ and $\tau_{\mathrm{unif}}$ are generally incomparable [20], while we always have $\mathsf{H} \le D$ [4] and $\mathsf{H} \le 8\tau_{\mathrm{unif}}$ [20]. It is possible for $\tau_{\mathrm{unif}} = \infty$, for instance if there are any policies which induce periodic Markov chains. Also, $D = \infty$ if there are any states which are transient under all policies. However, $\mathsf{H}$ and $\mathsf{B}$ are finite in any MDP with $S, A < \infty$. Also if $\tau_{\mathrm{unif}}$ is finite, Lemma 27 shows $\mathsf{B} \le 4\tau_{\mathrm{unif}}$.

We assume access to a generative model [10], also known as a simulator. This means we can obtain independent samples from $P(\cdot \mid s, a)$ for any given $s \in \mathcal{S}, a \in \mathcal{A}$, but $P$ itself is unknown. We assume the reward function $r$ is deterministic and known, which is standard in generative settings (e.g., [1, 12]) since otherwise estimating the mean rewards is relatively easy. Specifically, to learn an $\varepsilon$-optimal policy for the discounted MDP, we would need to estimate each entry of $r$ to accuracy $O((1-\gamma)\varepsilon)$, which requires a lower order number of samples $\widetilde{O}\left(\frac{SA}{(1-\gamma)^2 \varepsilon^2}\right)$. For this reason we assume (as in [20]) that $\mathsf{H} \ge 1$. Using samples from the generative model, our Algorithm 1 constructs an empirical transition kernel $\widehat{P}$. For a policy $\pi$, we use $\widehat{V}_\gamma^\pi(s)$ to denote the value function computed with respect to the Markov chain with transition matrix $\widehat{P}_\pi$ (as opposed to $P_\pi$). Our Algorithm 1 also utilizes a perturbed reward function $\widetilde{r}$, and we use the notation $V_{\gamma,\mathrm{p}}^\pi(s)$ to denote a value function computed using this reward (and $P_\pi$); more concretely, we replace $R_t$ with $\widetilde{R}_t = \widetilde{r}(S_t, A_t)$ in the definition above of $V_\gamma^\pi$. We use the notation $\widehat{V}_{\gamma,\mathrm{p}}^\pi$ when using $\widehat{P}$ and $\widetilde{r}$ simultaneously.

## 3  Main results for weakly communicating MDPs

Our approach is based on reducing the average-reward problem to a discounted problem. We first present our algorithm and guarantees for the discounted MDP setting. As discussed in Subsection 1.1, our algorithm of choice, Algorithm 1, is essentially the same as the one presented in [12], with a slightly different perturbation level $\xi$. Algorithm 1 constructs an empirical transition kernel $\widehat{P}$ using $n$ samples per state-action pair from the generative model, and then solves the resulting empirical (perturbed) MDP $(\widehat{P}, \widetilde{r}, \gamma)$. As noted in [12], the perturbation ensures $\widehat{\pi}_{\gamma,\mathrm{p}}^\star$ can be computed exactly in $\mathrm{poly}(\frac{1}{1-\gamma}, S, A, \log(1/\delta\varepsilon))$ time by multiple standard MDP solvers. We remark in passing that the $SA$-by-$S$ transition matrix $\widehat{P}$ has at most $nSA$ nonzero entries.

Our Theorem 1 provides an improved sample complexity bound for Algorithm 1 under the setting that the MDP is weakly communicating.

**Theorem 1** (Sample Complexity of Weakly Communicating DMDP)**.** *Suppose the discounted MDP $(P, r, \gamma)$ is weakly communicating, $\mathsf{H} \le \frac{1}{1-\gamma}$, and $\varepsilon \le \mathsf{H}$. There exists a constant $C_2 > 0$ such that, for any $\delta \in (0, 1)$, if $n \ge C_2 \frac{\mathsf{H}}{(1-\gamma)^2 \varepsilon^2} \log\left(\frac{SA}{(1-\gamma)\delta\varepsilon}\right)$, then with probability at least $1 - \delta$, the policy $\widehat{\pi}_{\gamma,\mathrm{p}}^\star$ output by Algorithm 1 satisfies $\left\|V_\gamma^\star - V_\gamma^{\widehat{\pi}_{\gamma,\mathrm{p}}^\star}\right\|_\infty \le \varepsilon$.*

---

**Algorithm 1** Perturbed Empirical Model-Based Planning

---

**input:** Sample size per state-action pair $n$, target accuracy $\varepsilon$, discount factor $\gamma$
 1: **for** each state-action pair $(s, a) \in \mathcal{S} \times \mathcal{A}$ **do**
 2:      Collect $n$ samples $S_{s,a}^1, \ldots, S_{s,a}^n$ from $P(\cdot \mid s, a)$
 3:      Form the empirical transition kernel $\widehat{P}(s' \mid s, a) = \frac{1}{n} \sum_{i=1}^n \mathbb{I}\{S_{s,a}^i = s'\}$, for all $s' \in \mathcal{S}$
 4: **end for**
 5: Set perturbation level $\xi = (1 - \gamma)\varepsilon/6$
 6: Form perturbed reward $\widetilde{r} = r + Z$ where $Z(s, a) \overset{\text{i.i.d.}}{\sim} \text{Unif}(0, \xi)$
 7: Compute a policy $\widehat{\pi}_{\gamma,\text{p}}^\star$ which is optimal for the perturbed empirical discounted MDP $(\widehat{P}, \widetilde{r}, \gamma)$
 8: **return** $\widehat{\pi}_{\gamma,\text{p}}^\star$

---

Since we observe $n$ samples for each state-action pair, Theorem 1 shows that a total number of $\widetilde{O}\big(\frac{\mathsf{H}SA}{(1-\gamma)^2\varepsilon^2}\big)$ samples suffices to learn an $\varepsilon$-optimal policy. This bound improves on the $\widetilde{O}\big(\frac{SA}{(1-\gamma)^3\varepsilon^2}\big)$ complexity bound from [12] when the span $\mathsf{H}$ is no larger than the effective horizon $\frac{1}{1-\gamma}$. This assumption holds in many situations, as can be seen by using the relationships $\mathsf{H} \le D$ or $\mathsf{H} \le 8\tau_{\text{unif}}$. On the other hand, in the regime with $\mathsf{H} > \frac{1}{1-\gamma}$, the existing bound $\widetilde{O}\big(\frac{SA}{(1-\gamma)^3\varepsilon^2}\big)$, also achieved by Algorithm 1, is superior. In this regime, the discounting effectively truncates the MDP at a short horizon $\frac{1}{1-\gamma}$ before the long-run behavior of the optimal policy (as captured by $\mathsf{H}$) kicks in.

*Proof highlights for Theorem 1.* The key to obtaining this improved complexity is a careful analysis of certain instance-specific variance parameters. It suffices to bound $\left\|\widehat{V}_{\gamma,\text{p}}^{\pi_\gamma^\star} - V_\gamma^{\pi_\gamma^\star}\right\|_\infty$ and $\left\|\widehat{V}_{\gamma,\text{p}}^{\widehat{\pi}_{\gamma,\text{p}}^\star} - V_\gamma^{\widehat{\pi}_{\gamma,\text{p}}^\star}\right\|_\infty$ by $O(\varepsilon)$. The prior DMDP complexity of $\frac{SA}{(1-\gamma)^3\varepsilon^2}$ is obtained using the well-known law-of-total-variance argument [3, 1, 12], which ultimately yields a sample complexity like $\widetilde{O}\left(\sqrt{\frac{SA}{(1-\gamma)\varepsilon^2} \left\|\mathbb{V}^{\pi_\gamma^\star}\left[\sum_{t=0}^\infty \gamma^t R_t\right]\right\|_\infty}\right)$ to bound $\left\|\widehat{V}_{\gamma,\text{p}}^{\pi_\gamma^\star} - V_\gamma^{\pi_\gamma^\star}\right\|_\infty \le O(\varepsilon)$. From here, the variance of the cumulative discounted reward $\left\|\mathbb{V}^{\pi_\gamma^\star}\left[\sum_{t=0}^\infty \gamma^t R_t\right]\right\|_\infty$ is bounded by $\frac{1}{(1-\gamma)^2}$, since the total reward in a trajectory is within $[0, \frac{1}{1-\gamma}]$. We instead seek to bound $\left\|\mathbb{V}^{\pi_\gamma^\star}\left[\sum_{t=0}^\infty \gamma^t R_t\right]\right\|_\infty \le O\left(\frac{\mathsf{H}}{1-\gamma}\right)$. Assume $\mathsf{H}$ is an integer. The first step is to decompose $\mathbb{V}^{\pi_\gamma^\star}\left[\sum_{t=0}^\infty \gamma^t R_t\right]$ recursively like

$$\mathbb{V}^{\pi_\gamma^\star}\left[\sum_{t=0}^\infty \gamma^t R_t\right] = \mathbb{V}^{\pi_\gamma^\star}\left[\sum_{t=0}^{\mathsf{H}-1} \gamma^t R_t + \gamma^\mathsf{H} V_\gamma^{\pi_\gamma^\star}(S_\mathsf{H})\right] + \gamma^{2\mathsf{H}}\left(P_{\pi_\gamma^\star}\right)^\mathsf{H}\mathbb{V}^{\pi_\gamma^\star}\left[\sum_{t=0}^\infty \gamma^t R_t\right]$$

(see our Lemma 13). This is a multi-step version of the standard variance Bellman equation (e.g., [16, Theorem 1]). Ordinarily an $\mathsf{H}$-step expansion would not be useful, since the term $V_\gamma^{\pi_\gamma^\star}(S_\mathsf{H})$ by itself appears to have fluctuations on the order of $\frac{1}{1-\gamma}$ in the worst case depending on $S_\mathsf{H}$ (note $S_\mathsf{H}$ is the random state encountered at time $\mathsf{H}$). However, in our setting, we should have $V_\gamma^{\pi_\gamma^\star}(S_\mathsf{H}) \approx \frac{1}{1-\gamma}\rho^\star + h^\star(S_\mathsf{H})$, reducing the magnitude of the random fluctuations to order $\mathsf{H} = \|h^\star\|_{\text{span}}$. (See Lemma 11 for a formalization of this approximation which first appeared in [23].) Therefore expansion to $\mathsf{H}$ steps achieves the optimal tradeoff between maintaining $\mathbb{V}^{\pi_\gamma^\star}\left[\sum_{t=0}^{\mathsf{H}-1} \gamma^t R_t + \gamma^\mathsf{H} V_\gamma^{\pi_\gamma^\star}(S_\mathsf{H})\right] \le O\left(\mathsf{H}^2\right)$ and minimizing $\gamma^{2\mathsf{H}}$. As desired this yields $\left\|\mathbb{V}^{\pi_\gamma^\star}\left[\sum_{t=0}^\infty \gamma^t R_t\right]\right\|_\infty \le O\big(\frac{\mathsf{H}^2}{1-\gamma^{2\mathsf{H}}}\big) = O\big(\frac{\mathsf{H}}{1-\gamma}\big)$, where $\frac{1}{1-\gamma^{2\mathsf{H}}} \le O\big(\frac{1}{\mathsf{H}(1-\gamma)}\big)$ requires $\frac{1}{1-\gamma} \ge \mathsf{H}$. See Lemma 15 for the complete argument.

We would like to use a similar argument as above to bound the second term $\left\|\widehat{V}_{\gamma,\text{p}}^{\widehat{\pi}_{\gamma,\text{p}}^\star} - V_\gamma^{\widehat{\pi}_{\gamma,\text{p}}^\star}\right\|_\infty$, which is the "evaluation error" of the *empirically* optimal policy $\widehat{\pi}_{\gamma,\text{p}}^\star$. However, applying the same argument would give a bound in terms of $\left\|V_\gamma^{\widehat{\pi}_{\gamma,\text{p}}^\star}\right\|_{\text{span}}$, which, unlike for the analogous term involving the *true* optimal policy $\pi_\gamma^\star$, is not a priori bounded in terms of $H$. (If we instead assumed uniform mixing, we could immediately bound this by $O(\tau_{\text{unif}})$.) Thus, to control the variance associated with evaluating $\widehat{\pi}_{\gamma,\text{p}}^\star$, we are able to recursively bound $\left\|V_\gamma^{\widehat{\pi}_{\gamma,\text{p}}^\star}\right\|_{\text{span}} \le O\big(H + \left\|\widehat{V}_{\gamma,\text{p}}^{\widehat{\pi}_{\gamma,\text{p}}^\star} - V_\gamma^{\widehat{\pi}_{\gamma,\text{p}}^\star}\right\|_\infty\big)$, which can be shown to yield the desired sample complexity. $\qquad\square$

Now we present our main result for the average-reward problem in the weakly communicating setting. Applied in this setting with a DMDP target accuracy of $\overline{\varepsilon} = \mathsf{H}$, our Algorithm 2 reduces the problem to $\overline{\gamma}$-discounted MDP with $\overline{\gamma} = 1 - \frac{\varepsilon}{12\mathsf{H}}$ and then calls Algorithm 1 with target accuracy $\mathsf{H}$.

---

**Algorithm 2** Average-to-Discount Reduction

---

**input:** Sample size per state-action pair $n$, target accuracy $\varepsilon \in (0, 1]$, DMDP target accuracy $\overline{\varepsilon}$
  1: Set $\overline{\gamma} = 1 - \frac{\varepsilon}{12\overline{\varepsilon}}$
  2: Obtain $\widehat{\pi}^{\star}$ from Algorithm 1 with sample size per state-action pair $n$, accuracy $\overline{\varepsilon}$, discount $\overline{\gamma}$
  3: **return** $\widehat{\pi}^{\star}$

---

We have the following sample complexity bound for Algorithm 2.

**Theorem 2** (Sample Complexity of Weakly Communicating AMDP). *Suppose the MDP $(P, r)$ is weakly communicating. There exists a constant $C_1 > 0$ such that for any $\delta, \varepsilon \in (0, 1)$, if $n \geq C_1 \frac{\mathsf{H}}{\varepsilon^2} \log\left(\frac{SA\mathsf{H}}{\delta\varepsilon}\right)$ and we call Algorithm 2 with $\overline{\varepsilon} = \mathsf{H}$, then with probability at least $1 - \delta$, the output policy $\widehat{\pi}^{\star}$ satisfies the elementwise inequality $\rho^{\star} - \rho^{\widehat{\pi}^{\star}} \leq \varepsilon \mathbf{1}$.*

Again, since we observe $n$ samples for each state-action pair, this result shows that $\widetilde{O}\left(\frac{\mathsf{H}SA}{\varepsilon^2}\right)$ total samples suffice to learn an $\varepsilon$-optimal policy for the average reward MDP. This bound matches the minimax lower bound in [20] and is superior to existing results for weakly communicating MDPs (see Table 1). We note that the proof of Theorem 1 works so long as $\mathsf{H}$ is any upper bound of $\|h^{\star}\|_{\mathrm{span}}$, hence Algorithm 2 also only needs an upper bound for $\|h^{\star}\|_{\mathrm{span}}$.

We show in the following theorem that it is in general impossible to obtain a useful upper bound on $\|h^{\star}\|_{\mathrm{span}}$ with a sample complexity that is a function of only $\|h^{\star}\|_{\mathrm{span}}$. This suggests that it is not easy to remove the need for knowledge of $\|h^{\star}\|_{\mathrm{span}}$.

**Theorem 3.** *For any given $n, T \geq 1$, there exist two MDPs $\mathcal{M}_0$ and $\mathcal{M}_1$ with $S = 4$, $A = 1$ such that $\mathcal{M}_0$ has optimal bias span $1$, $\mathcal{M}_1$ has optimal bias span $T$, and it is impossible to distinguish between $\mathcal{M}_0$ and $\mathcal{M}_1$ with probability $\geq \frac{3}{4}$ with $n$ samples from each state-action pair.*

Thus even for an MDP with a small span, there exists another MDP that has an arbitrarily large span and is arbitrarily statistically close (that is, cannot be distinguished even with a large sample size $n$). We emphasize that all previous algorithms in Table 1 also require knowledge of their respective complexity parameters, and such assumptions are pervasive throughout the literature on average-reward RL. The only exception of which we are aware is the contemporaneous work [7], which achieves a suboptimal $\widetilde{O}(SA\frac{\tau_{\mathrm{unif}}^8}{\varepsilon^8})$ sample complexity without knowledge of $\tau_{\mathrm{unif}}$ in the uniformly mixing setting. It is unclear if $\mathsf{H}$-based sample complexities are possible without knowing $\mathsf{H}$. Besides the evidence offered by Theorem 3, in the online setting, it has been conjectured that knowledge of $\mathsf{H}$ is necessary to obtain an $\mathsf{H}$-dependent regret bound [6, 5, 25]. Moreover, even with knowledge of $\mathsf{H}$, the only known online algorithm with optimal regret is computationally inefficient [25], making it somewhat surprising that our Theorem 2 uses a simple and efficient algorithm.

Nevertheless, when $\mathsf{H}$ is unknown, one can replace $\mathsf{H}$ with the diameter $D$ (since $\mathsf{H} \leq D$). The diameter is known to be estimable [25, 17] and is often a more refined complexity parameter than $\tau_{\mathrm{unif}}$. Our Theorem 2 is the first to imply the optimal diameter-based complexity $\widetilde{O}(\frac{SAD}{\varepsilon^2})$, given knowledge of $D$ or using a constant-factor upper bound obtained from some estimation procedure.

## 4 Main results for general MDPs

Our starting point for general MDPs is that unlike the weakly communicating setting, their complexity *cannot* be captured solely by $\|h^{\star}\|_{\mathrm{span}}$. We first argue this point informally using the simple example in Figure 1, which is parameterized by a value $T > 1$. Only state $1$ contains multiple actions, and action $2$ is optimal since it leads to state $2$ which collects reward $0.5$ forever, while taking action $1$ will always eventually lead to state $3$ where the reward is $0$ forever. We thus have $\rho^{\star} = [0.5, 0.5, 0]^{\top}$ and $\|h^{\star}\|_{\mathrm{span}} = 0$. However, clearly $\Omega(T)$ samples are required to even observe a transition $1 \to 3$, so the sample complexity must depend on $T \gg \mathsf{H}$ (without observing a transition $1 \to 3$, we cannot determine that action $1$ is not optimal). Taking action $1$ leads to a large reward of $1$ in the short

term (for $T$ steps in expectation), so even if we had perfect knowledge of the environment, the optimal $\gamma$-discounted policy would not choose the optimal action $a = 2$ until the effective horizon $\frac{1}{1-\gamma} \geq \Omega(T)$. Thus $\frac{1}{1-\gamma} \approx \mathsf{H}$ is insufficient for the reduction to discounted MDP. Note that this instance has its bounded transient time parameter $\mathsf{B} = T$. This example reflects that transient states play a categorically different role in general MDPs: in the weakly communicating setting, states which are transient under all policies can be completely ignored, whereas in this example our action at state $1$ fully determines our reward even though state $1$ is transient under all policies.

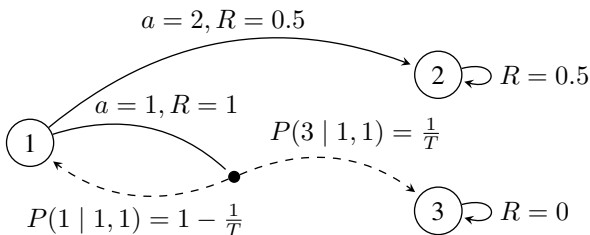

Figure 1: A general MDP where $\gamma$-discounted approximation fails unless $\frac{1}{1-\gamma} = \Omega(T) \gg \|h^\star\|_{\mathrm{span}}$.

The statistical hardness is formally captured by the following theorem, which uses improved instances to obtain the correct dependence on $\varepsilon$.

**Theorem 4** (Lower Bound for General AMDPs). *For any $\varepsilon \in (0, 1/4)$, $B \geq 1$, $A \geq 4$ and $S \in 8\mathbb{N}$, for any algorithm Alg which is guaranteed to return an $\varepsilon/3$-optimal policy for any input average-reward MDP with probability at least $\frac{3}{4}$, there exists an MDP $\mathcal{M} = (P, r)$ such that:*

1. *$\mathcal{M}$ has $S$ states and $A$ actions.*

2. *Letting $h^\star$ be the bias of the Blackwell-optimal policy for $\mathcal{M}$, we have $\|h^\star\|_{\mathrm{span}} = 0$.*

3. *$\mathcal{M}$ satisfies the bounded transient time assumption with parameter $B$.*

4. *Alg requires $\Omega\left(\frac{B\log(SA)}{\varepsilon^2}\right)$ samples per state-action pair on $\mathcal{M}$.*

A similar minimax lower bound holds for the discounted setting.

**Theorem 5** (Lower Bound for General DMDP). *For any $\varepsilon \in (0, 1/4)$, $B \geq 1$, $A \geq 4$ and $S \in 8\mathbb{N}$ for any algorithm Alg which is guaranteed to return an $\varepsilon/3$-optimal policy for any input discounted MDP with probability at least $\frac{3}{4}$, there exists a discounted MDP $\mathcal{M} = (P, r, \gamma)$ such that:*

1. *$\mathcal{M}$ has $S$ states and $A$ actions.*

2. *$\mathcal{M}$ satisfies the bounded transient time assumption with parameter $B$.*

3. *Alg requires $\Omega\left(\frac{B\log(SA)}{(1-\gamma)^2\varepsilon^2}\right)$ samples per state-action pair on $\mathcal{M}$.*

The lower bounds of $\widetilde{O}\left(\frac{\mathsf{H}}{\varepsilon^2}\right)$ from the weakly communicating setting still apply in the general setting. Together with Theorem 4 they imply a $\widetilde{O}\left(\frac{\mathsf{H}+\mathsf{B}}{\varepsilon^2}\right)$ lower bound for general average-reward MDPs.

Figure 1 demonstrates that, unlike the weakly communicating setting, discounted reduction with $\frac{1}{1-\gamma}$ set in terms of only $\mathsf{H}$ cannot succeed for general MDPs. (Contrast with Lemma 9 for the analogous theorem from [20] for weakly communicating MDPs.) We remedy this issue and lay the foundation for our matching upper bound by proving a new reduction theorem in terms of $\mathsf{H}$ *and* $\mathsf{B}$; in particular, $\mathsf{B}$ measures how much farther ahead we must look in order to determine which closed communicating class will be reached. By Lemma 27 $\mathsf{B} \leq 4\tau_{\mathrm{unif}}$, although $\mathsf{B}$ is always finite unlike $\tau_{\mathrm{unif}}$.

**Theorem 6** (Average-to-Discount Reduction for General MDP). *Suppose $(P, r)$ is a general MDP, has an optimal bias function $h^\star$ satisfying $\|h^\star\|_{\mathrm{span}} \leq \mathsf{H}$, and satisfies the bounded transient time assumption with parameter $\mathsf{B}$. Fix $\varepsilon \in (0, 1]$ and set $\gamma = 1 - \frac{\varepsilon}{\mathsf{B}+\mathsf{H}}$. For any $\varepsilon_\gamma \in [0, \frac{1}{1-\gamma}]$, if $\pi$ is any $\varepsilon_\gamma$-optimal policy for the discounted MDP $(P, r, \gamma)$, then $\rho^\star - \rho^\pi \leq \left(3 + 2\frac{\varepsilon_\gamma}{\mathsf{B}+\mathsf{H}}\right)\varepsilon\mathbf{1}$.*

*Proof highlights.* Letting $\pi^\star_\gamma$ be the optimal policy for the $\gamma$-discounted MDP, our first key observation is that $\rho^\star$ is constant within any irreducible closed recurrent block of the Markov chain $P_{\pi^\star_\gamma}$, essentially

because all states in this block must be reachable from each other with probability one (see Lemma 17). Leveraging the optimality of $\pi_\gamma^\star$, this enables us to bound both $\left|V_\gamma^{\pi_\gamma^\star}(s) - \frac{1}{1-\gamma}\rho^\star(s)\right|$ and $\left|V_\gamma^{\pi_\gamma^\star}(s) - \frac{1}{1-\gamma}\rho^{\pi_\gamma^\star}(s)\right|$ by $O\big(\|h^\star\|_{\mathrm{span}}\big)$ for any $s$ which is recurrent under $\pi_\gamma^\star$, which when combined demonstrate that the gain $\rho^{\pi_\gamma^\star}(s)$ of $\pi_\gamma^\star$ is near-optimal for its recurrent states. See Lemma 21. We then leverage the bounded transient time assumption to guarantee that for transient $s$, $V_\gamma^{\pi_\gamma^\star}(s)$ is dominated by the expected returns from recurrent states, since at most $O(\mathsf{B})$ time is spent in transient states. We complete the proof of Theorem 6 by combining these facts, as well as extending them to accommodate approximately optimal policies. $\qquad\square$

Next we establish an improved sample complexity for the discounted problem in the setting relevant to this reduction. This bound matches the lower bound in Theorem 5 up to log factors.

**Theorem 7** (Sample Complexity of General DMDP). *Suppose* $\mathsf{B} + \mathsf{H} \le \frac{1}{1-\gamma}$ *and* $\varepsilon \le \mathsf{B} + \mathsf{H}$. *There exists a constant* $C_3 > 0$ *such that, for any* $\delta \in (0,1)$, *if* $n \ge C_3 \frac{\mathsf{B}+\mathsf{H}}{(1-\gamma)^2\varepsilon^2}\log\left(\frac{SA}{(1-\gamma)\delta\varepsilon}\right)$, *then with probability* $1 - \delta$, *the policy* $\widehat\pi_{\gamma,\mathrm{p}}^\star$ *output by Algorithm 1 satisfies* $\left\|V_\gamma^\star - V_\gamma^{\widehat\pi_{\gamma,\mathrm{p}}^\star}\right\|_\infty \le \varepsilon$.

Finally, we present our result for the sample complexity of general average-reward MDPs, matching the lower bound in Theorem 4 up to log factors. We again use the reduction Algorithm 2, this time with the larger DMDP target accuracy $\overline\varepsilon = \mathsf{B} + \mathsf{H}$, leading to a discount factor of $\overline\gamma = 1 - \frac{\varepsilon}{12(\mathsf{B}+\mathsf{H})}$.

**Theorem 8** (Sample Complexity of General AMDP). *There exists a constant* $C_4 > 0$ *such that for any* $\delta, \varepsilon \in (0,1)$, *if* $n \ge C_4 \frac{\mathsf{B}+\mathsf{H}}{\varepsilon^2}\log\left(\frac{SA(\mathsf{B}+\mathsf{H})}{\delta\varepsilon}\right)$ *and we call Algorithm 2 with* $\varepsilon = \mathsf{B} + \mathsf{H}$, *then with probability at least* $1 - \delta$, *the output policy* $\widehat\pi^\star$ *satisfies the elementwise inequality* $\rho^\star - \rho^{\widehat\pi^\star} \le \varepsilon\mathbf{1}$.

*Proof highlights.* Similarly to Theorem 2, we seek to bound certain variance parameters, and this time it would suffice to bound the variance of the cumulative discounted reward starting from any state $s$ like $\left|\mathbb{V}_s^{\pi_\gamma^\star}\left[\sum_{t=0}^\infty \gamma^t R_t\right]\right| \le O\big(\frac{\mathsf{H}+\mathsf{B}}{1-\gamma}\big)$. Such a bound indeed holds for states $s$ that are recurrent under $\pi_\gamma^\star$, because $\rho^\star(S_t)$ will remain constant to $\rho^\star(s)$ for all $t$, since, as mentioned above, $\rho^\star$ is constant on closed irreducible recurrent blocks, and all $(S_t)_{t\ge0}$ will stay in the same block as $s$. Therefore, we can almost reuse our argument from the weakly communicating case. However, if $s$ is transient, it is easy to see that $\left|\mathbb{V}_s^{\pi_\gamma^\star}\left[\sum_{t=0}^\infty \gamma^t R_t\right]\right| = \Omega\big(\big(\frac{1}{1-\gamma}\big)^2\big)$ in general (even under the bounded transient time assumption), as we can consider an example where from $s$ we transition to either an absorbing reward 1 state or an absorbing reward 0 state. Thus, when $s$ is transient, instead of bounding $\left|\mathbb{V}_s^{\pi_\gamma^\star}\left[\sum_{t=0}^\infty \gamma^t R_t\right]\right|$, we directly work with the sharper variance parameter $\left|e_s^\top(I - \gamma P_{\pi_\gamma^\star})^{-1}\sqrt{\mathbb{V}_{P_{\pi_\gamma^\star}}\left[V_\gamma^{\pi_\gamma^\star}\right]}\right|$, which is also common to the analysis of DMDPs [3, 1, 12] (and in these previous works is bounded in terms of $\left\|\mathbb{V}^{\pi_\gamma^\star}\left[\sum_{t=0}^\infty \gamma^t R_t\right]\right\|_\infty$; see Lemma 12 for this relationship). We instead develop a novel law-of-total-variance-style argument which limits the total contribution of transient states to this sharper variance parameter. See Lemma 26 for details. $\qquad\square$

## 5 Conclusion

In this paper we obtained optimal sample complexities for weakly communicating and general average reward MDPs by improving the analysis of discounted MDPs, revealing a quadratic rather than cubic dependence on the effective horizon for a fixed instance. A limitation of our results (as well as of all previous results) is that the average-to-discounted reduction requires prior knowledge of parameters for optimal complexity, and an interesting open question is whether it is possible to remove this assumption. In conclusion, we believe our results shed greater light on the relationship between the discounted and average reward settings as well as the fundamental complexity of the discounted setting, and we hope that our technical developments can be useful in future work, such as leading to efficient optimal algorithms in the online setting.

## Acknowledgments and Disclosure of Funding

Y. Chen and M. Zurek were supported in part by National Science Foundation CCF-2233152 and DMS-2023239.

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

# A Proofs for weakly communicating MDPs

In this section, we provide the proofs for our main results in Section 3 for weakly communicating MDPs. Before beginning, we note that given that $\mathsf{H} \geq 1$, we may assume that $\mathsf{H}$ is an integer by setting $\mathsf{H} \leftarrow \lceil \mathsf{H} \rceil$, which only affects the sample complexity by a constant multiple $< 2$ relative to the original parameter $\mathsf{H}$. Let $\|M\|_{\infty \to \infty} := \sup_{v:\|v\|_\infty \leq 1} \|Mv\|_\infty$ denote the $\ell_\infty$ operator norm of a matrix $M$. We record the standard and useful fact that $\left\|(I - \gamma P')^{-1}\right\|_{\infty \to \infty} \leq \frac{1}{1-\gamma}$ for any transition probability matrix $P'$, which follows from the Neumann series $(I - \gamma P')^{-1} = \sum_{t \geq 0} (\gamma P')^t$ and the elementary fact that $\|P'\|_{\infty \to \infty} \leq 1$.

## A.1 Technical lemmas

First we formally state the main theorem from [20], which gives a reduction from weakly communicating average-reward problems to discounted problems.

**Lemma 9.** *Suppose $(P, r)$ is an MDP which is weakly communicating and has an optimal bias function $h^\star$ satisfying $\|h^\star\|_{\mathrm{span}} \leq \mathsf{H}$. Fix $\varepsilon \in (0, 1]$ and set $\gamma = 1 - \frac{\varepsilon}{\mathsf{H}}$. For any $\varepsilon_\gamma \in [0, \frac{1}{1-\gamma}]$, if $\pi$ is any $\varepsilon_\gamma$-optimal policy for the discounted MDP $(P, r, \gamma)$, then*

$$\rho^\star - \rho^\pi \leq \left(8 + 3\frac{\varepsilon_\gamma}{\mathsf{H}}\right)\varepsilon \mathbf{1}.$$

From here, we will first establish lemmas which are useful for proving Theorem 1 on discounted MDPs, and then we will apply the reduction approach of Lemma 9 to prove Theorem 2 on average-reward MDPs. As mentioned in the introduction, a key technical component of our approach is to establish superior bounds on a certain instance-dependent variance quantity which replace a factor of $\frac{1}{1-\gamma}$ with a factor of $\mathsf{H}$. Before reaching this step however, to make use of such a bound, we require an algorithm for discounted MDPs which enjoys a variance-dependent guarantee.

The work [12] obtains bounds with variance dependence that suffice for our purposes. However, they do not directly present said variance-dependent bounds, so we must slightly repackage their arguments in the form we require.

**Lemma 10.** *There exist absolute constants $c_1, c_2$ such that for any $\delta \in (0, 1)$, if $n \geq \frac{c_2}{1-\gamma} \log\left(\frac{SA}{(1-\gamma)\delta\varepsilon}\right)$, then with probability at least $1 - \delta$, after running Algorithm 1, we have*

$$
\left\|\widehat{V}_{\gamma,\mathrm{P}}^{\pi_\gamma^\star} - V_\gamma^{\pi_\gamma^\star}\right\|_\infty \leq \gamma \sqrt{\frac{c_1 \log\left(\frac{SA}{(1-\gamma)\delta\varepsilon}\right)}{n}} \left\|(I - \gamma P_{\pi_\gamma^\star})^{-1}\sqrt{\mathbb{V}_{P_{\pi_\gamma^\star}}\left[V_\gamma^{\pi_\gamma^\star}\right]}\right\|_\infty \tag{1}
$$
$$
+ c_1 \gamma \frac{\log\left(\frac{SA}{(1-\gamma)\delta\varepsilon}\right)}{(1-\gamma)n}\left\|V_\gamma^{\pi_\gamma^\star}\right\|_\infty + \frac{\varepsilon}{6}
$$

*and*

$$
\left\|\widehat{V}_{\gamma,\mathrm{P}}^{\widehat{\pi}_{\gamma,\mathrm{P}}^\star} - V_\gamma^{\widehat{\pi}_{\gamma,\mathrm{P}}^\star}\right\|_\infty \leq \gamma \sqrt{\frac{c_1 \log\left(\frac{SA}{(1-\gamma)\delta\varepsilon}\right)}{n}} \left\|(I - \gamma P_{\widehat{\pi}_{\gamma,\mathrm{P}}^\star})^{-1}\sqrt{\mathbb{V}_{P_{\widehat{\pi}_{\gamma,\mathrm{P}}^\star}}\left[V_{\gamma,\mathrm{P}}^{\widehat{\pi}_{\gamma,\mathrm{P}}^\star}\right]}\right\|_\infty \tag{2}
$$
$$
+ c_1 \gamma \frac{\log\left(\frac{SA}{(1-\gamma)\delta\varepsilon}\right)}{(1-\gamma)n}\left\|V_{\gamma,\mathrm{P}}^{\widehat{\pi}_{\gamma,\mathrm{P}}^\star}\right\|_\infty + \frac{\varepsilon}{6}.
$$

*Proof.* First we establish equation (1). The proof of [12, Lemma 1] shows that when $n \geq \frac{16e^2}{1-\gamma} 2\log\left(\frac{4S \log \frac{e}{1-\gamma}}{\delta}\right)$, with probability at least $1 - \delta$ we have

$$
\left\|\widehat{V}_\gamma^{\pi_\gamma^\star} - V_\gamma^{\pi_\gamma^\star}\right\|_\infty \leq 4\gamma \sqrt{\frac{2 \log\left(\frac{4S \log \frac{e}{1-\gamma}}{\delta}\right)}{n}} \left\|(I - \gamma P_{\pi_\gamma^\star})^{-1}\sqrt{\mathbb{V}_{P_{\pi_\gamma^\star}}\left[V_\gamma^{\pi_\gamma^\star}\right]}\right\|_\infty \tag{3}
$$
$$
+ \gamma \frac{2 \log\left(\frac{4S \log \frac{e}{1-\gamma}}{\delta}\right)}{(1-\gamma)n}\left\|V_\gamma^{\pi_\gamma^\star}\right\|_\infty.
$$

Now since

$$\left\|\widehat{V}_{\gamma,\mathrm{p}}^{\pi_\gamma^\star} - \widehat{V}_\gamma^{\pi_\gamma^\star}\right\|_\infty = \left\|(I - \gamma\widehat{P}_{\pi_\gamma^\star})^{-1}\widetilde{r}_{\pi_\gamma^\star} - (I - \gamma\widehat{P}_{\pi_\gamma^\star})^{-1}r_{\pi_\gamma^\star}\right\|_\infty$$

$$\leq \left\|(I - \gamma\widehat{P}_{\pi_\gamma^\star})^{-1}\right\|_{\infty\to\infty}\|\widetilde{r} - r\|_\infty$$

$$\leq \frac{\xi}{1-\gamma} = \frac{\varepsilon}{6},$$

we can obtain equation (1) by triangle inequality (although we will choose the constant $c_1$ below).

Next we establish equation (2). Using [12, Lemma 6], with probability at least $1 - \delta$ we have that

$$\left|\widehat{Q}_{\gamma,\mathrm{p}}^\star(s, \widehat{\pi}_{\gamma,\mathrm{p}}^\star(s)) - \widehat{Q}_{\gamma,\mathrm{p}}^\star(s, a)\right| > \frac{\xi\delta(1-\gamma)}{3SA^2} = \frac{\varepsilon\delta(1-\gamma)^2}{18SA^2} \tag{4}$$

uniformly over all $s$ and all $a \neq \widehat{\pi}_{\gamma,\mathrm{p}}^\star(s)$. From this separation condition (4), the assumptions of [12, Lemma 5] hold (with $\omega = \frac{\varepsilon\delta(1-\gamma)^2}{18SA^2}$ in their notation) for the MDP with the perturbed reward $\widetilde{r}$. The proof of [12, Lemma 5] shows that under the event (4) holds, the conditions for [12, Lemma 2] are satisfied (with, in their notation, $\beta_1 = 2\log\left(\frac{32}{(1-\gamma)^2\omega\delta}SA\log\frac{e}{1-\gamma}\right) = 2\log\left(\frac{576S^2A^3}{(1-\gamma)^4\delta^2\varepsilon}\log\frac{e}{1-\gamma}\right)$) with additional failure probability $\leq \delta$. The proof of [12, Lemma 2] then shows that, assuming $n > \frac{16e^2}{1-\gamma}2\log\left(\frac{576S^2A^3}{(1-\gamma)^4\delta^2\varepsilon}\log\frac{e}{1-\gamma}\right)$, we have

$$\left\|\widehat{V}_{\gamma,\mathrm{p}}^{\widehat{\pi}_{\gamma,\mathrm{p}}^\star} - V_{\gamma,\mathrm{p}}^{\widehat{\pi}_{\gamma,\mathrm{p}}^\star}\right\|_\infty \leq 4\gamma\sqrt{\frac{\beta_1}{n}}\left\|(I - \gamma P_{\widehat{\pi}_{\gamma,\mathrm{p}}^\star})^{-1}\sqrt{\mathbb{V}_{P_{\widehat{\pi}_{\gamma,\mathrm{p}}^\star}}\left[V_{\gamma,\mathrm{p}}^{\widehat{\pi}_{\gamma,\mathrm{p}}^\star}\right]}\right\|_\infty + \frac{\gamma\beta_1}{(1-\gamma)n}\left\|V_{\gamma,\mathrm{p}}^{\widehat{\pi}_{\gamma,\mathrm{p}}^\star}\right\|_\infty \tag{5}$$

where we abbreviated $\beta_1 = 2\log\left(\frac{576S^2A^3}{(1-\gamma)^4\delta^2\varepsilon}\log\frac{e}{1-\gamma}\right)$ for notational convenience.

We can again calculate that

$$\left\|V_{\gamma,\mathrm{p}}^{\widehat{\pi}_{\gamma,\mathrm{p}}^\star} - V_\gamma^{\widehat{\pi}_{\gamma,\mathrm{p}}^\star}\right\|_\infty = \left\|(I - \gamma P_{\widehat{\pi}_{\gamma,\mathrm{p}}^\star})^{-1}\widetilde{r}_{\widehat{\pi}_{\gamma,\mathrm{p}}^\star} - (I - \gamma P_{\widehat{\pi}_{\gamma,\mathrm{p}}^\star})^{-1}r_{\widehat{\pi}_{\gamma,\mathrm{p}}^\star}\right\|_\infty$$

$$\leq \left\|(I - \gamma P_{\widehat{\pi}_{\gamma,\mathrm{p}}^\star})^{-1}\right\|_{\infty\to\infty}\|\widetilde{r} - r\|_\infty$$

$$\leq \frac{\xi}{1-\gamma} = \frac{\varepsilon}{6},$$

so $\left\|\widehat{V}_{\gamma,\mathrm{p}}^{\widehat{\pi}_{\gamma,\mathrm{p}}^\star} - V_\gamma^{\widehat{\pi}_{\gamma,\mathrm{p}}^\star}\right\|_\infty \leq \left\|\widehat{V}_{\gamma,\mathrm{p}}^{\widehat{\pi}_{\gamma,\mathrm{p}}^\star} - V_{\gamma,\mathrm{p}}^{\widehat{\pi}_{\gamma,\mathrm{p}}^\star}\right\|_\infty + \frac{\varepsilon}{6}$ by triangle inequality, essentially giving (2).

Finally, to choose the constants $c_1$ and $c_2$, we first note that $2\log\left(\frac{4S\log\frac{e}{1-\gamma}}{\delta}\right) \leq \beta_1 < c_1'\log\left(\frac{SA}{(1-\gamma)\delta\varepsilon}\right)$ for some absolute constant $c_1'$, and therefore also all our requirements on $n$ are fulfilled when $n \geq \frac{16e^2}{1-\gamma}c_1'\log\left(\frac{SA}{(1-\gamma)\delta\varepsilon}\right) = \frac{c_2'}{1-\gamma}\log\left(\frac{SA}{(1-\gamma)\delta\varepsilon}\right)$ for another absolute constant $c_2'$. Lastly we note that by the union bound the total failure probability is at most $3\delta$, so to obtain a failure probability of $\delta'$ we may set $\delta = \delta'/3$ and absorb the additional constant when defining $c_1, c_2$ in terms of $c_1', c_2'$, and we also then increase $c_1$ by a factor of $4$ to absorb the factor of $4$ appearing in the first terms within (3) and (5). $\square$

Now we can analyze the variance parameters

$$\left\|(I - \gamma P_{\pi_\gamma^\star})^{-1}\sqrt{\mathbb{V}_{P_{\pi_\gamma^\star}}\left[V_\gamma^{\pi_\gamma^\star}\right]}\right\|_\infty \quad \text{and} \quad \left\|(I - \gamma P_{\widehat{\pi}_{\gamma,\mathrm{p}}^\star})^{-1}\sqrt{\mathbb{V}_{P_{\widehat{\pi}_{\gamma,\mathrm{p}}^\star}}\left[V_{\gamma,\mathrm{p}}^{\widehat{\pi}_{\gamma,\mathrm{p}}^\star}\right]}\right\|_\infty,$$

which appear in the error bounds in Lemma 10. We begin by reproducing the following inequality from [23, Lemma 2].

**Lemma 11.** *In a weakly communicating MDP, for all $\gamma \in [0, 1)$, it holds that*

$$\sup_s\left|V_\gamma^{\pi_\gamma^\star}(s) - \frac{1}{1-\gamma}\rho^\star\right| \leq \mathsf{H}.$$

The following relates the variance parameter of interest to another parameter, the variance of the total discounted rewards. This result essentially appears in [1, Lemma 4] (which was in turn inspired by [3, Lemma 8]), but since their result pertains to objects slightly different than $P_\pi$ and $\mathbb{V}_{P_\pi}\left[V_\gamma^\pi\right]$, we provide the full argument for completeness.

**Lemma 12.** *For any deterministic stationary policy $\pi$, we have*

$$\gamma \left\| (I - \gamma P_\pi)^{-1} \sqrt{\mathbb{V}_{P_\pi}\left[V_\gamma^\pi\right]} \right\|_\infty \leq \sqrt{\frac{2}{1-\gamma}} \sqrt{\left\| \mathbb{V}^\pi \left[ \sum_{t=0}^\infty \gamma^t R_t \right] \right\|_\infty}.$$

*Proof.* First we note the well-known variance Bellman equation (see for instance [16, Theorem 1]):

$$\mathbb{V}^\pi \left[ \sum_{t=0}^\infty \gamma^t R_t \right] = \gamma^2 \mathbb{V}_{P_\pi}\left[V_\gamma^\pi\right] + \gamma^2 P_\pi \mathbb{V}^\pi \left[ \sum_{t=0}^\infty \gamma^t R_t \right]. \tag{6}$$

Now we can basically identically follow the argument of [1, Lemma 4]. The matrix $(1-\gamma)(I-\gamma P_\pi)^{-1}$ has rows which are each probability distributions (are non-negative and sum to 1). Therefore, by Jensen's inequality and the concavity of the function $x \mapsto \sqrt{x}$, for each row $s \in \mathcal{S}$ we have

$$\left| (1-\gamma)e_s^\top (I - \gamma P_\pi)^{-1} \sqrt{\mathbb{V}_{P_\pi}\left[V_\gamma^\pi\right]} \right| \leq \sqrt{\left| (1-\gamma)e_s^\top (I - \gamma P_\pi)^{-1} \mathbb{V}_{P_\pi}\left[V_\gamma^\pi\right] \right|}.$$

Using this fact we can calculate that, abbreviating $v = \mathbb{V}_{P_\pi}\left[V_\gamma^\pi\right]$,

$$\gamma \left\| (I - \gamma P_\pi)^{-1} \sqrt{v} \right\|_\infty = \gamma \frac{1}{1-\gamma} \left\| (1-\gamma)(I - \gamma P_\pi)^{-1} \sqrt{v} \right\|_\infty$$

$$\leq \gamma \frac{1}{1-\gamma} \sqrt{\left\| (1-\gamma)(I - \gamma P_\pi)^{-1} v \right\|_\infty}$$

$$= \gamma \frac{1}{\sqrt{1-\gamma}} \sqrt{\left\| (I - \gamma P_\pi)^{-1} v \right\|_\infty}.$$

In order to relate $\left\| (I - \gamma P_\pi)^{-1} v \right\|_\infty$ to $\left\| (I - \gamma^2 P_\pi)^{-1} v \right\|_\infty$ in order to apply the variance Bellman equation (6), we calculate

$$\left\| (I - \gamma P_\pi)^{-1} v \right\|_\infty = \left\| (I - \gamma P_\pi)^{-1}(I - \gamma^2 P_\pi)(I - \gamma^2 P_\pi)^{-1} v \right\|_\infty$$

$$= \left\| (I - \gamma P_\pi)^{-1} \left( (1-\gamma)I + \gamma(I - \gamma P_\pi) \right)(I - \gamma^2 P_\pi)^{-1} v \right\|_\infty$$

$$= \left\| \left( (1-\gamma)(I - \gamma P_\pi)^{-1} + \gamma I \right)(I - \gamma^2 P_\pi)^{-1} v \right\|_\infty$$

$$\leq \left\| (1-\gamma)(I - \gamma P_\pi)^{-1}(I - \gamma^2 P_\pi)^{-1} v \right\|_\infty + \gamma \left\| (I - \gamma^2 P_\pi)^{-1} v \right\|_\infty$$

$$\leq (1-\gamma) \left\| (I - \gamma P_\pi)^{-1} \right\|_{\infty \to \infty} \left\| (I - \gamma^2 P_\pi)^{-1} v \right\|_\infty + \gamma \left\| (I - \gamma^2 P_\pi)^{-1} v \right\|_\infty$$

$$\leq (1+\gamma) \left\| (I - \gamma^2 P_\pi)^{-1} v \right\|_\infty$$

$$\leq 2 \left\| (I - \gamma^2 P_\pi)^{-1} v \right\|_\infty$$

Combining these calculations with the variance Bellman equation (6), we conclude that

$$\gamma \left\| (I - \gamma P_\pi)^{-1} \sqrt{v} \right\|_\infty \leq \gamma \frac{1}{\sqrt{1-\gamma}} \sqrt{2 \left\| (I - \gamma^2 P_\pi)^{-1} v \right\|_\infty} \leq \sqrt{\frac{2}{1-\gamma}} \sqrt{\left\| \mathbb{V}^\pi \left[ \sum_{t=0}^\infty \gamma^t R_t \right] \right\|_\infty}$$

as desired. $\square$

The following is a multi-step version of the variance Bellman equation, which we will later apply with $T = \mathsf{H}$ but holds for arbitrary $T$.

**Lemma 13.** *For any integer $T \geq 1$, for any deterministic stationary policy $\pi$, we have*

$$\mathbb{V}^\pi \left[ \sum_{t=0}^\infty \gamma^t R_t \right] = \mathbb{V}^\pi \left[ \sum_{t=0}^{T-1} \gamma^t R_t + \gamma^T V_\gamma^\pi(S_T) \right] + \gamma^{2T} P_\pi^T \mathbb{V}^\pi \left[ \sum_{t=0}^\infty \gamma^t R_t \right]$$

*and consequently*

$$\left\| \mathbb{V}^{\pi} \left[ \sum_{t=0}^{\infty} \gamma^t R_t \right] \right\|_{\infty} \leq \frac{\left\| \mathbb{V}^{\pi} \left[ \sum_{t=0}^{T-1} \gamma^t R_t + \gamma^T V_{\gamma}^{\pi}(S_T) \right] \right\|_{\infty}}{1 - \gamma^{2T}}.$$

*Proof.* Fix a state $s_0 \in \mathcal{S}$. Letting $\mathcal{F}_T$ be the $\sigma$-algebra generated by $(S_1, \ldots, S_T)$, we calculate that

$$
\begin{aligned}
\mathbb{V}_{s_0}^{\pi} \left[ \sum_{t=0}^{\infty} \gamma^t R_t \right] &= \mathbb{E}_{s_0}^{\pi} \left( \sum_{t=0}^{\infty} \gamma^t R_t - V_{\gamma}^{\pi}(s_0) \right)^2 \\
&= \mathbb{E}_{s_0}^{\pi} \left( \sum_{t=0}^{T-1} \gamma^t R_t + \gamma^T V_{\gamma}^{\pi}(S_T) - V_{\gamma}^{\pi}(s_0) + \sum_{t=T}^{\infty} \gamma^t R_t - \gamma^T V_{\gamma}^{\pi}(S_T) \right)^2 \\
&= \mathbb{E}_{s_0}^{\pi} \left[ \mathbb{E}_{s_0}^{\pi} \left[ \left( \underbrace{\sum_{t=0}^{T-1} \gamma^t R_t + \gamma^T V_{\gamma}^{\pi}(S_T) - V_{\gamma}^{\pi}(s_0)}_{A} + \underbrace{\sum_{t=T}^{\infty} \gamma^t R_t - \gamma^T V_{\gamma}^{\pi}(S_T)}_{B} \right)^2 \Bigg| \mathcal{F}_T \right] \right]
\end{aligned}
$$

Using the above shorthands and opening the square, we obtain

$$
\begin{aligned}
\mathbb{V}_{s_0}^{\pi} \left[ \sum_{t=0}^{\infty} \gamma^t R_t \right] &= \mathbb{E}_{s_0}^{\pi} \left[ \mathbb{E}_{s_0}^{\pi} \left[ A^2 + B^2 + 2AB \big| \mathcal{F}_T \right] \right] \\
&= \mathbb{E}_{s_0}^{\pi} \left[ A^2 + \mathbb{E}_{s_0}^{\pi} \left[ B^2 \big| \mathcal{F}_T \right] + 2A \mathbb{E}_{s_0}^{\pi} \left[ B | \mathcal{F}_T \right] \right] \\
&= \mathbb{E}_{s_0}^{\pi} \left[ A^2 + \mathbb{E}_{S_T}^{\pi} \left[ B^2 \right] \right] \\
&= \mathbb{E}_{s_0}^{\pi} \left[ \left( \sum_{t=0}^{T-1} \gamma^t R_t + \gamma^T V_{\gamma}^{\pi}(S_T) - V_{\gamma}^{\pi}(s_0) \right)^2 + \mathbb{E}_{S_T}^{\pi} \left[ \left( \sum_{t=T}^{\infty} \gamma^t R_t - \gamma^T V_{\gamma}^{\pi}(S_T) \right)^2 \right] \right] \\
&= \mathbb{E}_{s_0}^{\pi} \left[ \left( \sum_{t=0}^{T-1} \gamma^t R_t + \gamma^T V_{\gamma}^{\pi}(S_T) - V_{\gamma}^{\pi}(s_0) \right)^2 + \gamma^{2T} \mathbb{E}_{S_T}^{\pi} \left[ \left( \sum_{t=0}^{\infty} \gamma^t R_t - V_{\gamma}^{\pi}(S_T) \right)^2 \right] \right] \\
&= \mathbb{V}_{s_0}^{\pi} \left[ \sum_{t=0}^{T-1} \gamma^t R_t + \gamma^T V_{\gamma}^{\pi}(S_T) \right] + \gamma^{2T} e_{s_0}^{\top} P_{\pi}^T \mathbb{V}^{\pi} \left[ \sum_{t=0}^{\infty} \gamma^t R_t \right],
\end{aligned}
$$

where we used the tower property, the Markov property, and the fact that $\mathbb{E}_{s_0}^{\pi}[B|\mathcal{F}_T] = 0$ (which is immediate from the definition of $V_{\gamma}^{\pi}$). Since $e_{s_0}^{\top} P_{\pi}^T$ is a probability distribution, it follows from Holder's inequality that $\left| e_{s_0}^{\top} P_{\pi}^T \mathbb{V}^{\pi} \left[ \sum_{t=0}^{\infty} \gamma^t R_t \right] \right| \leq \left\| \mathbb{V}^{\pi} \left[ \sum_{t=0}^{\infty} \gamma^t R_t \right] \right\|_{\infty}$. Therefore, it holds that

$$\left\| \mathbb{V}_{s_0}^{\pi} \left[ \sum_{t=0}^{\infty} \gamma^t R_t \right] \right\|_{\infty} \leq \left\| \mathbb{V}^{\pi} \left[ \sum_{t=0}^{T-1} \gamma^t R_t + \gamma^T V_{\gamma}^{\pi}(S_T) \right] \right\|_{\infty} + \gamma^{2T} \left\| \mathbb{V}_{s_0}^{\pi} \left[ \sum_{t=0}^{\infty} \gamma^t R_t \right] \right\|_{\infty}$$

and we can obtain the desired conclusion after rearranging terms. $\qquad \square$

We also need the following elemetary inequality.

**Lemma 14.** *If $\gamma \geq 1 - \frac{1}{T}$ for some integer $T \geq 1$, then*

$$\frac{1 - \gamma^{2T}}{1 - \gamma} \geq \left( 1 - \frac{1}{e^2} \right) T \geq \frac{4}{5} T.$$

*Proof.* Fixing $T \geq 1$, we have

$$\frac{1 - \gamma^{2T}}{1 - \gamma} = 1 + \gamma + \gamma^2 + \cdots + \gamma^{2T-1}$$

which is increasing in $\gamma$, so $\inf_{\gamma \geq 1 - \frac{1}{T}} \frac{1 - \gamma^{2T}}{1 - \gamma}$ is attained at $\gamma = 1 - \frac{1}{T}$. Now allowing $T \geq 1$ to be arbitrary, note $\frac{1 - \left(1 - \frac{1}{T}\right)^{2T}}{1 - \left(1 - \frac{1}{T}\right)} = T \left(1 - \left(1 - \frac{1}{T}\right)^{2T}\right)$ so it suffices to show that $1 - \left(1 - \frac{1}{T}\right)^{2T} \geq 1 - e^2$ for all $T \geq 1$. By computing the derivative, one finds that $1 - \left(1 - \frac{1}{T}\right)^{2T}$ is monotonically decreasing, so

$$1 - \left(1 - \frac{1}{T}\right)^{2T} \geq \lim_{T \to \infty} 1 - \left(1 - \frac{1}{T}\right)^{2T} = 1 - \frac{1}{e^2}.$$

$\square$

We can now provide a bound on the variance of the total discounted rewards under $\pi_\gamma^\star$.

**Lemma 15.** *Letting $\pi_\gamma^\star$ be the optimal policy for the weakly communicating discounted MDP $(P, r, \gamma)$, if $\gamma \geq 1 - \frac{1}{H}$, we have*

$$\left\| \mathbb{V}^{\pi_\gamma^\star} \left[ \sum_{t=0}^{\infty} \gamma^t R_t \right] \right\|_{\infty} \leq 5 \frac{H}{1 - \gamma}.$$

*Proof.* By using the multi-step variance Bellman equation in Lemma 13, it suffices to bound the quantity $\left\| \mathbb{V}^{\pi_\gamma^\star} \left[ \sum_{t=0}^{H-1} \gamma^t R_t + \gamma^H V_\gamma^{\pi_\gamma^\star}(S_H) \right] \right\|_{\infty}$.

Fixing a state $s_0 \in \mathcal{S}$,

$$\mathbb{V}_{s_0}^{\pi_\gamma^\star} \left[ \sum_{t=0}^{H-1} \gamma^t R_t + \gamma^H V_\gamma^{\pi_\gamma^\star}(S_H) \right] = \mathbb{V}_{s_0}^{\pi_\gamma^\star} \left[ \sum_{t=0}^{H-1} \gamma^t R_t + \gamma^H \left( V_\gamma^{\pi_\gamma^\star}(S_H) - \frac{1}{1 - \gamma} \rho^\star \right) \right]$$

$$\leq \mathbb{E}_{s_0}^{\pi_\gamma^\star} \left| \sum_{t=0}^{H-1} \gamma^t R_t + \gamma^H \left( V_\gamma^{\pi_\gamma^\star}(S_H) - \frac{1}{1 - \gamma} \rho^\star \right) \right|^2$$

$$\leq 2 \mathbb{E}_{s_0}^{\pi_\gamma^\star} \left| \sum_{t=0}^{H-1} \gamma^t R_t \right|^2 + 2 \mathbb{E}_{s_0}^{\pi_\gamma^\star} \left| \gamma^H \left( V_\gamma^{\pi_\gamma^\star}(S_H) - \frac{1}{1 - \gamma} \rho^\star \right) \right|^2$$

$$\leq 2H^2 + 2 \sup_s \left( V_\gamma^{\pi_\gamma^\star}(s) - \frac{1}{1 - \gamma} \rho^\star \right)^2$$

$$\leq 4H^2$$

where in the final inequality we used Lemma 11. Taking the maximum over all states $s$ and combining with Lemma 13 we obtain

$$\left\| \mathbb{V}^{\pi_\gamma^\star} \left[ \sum_{t=0}^{\infty} \gamma^t R_t \right] \right\|_{\infty} \leq \frac{4H^2}{1 - \gamma^{2H}}.$$

Combining this bound with the elementary inequality in Lemma 14, which can be rearranged to show that $\frac{1}{1 - \gamma^{2H}} \leq \frac{5}{4} \frac{1}{(1 - \gamma)H}$, we complete the proof. $\square$

We also need to control the variance under $\widehat{\pi}_{\gamma, \mathrm{p}}^\star$, which requires additional steps. This is done in the following lemma.

**Lemma 16.** *We have*

$$\left\| \mathbb{V}^{\widehat{\pi}_{\gamma, \mathrm{p}}^\star} \left[ \sum_{t=0}^{\infty} \gamma^t \widetilde{R}_t \right] \right\|_{\infty} \leq 15 \frac{H^2 + \left\| V_\gamma^{\widehat{\pi}_{\gamma, \mathrm{p}}^\star} - \widehat{V}_{\gamma, \mathrm{p}}^{\widehat{\pi}_{\gamma, \mathrm{p}}^\star} \right\|_{\infty}^2 + \left\| V_\gamma^{\pi_\gamma^\star} - \widehat{V}_{\gamma, \mathrm{p}}^{\pi_\gamma^\star} \right\|_{\infty}^2}{H(1 - \gamma)}.$$

*Proof.* In light of the multi-step variance Bellman equation in Lemma 13, it suffices to give a bound on $\left\| \mathbb{V}^{\widehat{\pi}^\star_{\gamma,\mathrm{P}}} \left[ \sum_{t=0}^{\mathsf{H}-1} \gamma^t \widetilde{R}_t + \gamma^{\mathsf{H}} V_{\gamma,\mathrm{P}}^{\widehat{\pi}^\star_{\gamma,\mathrm{P}}}(S_{\mathsf{H}}) \right] \right\|_\infty$. We have for any state $s_0$ that

$$
\mathbb{V}^{\widehat{\pi}^\star_{\gamma,\mathrm{P}}}_{s_0} \left[ \sum_{t=0}^{\mathsf{H}-1} \gamma^t \widetilde{R}_t + \gamma^{\mathsf{H}} V_{\gamma,\mathrm{P}}^{\widehat{\pi}^\star_{\gamma,\mathrm{P}}}(S_{\mathsf{H}}) \right]
$$

$$
= \mathbb{V}^{\widehat{\pi}^\star_{\gamma,\mathrm{P}}}_{s_0} \left[ \sum_{t=0}^{\mathsf{H}-1} \gamma^t \widetilde{R}_t + \gamma^{\mathsf{H}} V_{\gamma,\mathrm{P}}^{\widehat{\pi}^\star_{\gamma,\mathrm{P}}}(S_{\mathsf{H}}) - \gamma^{\mathsf{H}} \frac{1}{1-\gamma} \rho^\star \right]
$$

$$
\leq \mathbb{E}^{\widehat{\pi}^\star_{\gamma,\mathrm{P}}}_{s_0} \left( \sum_{t=0}^{\mathsf{H}-1} \gamma^t \widetilde{R}_t + \gamma^{\mathsf{H}} V_{\gamma,\mathrm{P}}^{\widehat{\pi}^\star_{\gamma,\mathrm{P}}}(S_{\mathsf{H}}) - \gamma^{\mathsf{H}} \frac{1}{1-\gamma} \rho^\star \right)^2
$$

$$
= \mathbb{E}^{\widehat{\pi}^\star_{\gamma,\mathrm{P}}}_{s_0} \left( \sum_{t=0}^{\mathsf{H}-1} \gamma^t \widetilde{R}_t + \gamma^{\mathsf{H}} \left( V_{\gamma,\mathrm{P}}^{\widehat{\pi}^\star_{\gamma,\mathrm{P}}}(S_{\mathsf{H}}) - V_\gamma^{\pi^\star_\gamma}(S_{\mathsf{H}}) \right) + \gamma^{\mathsf{H}} \left( V_\gamma^{\pi^\star_\gamma}(S_{\mathsf{H}}) - \frac{1}{1-\gamma} \rho^\star \right) \right)^2
$$

$$
\leq 3 \mathbb{E}^{\widehat{\pi}^\star_{\gamma,\mathrm{P}}}_{s_0} \left( \sum_{t=0}^{\mathsf{H}-1} \gamma^t \widetilde{R}_t \right)^2 + 3\gamma^{2\mathsf{H}} \mathbb{E}^{\widehat{\pi}^\star_{\gamma,\mathrm{P}}}_{s_0} \left( V_{\gamma,\mathrm{P}}^{\widehat{\pi}^\star_{\gamma,\mathrm{P}}}(S_{\mathsf{H}}) - V_\gamma^{\pi^\star_\gamma}(S_{\mathsf{H}}) \right)^2
$$

$$
\quad + 3\gamma^{2\mathsf{H}} \mathbb{E}^{\widehat{\pi}^\star_{\gamma,\mathrm{P}}}_{s_0} \left( V_\gamma^{\pi^\star_\gamma}(S_{\mathsf{H}}) - \frac{1}{1-\gamma} \rho^\star \right)^2
$$

$$
\leq 3 \mathbb{E}^{\widehat{\pi}^\star_{\gamma,\mathrm{P}}}_{s_0} \left( \sum_{t=0}^{\mathsf{H}-1} \gamma^t \widetilde{R}_t \right)^2 + 6\gamma^{2\mathsf{H}} \mathbb{E}^{\widehat{\pi}^\star_{\gamma,\mathrm{P}}}_{s_0} \left( V_\gamma^{\widehat{\pi}^\star_{\gamma,\mathrm{P}}}(S_{\mathsf{H}}) - V_\gamma^{\pi^\star_\gamma}(S_{\mathsf{H}}) \right)^2 + 6\gamma^{2\mathsf{H}} \left\| V_{\gamma,\mathrm{P}}^{\widehat{\pi}^\star_{\gamma,\mathrm{P}}} - V_\gamma^{\widehat{\pi}^\star_{\gamma,\mathrm{P}}} \right\|_\infty^2
$$

$$
\quad + 3\gamma^{2\mathsf{H}} \mathbb{E}^{\widehat{\pi}^\star_{\gamma,\mathrm{P}}}_{s_0} \left( V_\gamma^{\pi^\star_\gamma}(S_{\mathsf{H}}) - \frac{1}{1-\gamma} \rho^\star \right)^2, \tag{7}
$$

where we have used triangle inequality and the inequalities $(a+b)^2 \leq 2a^2 + 2b^2$ and $(a+b+c)^2 \leq 3a^2 + 3b^2 + 3c^2$. Now we bound each term of (7). First, we have

$$
3 \mathbb{E}^{\widehat{\pi}^\star_{\gamma,\mathrm{P}}}_{s_0} \left( \sum_{t=0}^{\mathsf{H}-1} \gamma^t \widetilde{R}_t \right)^2 \leq 3 \left( \mathsf{H} \|\widetilde{r}\|_\infty \right)^2 \leq 3\mathsf{H}^2 (\|r\|_\infty + \xi)^2 \leq 6\mathsf{H}^2 \left( 1 + \left( \frac{(1-\gamma)\varepsilon}{6} \right)^2 \right) \leq 6\mathsf{H}^2 \left( \frac{7}{6} \right)^2,
$$

where we had $\frac{(1-\gamma)\varepsilon}{6} \leq \frac{\varepsilon}{6\mathsf{H}} \leq \frac{1}{6}$ because $\frac{1}{1-\gamma} \geq \mathsf{H}$ and $\varepsilon \leq \mathsf{H}$. Clearly it holds that

$$
6\gamma^{2\mathsf{H}} \mathbb{E}^{\widehat{\pi}^\star_{\gamma,\mathrm{P}}}_{s_0} \left( V_\gamma^{\widehat{\pi}^\star_{\gamma,\mathrm{P}}}(S_{\mathsf{H}}) - V_\gamma^{\pi^\star_\gamma}(S_{\mathsf{H}}) \right)^2 \leq 6 \left\| V_\gamma^{\widehat{\pi}^\star_{\gamma,\mathrm{P}}} - V_\gamma^{\pi^\star_\gamma} \right\|_\infty^2.
$$

By an argument identical to those used in the proof of the error bounds in Lemma 10, we get

$$
\left\| V_{\gamma,\mathrm{P}}^{\widehat{\pi}^\star_{\gamma,\mathrm{P}}} - V_\gamma^{\widehat{\pi}^\star_{\gamma,\mathrm{P}}} \right\|_\infty \leq \frac{1}{1-\gamma} \xi = \frac{\varepsilon}{6},
$$

so $6\gamma^{2\mathsf{H}} \left\| V_{\gamma,\mathrm{P}}^{\widehat{\pi}^\star_{\gamma,\mathrm{P}}} - V_\gamma^{\widehat{\pi}^\star_{\gamma,\mathrm{P}}} \right\|_\infty^2 \leq \frac{\varepsilon^2}{6} \leq \frac{\mathsf{H}^2}{6}$ since $\varepsilon \leq \mathsf{H}$. Finally, using Lemma 11, we obtain

$$
3\gamma^{2\mathsf{H}} \mathbb{E}^{\widehat{\pi}^\star_{\gamma,\mathrm{P}}}_{s_0} \left( V_\gamma^{\pi^\star_\gamma}(S_{\mathsf{H}}) - \frac{1}{1-\gamma} \rho^\star \right)^2 \leq 3 \sup_s \left| V_\gamma^{\pi^\star_\gamma}(S_{\mathsf{H}}) - \frac{1}{1-\gamma} \rho^\star \right|^2 \leq 3\mathsf{H}^2.
$$

Using all these bounds in (7), we have

$$
\mathbb{V}_{s_0}^{\widehat{\pi}_{\gamma}^{\star},\mathrm{P}} \left[ \sum_{t=0}^{\mathsf{H}-1} \gamma^t \widetilde{R}_t + \gamma^{\mathsf{H}} V_{\gamma,\mathrm{P}}^{\widehat{\pi}_{\gamma}^{\star},\mathrm{P}}(S_{\mathsf{H}}) \right]
$$

$$
\leq 3\mathbb{E}_{s_0}^{\widehat{\pi}_{\gamma}^{\star},\mathrm{P}} \left( \sum_{t=0}^{\mathsf{H}-1} \gamma^t \widetilde{R}_t \right)^2 + 6\gamma^{2\mathsf{H}} \mathbb{E}_{s_0}^{\widehat{\pi}_{\gamma}^{\star},\mathrm{P}} \left( V_{\gamma}^{\widehat{\pi}_{\gamma}^{\star},\mathrm{P}}(S_{\mathsf{H}}) - V_{\gamma}^{\pi_{\gamma}^{\star}}(S_{\mathsf{H}}) \right)^2 + 6\gamma^{2\mathsf{H}} \left\| V_{\gamma,\mathrm{P}}^{\widehat{\pi}_{\gamma}^{\star},\mathrm{P}} - V_{\gamma}^{\widehat{\pi}_{\gamma}^{\star},\mathrm{P}} \right\|_{\infty}^2
$$

$$
+ 3\gamma^{2\mathsf{H}} \mathbb{E}_{s_0}^{\widehat{\pi}_{\gamma}^{\star},\mathrm{P}} \left( V_{\gamma}^{\pi_{\gamma}^{\star}}(S_{\mathsf{H}}) - \frac{1}{1-\gamma}\rho^{\star} \right)^2
$$

$$
\leq \left( \frac{49}{6} + \frac{1}{6} + 3 \right) \mathsf{H}^2 + 6 \left\| V_{\gamma}^{\widehat{\pi}_{\gamma}^{\star},\mathrm{P}} - V_{\gamma}^{\pi_{\gamma}^{\star}} \right\|_{\infty}^2
$$

$$
\leq 12\mathsf{H}^2 + 6 \left\| V_{\gamma}^{\widehat{\pi}_{\gamma}^{\star},\mathrm{P}} - V_{\gamma}^{\pi_{\gamma}^{\star}} \right\|_{\infty}^2 . \tag{8}
$$

Finally, we use the elementwise inequality

$$
V_{\gamma}^{\pi_{\gamma}^{\star}} \geq V_{\gamma}^{\widehat{\pi}_{\gamma}^{\star},\mathrm{P}}
$$

$$
\geq \widehat{V}_{\gamma,\mathrm{P}}^{\widehat{\pi}_{\gamma}^{\star},\mathrm{P}} - \left\| \widehat{V}_{\gamma,\mathrm{P}}^{\widehat{\pi}_{\gamma}^{\star},\mathrm{P}} - V_{\gamma}^{\widehat{\pi}_{\gamma}^{\star},\mathrm{P}} \right\|_{\infty} \mathbf{1}
$$

$$
\geq \widehat{V}_{\gamma,\mathrm{P}}^{\pi_{\gamma}^{\star}} - \left\| \widehat{V}_{\gamma,\mathrm{P}}^{\widehat{\pi}_{\gamma}^{\star},\mathrm{P}} - V_{\gamma}^{\widehat{\pi}_{\gamma}^{\star},\mathrm{P}} \right\|_{\infty} \mathbf{1}
$$

$$
\geq V_{\gamma}^{\pi_{\gamma}^{\star}} - \left\| \widehat{V}_{\gamma,\mathrm{P}}^{\widehat{\pi}_{\gamma}^{\star},\mathrm{P}} - V_{\gamma}^{\widehat{\pi}_{\gamma}^{\star},\mathrm{P}} \right\|_{\infty} \mathbf{1} - \left\| \widehat{V}_{\gamma,\mathrm{P}}^{\pi_{\gamma}^{\star}} - V_{\gamma}^{\pi_{\gamma}^{\star}} \right\|_{\infty} \mathbf{1},
$$

from which it follows that $\left\| V_{\gamma}^{\widehat{\pi}_{\gamma}^{\star},\mathrm{P}} - V_{\gamma}^{\pi_{\gamma}^{\star}} \right\|_{\infty} \leq \left\| \widehat{V}_{\gamma,\mathrm{P}}^{\widehat{\pi}_{\gamma}^{\star},\mathrm{P}} - V_{\gamma}^{\widehat{\pi}_{\gamma}^{\star},\mathrm{P}} \right\|_{\infty} + \left\| \widehat{V}_{\gamma,\mathrm{P}}^{\pi_{\gamma}^{\star}} - V_{\gamma}^{\pi_{\gamma}^{\star}} \right\|_{\infty}$. Combining this with (8), we conclude

$$
\mathbb{V}_{s_0}^{\widehat{\pi}_{\gamma}^{\star},\mathrm{P}} \left[ \sum_{t=0}^{\mathsf{H}-1} \gamma^t \widetilde{R}_t + \gamma^{\mathsf{H}} V_{\gamma,\mathrm{P}}^{\widehat{\pi}_{\gamma}^{\star},\mathrm{P}}(S_{\mathsf{H}}) \right] \leq 12\mathsf{H}^2 + 12 \left\| \widehat{V}_{\gamma,\mathrm{P}}^{\widehat{\pi}_{\gamma}^{\star},\mathrm{P}} - V_{\gamma}^{\widehat{\pi}_{\gamma}^{\star},\mathrm{P}} \right\|_{\infty}^2 + 12 \left\| \widehat{V}_{\gamma,\mathrm{P}}^{\pi_{\gamma}^{\star}} - V_{\gamma}^{\pi_{\gamma}^{\star}} \right\|_{\infty}^2 . \tag{9}
$$

Now combining with Lemma 13 and then using Lemma 14, we have

$$
\left\| \mathbb{V}^{\widehat{\pi}_{\gamma}^{\star},\mathrm{P}} \left[ \sum_{t=0}^{\infty} \gamma^t \widetilde{R}_t \right] \right\|_{\infty} \leq \frac{\left\| \mathbb{V}^{\widehat{\pi}_{\gamma}^{\star},\mathrm{P}} \left[ \sum_{t=0}^{\mathsf{H}-1} \gamma^t \widetilde{R}_t + \gamma^{\mathsf{H}} V_{\gamma}^{\widehat{\pi}_{\gamma}^{\star},\mathrm{P}}(S_{\mathsf{H}}) \right] \right\|_{\infty}}{1 - \gamma^{2\mathsf{H}}}
$$

$$
\leq 12 \frac{\mathsf{H}^2 + \left\| V_{\gamma}^{\widehat{\pi}_{\gamma}^{\star},\mathrm{P}} - \widehat{V}_{\gamma,\mathrm{P}}^{\widehat{\pi}_{\gamma}^{\star},\mathrm{P}} \right\|_{\infty}^2 + \left\| V_{\gamma}^{\pi_{\gamma}^{\star}} - \widehat{V}_{\gamma,\mathrm{P}}^{\pi_{\gamma}^{\star}} \right\|_{\infty}^2}{1 - \gamma^{2\mathsf{H}}}
$$

$$
\leq 12\frac{5}{4} \frac{\mathsf{H}^2 + \left\| V_{\gamma}^{\widehat{\pi}_{\gamma}^{\star},\mathrm{P}} - \widehat{V}_{\gamma,\mathrm{P}}^{\widehat{\pi}_{\gamma}^{\star},\mathrm{P}} \right\|_{\infty}^2 + \left\| V_{\gamma}^{\pi_{\gamma}^{\star}} - \widehat{V}_{\gamma,\mathrm{P}}^{\pi_{\gamma}^{\star}} \right\|_{\infty}^2}{\mathsf{H}(1 - \gamma)}
$$

$$
= 15 \frac{\mathsf{H}^2 + \left\| V_{\gamma}^{\widehat{\pi}_{\gamma}^{\star},\mathrm{P}} - \widehat{V}_{\gamma,\mathrm{P}}^{\widehat{\pi}_{\gamma}^{\star},\mathrm{P}} \right\|_{\infty}^2 + \left\| V_{\gamma}^{\pi_{\gamma}^{\star}} - \widehat{V}_{\gamma,\mathrm{P}}^{\pi_{\gamma}^{\star}} \right\|_{\infty}^2}{\mathsf{H}(1 - \gamma)}
$$

as desired. $\qquad\square$

## A.2 Proofs of Theorems 1 and 2

With the above lemmas we can complete the proof of Theorem 1 on discounted MDPs.

*Proof of Theorem 1.* Our approach will be to utilize our variance bounds within the error bounds from Lemma 10. We will find a value for $n$ which guarantees that $\left\| \widehat{V}_{\gamma,\mathrm{P}}^{\pi_{\gamma}^{\star}} - V_{\gamma}^{\pi_{\gamma}^{\star}} \right\|_{\infty}$ and $\left\| \widehat{V}_{\gamma,\mathrm{P}}^{\widehat{\pi}_{\gamma}^{\star},\mathrm{P}} - V_{\gamma}^{\widehat{\pi}_{\gamma}^{\star},\mathrm{P}} \right\|_{\infty}$ are both $\leq \varepsilon/2$, which guarantees that $\left\| V_{\gamma}^{\widehat{\pi}_{\gamma}^{\star},\mathrm{P}} - V_{\gamma}^{\pi_{\gamma}^{\star}} \right\|_{\infty} \leq \varepsilon$.

First we note that the conclusions of Lemma 10 require $n \geq \frac{c_2}{1-\gamma} \log\left(\frac{SA}{(1-\gamma)\delta\varepsilon}\right)$ so we assume $n$ is large enough that this holds.

Now we bound $\left\|\widehat{V}_{\gamma,\mathrm{P}}^{\pi_\gamma^\star} - V_\gamma^{\pi_\gamma^\star}\right\|_\infty$. Starting with inequality (1) from Lemma 10 and then applying our variance bounds through Lemma 12 and then Lemma 15, we have

$$\left\|\widehat{V}_{\gamma,\mathrm{P}}^{\pi_\gamma^\star} - V_\gamma^{\pi_\gamma^\star}\right\|_\infty$$

$$\leq \gamma\sqrt{\frac{c_1 \log\left(\frac{SA}{(1-\gamma)\delta\varepsilon}\right)}{n}} \left\|(I - \gamma P_{\pi_\gamma^\star})^{-1}\sqrt{\mathbb{V}_{P_{\pi_\gamma^\star}}\left[V_\gamma^{\pi_\gamma^\star}\right]}\right\|_\infty + c_1\gamma\frac{\log\left(\frac{SA}{(1-\gamma)\delta\varepsilon}\right)}{(1-\gamma)n}\left\|V_\gamma^{\pi_\gamma^\star}\right\|_\infty + \frac{\varepsilon}{6}$$

$$\leq \sqrt{\frac{c_1 \log\left(\frac{SA}{(1-\gamma)\delta\varepsilon}\right)}{n}}\sqrt{\frac{2}{1-\gamma}}\sqrt{\left\|\mathbb{V}^{\pi_\gamma^\star}\left[\sum_{t=0}^{\infty}\gamma^t R_t\right]\right\|_\infty} + c_1\gamma\frac{\log\left(\frac{SA}{(1-\gamma)\delta\varepsilon}\right)}{(1-\gamma)n}\left\|V_\gamma^{\pi_\gamma^\star}\right\|_\infty + \frac{\varepsilon}{6}$$

$$\leq \sqrt{\frac{c_1 \log\left(\frac{SA}{(1-\gamma)\delta\varepsilon}\right)}{n}}\sqrt{\frac{2}{1-\gamma}}\sqrt{5\frac{\mathsf{H}}{1-\gamma}} + c_1\gamma\frac{\log\left(\frac{SA}{(1-\gamma)\delta\varepsilon}\right)}{(1-\gamma)n}\left\|V_\gamma^{\pi_\gamma^\star}\right\|_\infty + \frac{\varepsilon}{6}$$

$$\leq \sqrt{\frac{c_1 \log\left(\frac{SA}{(1-\gamma)\delta\varepsilon}\right)}{n}}\sqrt{10\frac{\mathsf{H}}{(1-\gamma)^2}} + c_1\frac{\log\left(\frac{SA}{(1-\gamma)\delta\varepsilon}\right)}{(1-\gamma)^2 n} + \frac{\varepsilon}{6}$$

where in the last inequality we used the facts that $\left\|V_\gamma^{\pi_\gamma^\star}\right\|_\infty \leq \frac{1}{1-\gamma}$ and $\gamma \leq 1$. Now if we assume $n \geq 360 c_1 \frac{\mathsf{H}}{(1-\gamma)^2\varepsilon^2}\log\left(\frac{SA}{(1-\gamma)\delta\varepsilon}\right)$, we have

$$\left\|\widehat{V}_{\gamma,\mathrm{P}}^{\pi_\gamma^\star} - V_\gamma^{\pi_\gamma^\star}\right\|_\infty \leq \sqrt{\frac{c_1 \log\left(\frac{SA}{(1-\gamma)\delta\varepsilon}\right)}{n}}\sqrt{10\frac{\mathsf{H}}{(1-\gamma)^2}} + c_1\frac{\log\left(\frac{SA}{(1-\gamma)\delta\varepsilon}\right)}{(1-\gamma)^2 n} + \frac{\varepsilon}{6}$$

$$\leq \frac{1}{6}\sqrt{\varepsilon^2} + \frac{1}{6}\frac{\varepsilon^2}{\mathsf{H}} + \frac{\varepsilon}{6}$$

$$\leq \varepsilon/2$$

due to the fact that $\varepsilon \leq \mathsf{H}$.

Next, to bound $\left\|\widehat{V}_{\gamma,\mathrm{P}}^{\widehat{\pi}_{\gamma,\mathrm{P}}^\star} - V_\gamma^{\widehat{\pi}_{\gamma,\mathrm{P}}^\star}\right\|_\infty$, starting from inequality (2) in Lemma 10 and then analogously applying Lemma 12 and then Lemma 16, we obtain

$$\left\|\widehat{V}_{\gamma,\mathrm{P}}^{\widehat{\pi}_{\gamma,\mathrm{P}}^\star} - V_\gamma^{\widehat{\pi}_{\gamma,\mathrm{P}}^\star}\right\|_\infty$$

$$\leq \gamma\sqrt{\frac{c_1 \log\left(\frac{SA}{(1-\gamma)\delta\varepsilon}\right)}{n}}\left\|(I - \gamma P_{\widehat{\pi}_{\gamma,\mathrm{P}}^\star})^{-1}\sqrt{\mathbb{V}_{P_{\widehat{\pi}_{\gamma,\mathrm{P}}^\star}}\left[V_{\gamma,\mathrm{P}}^{\widehat{\pi}_{\gamma,\mathrm{P}}^\star}\right]}\right\|_\infty + c_1\gamma\frac{\log\left(\frac{SA}{(1-\gamma)\delta\varepsilon}\right)}{(1-\gamma)n}\left\|V_{\gamma,\mathrm{P}}^{\widehat{\pi}_{\gamma,\mathrm{P}}^\star}\right\|_\infty + \frac{\varepsilon}{6}$$

$$\leq \sqrt{\frac{c_1 \log\left(\frac{SA}{(1-\gamma)\delta\varepsilon}\right)}{n}}\sqrt{\frac{2}{1-\gamma}}\sqrt{\left\|\mathbb{V}^{\widehat{\pi}_{\gamma,\mathrm{P}}^\star}\left[\sum_{t=0}^{\infty}\gamma^t \widetilde{R}_t\right]\right\|_\infty} + c_1\gamma\frac{\log\left(\frac{SA}{(1-\gamma)\delta\varepsilon}\right)}{(1-\gamma)n}\left\|V_{\gamma,\mathrm{P}}^{\widehat{\pi}_{\gamma,\mathrm{P}}^\star}\right\|_\infty + \frac{\varepsilon}{6}$$

$$\leq \sqrt{\frac{c_1 \log\left(\frac{SA}{(1-\gamma)\delta\varepsilon}\right)}{n}}\sqrt{\frac{2}{1-\gamma}}\sqrt{15\frac{\mathsf{H}^2 + \left\|V_\gamma^{\widehat{\pi}_{\gamma,\mathrm{P}}^\star} - \widehat{V}_{\gamma,\mathrm{P}}^{\widehat{\pi}_{\gamma,\mathrm{P}}^\star}\right\|_\infty^2 + \left\|V_\gamma^{\pi_\gamma^\star} - \widehat{V}_{\gamma,\mathrm{P}}^{\pi_\gamma^\star}\right\|_\infty^2}{\mathsf{H}(1-\gamma)}}$$

$$+ c_1\gamma\frac{\log\left(\frac{SA}{(1-\gamma)\delta\varepsilon}\right)}{(1-\gamma)n}\left\|V_{\gamma,\mathrm{P}}^{\widehat{\pi}_{\gamma,\mathrm{P}}^\star}\right\|_\infty + \frac{\varepsilon}{6}.$$

Combining with the fact from above that $\left\|\widehat{V}_{\gamma,\mathrm{p}}^{\pi_\gamma^\star} - V_\gamma^{\pi_\gamma^\star}\right\|_\infty \leq \frac{\mathsf{H}}{2}$, as well as the facts that $\left\|V_{\gamma,\mathrm{p}}^{\widehat{\pi}_{\gamma,\mathrm{p}}^\star}\right\|_\infty \leq \frac{1}{1-\gamma}$, $\gamma \leq 1$, and $\sqrt{a+b} \leq \sqrt{a} + \sqrt{b}$, we have

$$
\left\|\widehat{V}_{\gamma,\mathrm{p}}^{\widehat{\pi}_{\gamma,\mathrm{p}}^\star} - V_\gamma^{\widehat{\pi}_{\gamma,\mathrm{p}}^\star}\right\|_\infty \leq \sqrt{\frac{c_1 \log\left(\frac{SA}{(1-\gamma)\delta\varepsilon}\right)}{n}} \sqrt{\frac{2}{1-\gamma}} \sqrt{15 \frac{\frac{5}{4}\mathsf{H}^2 + \left\|V_\gamma^{\widehat{\pi}_{\gamma,\mathrm{p}}^\star} - \widehat{V}_{\gamma,\mathrm{p}}^{\widehat{\pi}_{\gamma,\mathrm{p}}^\star}\right\|_\infty^2}{\mathsf{H}(1-\gamma)}}
$$

$$
+ c_1 \frac{\log\left(\frac{SA}{(1-\gamma)\delta\varepsilon}\right)}{(1-\gamma)^2 n} + \frac{\varepsilon}{6}
$$

$$
\leq \sqrt{\frac{c_1 \log\left(\frac{SA}{(1-\gamma)\delta\varepsilon}\right)}{n}} \sqrt{\frac{30}{\mathsf{H}(1-\gamma)^2}} \left(\sqrt{\frac{5}{4}\mathsf{H}^2} + \sqrt{\left\|V_\gamma^{\widehat{\pi}_{\gamma,\mathrm{p}}^\star} - \widehat{V}_{\gamma,\mathrm{p}}^{\widehat{\pi}_{\gamma,\mathrm{p}}^\star}\right\|_\infty^2}\right)
$$

$$
+ c_1 \frac{\log\left(\frac{SA}{(1-\gamma)\delta\varepsilon}\right)}{(1-\gamma)^2 n} + \frac{\varepsilon}{6}
$$

$$
= \sqrt{\frac{c_1 \log\left(\frac{SA}{(1-\gamma)\delta\varepsilon}\right)}{n}} \sqrt{\frac{30}{\mathsf{H}(1-\gamma)^2}} \left(\sqrt{\frac{5}{4}}\mathsf{H} + \left\|V_\gamma^{\widehat{\pi}_{\gamma,\mathrm{p}}^\star} - \widehat{V}_{\gamma,\mathrm{p}}^{\widehat{\pi}_{\gamma,\mathrm{p}}^\star}\right\|_\infty\right)
$$

$$
+ c_1 \frac{\log\left(\frac{SA}{(1-\gamma)\delta\varepsilon}\right)}{(1-\gamma)^2 n} + \frac{\varepsilon}{6}.
$$

Rearranging terms gives

$$
\left(1 - \sqrt{\frac{c_1 \log\left(\frac{SA}{(1-\gamma)\delta\varepsilon}\right)}{n}} \sqrt{\frac{30}{\mathsf{H}(1-\gamma)^2}}\right) \left\|\widehat{V}_{\gamma,\mathrm{p}}^{\widehat{\pi}_{\gamma,\mathrm{p}}^\star} - V_\gamma^{\widehat{\pi}_{\gamma,\mathrm{p}}^\star}\right\|_\infty
$$

$$
\leq \sqrt{\frac{c_1 \log\left(\frac{SA}{(1-\gamma)\delta\varepsilon}\right)}{n}} \sqrt{\frac{75\mathsf{H}/2}{(1-\gamma)^2}} + c_1 \frac{\log\left(\frac{SA}{(1-\gamma)\delta\varepsilon}\right)}{(1-\gamma)^2 n} + \frac{\varepsilon}{6}.
$$

Assuming $n \geq 120 c_1 \frac{\mathsf{H}}{(1-\gamma)^2 \varepsilon^2} \log\left(\frac{SA}{(1-\gamma)\delta\varepsilon}\right)$, we have

$$
1 - \sqrt{\frac{c_1 \log\left(\frac{SA}{(1-\gamma)\delta\varepsilon}\right)}{n}} \sqrt{\frac{30}{\mathsf{H}(1-\gamma)^2}} \geq 1 - \frac{1}{2} \sqrt{\frac{\varepsilon^2(1-\gamma)^2}{\mathsf{H}} \frac{1}{\mathsf{H}(1-\gamma)^2}} = 1 - \frac{1}{2} \frac{\varepsilon}{\mathsf{H}} \geq \frac{1}{2}
$$

since $\varepsilon \leq \mathsf{H}$. Also assuming $n \geq (75/2) \cdot 24^2 c_1 \frac{\mathsf{H}}{(1-\gamma)^2 \varepsilon^2} \log\left(\frac{SA}{(1-\gamma)\delta\varepsilon}\right)$ we have similarly to before that

$$
\sqrt{\frac{c_1 \log\left(\frac{SA}{(1-\gamma)\delta\varepsilon}\right)}{n}} \sqrt{\frac{75\mathsf{H}/2}{(1-\gamma)^2}} + c_1 \frac{\log\left(\frac{SA}{(1-\gamma)\delta\varepsilon}\right)}{(1-\gamma)^2 n} + \frac{\varepsilon}{6}
$$

$$
\leq \frac{1}{24} \sqrt{\frac{(1-\gamma)^2 \varepsilon^2}{\mathsf{H}} \frac{\mathsf{H}}{(1-\gamma)^2}} + \frac{1}{24} \frac{(1-\gamma)^2 \varepsilon^2}{\mathsf{H}} \frac{1}{(1-\gamma)^2} + \frac{\varepsilon}{6}
$$

$$
\leq \frac{\varepsilon}{24} + \frac{\varepsilon}{24} + \frac{\varepsilon}{6} = \frac{\varepsilon}{4}.
$$

Combining these two calculations, we have $\frac{1}{2}\left\|\widehat{V}_{\gamma,\mathrm{p}}^{\widehat{\pi}_{\gamma,\mathrm{p}}^\star} - V_\gamma^{\widehat{\pi}_{\gamma,\mathrm{p}}^\star}\right\|_\infty \leq \frac{\varepsilon}{4}$, so $\left\|\widehat{V}_{\gamma,\mathrm{p}}^{\widehat{\pi}_{\gamma,\mathrm{p}}^\star} - V_\gamma^{\widehat{\pi}_{\gamma,\mathrm{p}}^\star}\right\|_\infty \leq \frac{\varepsilon}{2}$ as desired.

Since we have established that $\left\|\widehat{V}_{\gamma,\mathrm{p}}^{\pi_\gamma^\star} - V_\gamma^{\pi_\gamma^\star}\right\|_\infty, \left\|\widehat{V}_{\gamma,\mathrm{p}}^{\widehat{\pi}_{\gamma,\mathrm{p}}^\star} - V_\gamma^{\widehat{\pi}_{\gamma,\mathrm{p}}^\star}\right\|_\infty \leq \frac{\varepsilon}{2}$, since also $\widehat{V}_{\gamma,\mathrm{p}}^{\widehat{\pi}_{\gamma,\mathrm{p}}^\star} \geq \widehat{V}_{\gamma,\mathrm{p}}^{\pi_\gamma^\star}$, we can conclude that

$$
V_\gamma^{\pi_\gamma^\star} - V_\gamma^{\widehat{\pi}_{\gamma,\mathrm{p}}^\star} \leq \left\|\widehat{V}_{\gamma,\mathrm{p}}^{\pi_\gamma^\star} - V_\gamma^{\pi_\gamma^\star}\right\|_\infty \mathbf{1} + \left\|\widehat{V}_{\gamma,\mathrm{p}}^{\widehat{\pi}_{\gamma,\mathrm{p}}^\star} - V_\gamma^{\widehat{\pi}_{\gamma,\mathrm{p}}^\star}\right\|_\infty \mathbf{1} \leq \varepsilon\mathbf{1},
$$

that is that $\widehat{\pi}^\star_{\gamma,\mathrm{p}}$ is $\varepsilon$-optimal for the discounted MDP $(P, r, \gamma)$.

We finally note that all our requirements on the size of $n$ can be satisfied by requiring

$$n \geq C_2 \frac{\mathsf{H}}{(1-\gamma)^2 \varepsilon^2} \log\left(\frac{SA}{(1-\gamma)\delta\varepsilon}\right)$$

$$:= \max\left\{\frac{c_2 \mathsf{H}}{(1-\gamma)^2 \varepsilon^2}, \frac{360 c_1 \mathsf{H}}{(1-\gamma)^2 \varepsilon^2}, \frac{(75/2)24^2 c_1 \mathsf{H}}{(1-\gamma)^2 \varepsilon^2}\right\} \log\left(\frac{SA}{(1-\gamma)\delta\varepsilon}\right)$$

$$\geq \max\left\{\frac{c_2}{1-\gamma}, \frac{360 c_1 \mathsf{H}}{(1-\gamma)^2 \varepsilon^2}, \frac{(75/2)24^2 c_1 \mathsf{H}}{(1-\gamma)^2 \varepsilon^2}\right\} \log\left(\frac{SA}{(1-\gamma)\delta\varepsilon}\right)$$

where we used that $\frac{\mathsf{H}}{(1-\gamma)^2\varepsilon^2} \geq \frac{\mathsf{H}^2}{(1-\gamma)\varepsilon^2} \geq \frac{1}{1-\gamma}$ (since $\frac{1}{1-\gamma} \geq \mathsf{H}$ and $\mathsf{H} \geq \varepsilon$). $\qquad\square$

We next use Theorem 1 to prove Theorem 2 on average-reward MDPs.

*Proof of Theorem 2.* Using Theorem 1 with target accuracy $\mathsf{H}$ and discount factor $\overline{\gamma} = 1 - \frac{\varepsilon}{12\mathsf{H}}$, we obtain a $\mathsf{H}$-optimal policy for the discounted MDP $(P, r, \overline{\gamma})$ with probability at least $1 - \delta$ as long as

$$n \geq C_2 \frac{\mathsf{H}}{(1-\overline{\gamma})^2 \mathsf{H}^2} \log\left(\frac{SA}{(1-\overline{\gamma})\delta\varepsilon}\right)$$

$$= 12^2 C_2 \frac{\mathsf{H}}{\mathsf{H}^2} \frac{\mathsf{H}^2}{\varepsilon^2} \log\left(\frac{12\mathsf{H}}{\varepsilon} \frac{SA}{\delta\varepsilon}\right)$$

which is satisfied when $n \geq C_1 \frac{\mathsf{H}}{\varepsilon^2} \log\left(\frac{SA\mathsf{H}}{\delta\varepsilon}\right)$ for sufficiently large $C_1$.

Applying Lemma 9 (with error parameter $\frac{\varepsilon}{12}$ since we have chosen $\overline{\gamma} = 1 - \frac{\varepsilon/12}{\mathsf{H}}$), we have that

$$\rho^\star - \rho^{\widehat{\pi}^\star} \leq \left(8 + 3\frac{\mathsf{H}}{\mathsf{H}}\right) \frac{\varepsilon}{12} \leq \varepsilon \mathbf{1}$$

as desired. $\qquad\square$

## A.3 Proof of Theorem 3

*Proof of Theorem 3.* Fix $T, n \geq 1$. First we define the instances $\mathcal{M}_0$ and $\mathcal{M}_1$, which have parameters $B$ and $\varepsilon$ which we will choose later, using Figure 2. Note that in both MDPs, all states have only one action. The only difference is in the state transition distribution at state 1: For $\mathcal{M}_0$ this is a $\mathrm{Cat}(\frac{1}{2}, \frac{1}{2})$ distribution and for $\mathcal{M}_1$ this is a $\mathrm{Cat}(\frac{1}{2} + \varepsilon, \frac{1}{2} - \varepsilon)$ distribution, where $\mathrm{Cat}(p_1, p_2)$ denotes the categorical distribution with event probabilities $p_1$ and $p_2 = 1 - p_1$.

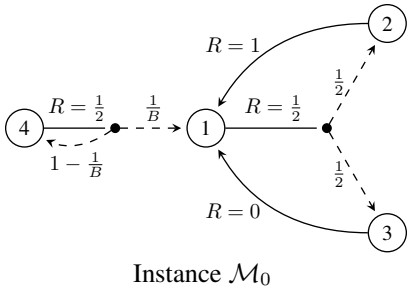

Instance $\mathcal{M}_0$

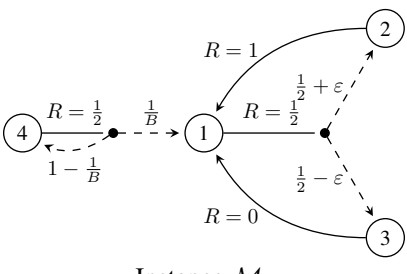

Instance $\mathcal{M}_1$

Figure 2: MDPs used in Theorem 3

Now we calculate the bias of instance $\mathcal{M}_1$. It is easy to check the stationary distribution is $\mu = [\frac{1}{2}, \frac{1}{4} + \frac{\varepsilon}{2}, \frac{1}{4} - \frac{\varepsilon}{2}, 0]$. Therefore it has optimal gain $\rho^\star = \frac{1}{2}\frac{1}{2} + \frac{1}{4} + \frac{\varepsilon}{2} = \frac{1}{2} + \frac{\varepsilon}{2}$. Now we claim that the optimal bias is

$$ h^\star = \begin{bmatrix} -\varepsilon/2 \\ \frac{1}{2} - \varepsilon/2 \\ -\frac{1}{2} - \varepsilon/2 \\ -(B+1)\frac{\varepsilon}{2} \end{bmatrix}. $$

We can check this by showing that $\mu h^\star = 0$ and that $\rho^\star \mathbf{1} + h^\star = r + P h^\star$, where $P$ is the transition matrix of the above MDP (again, note that each state has only one action, so there is only one policy, and we use this policy to induce the markov chain with transition matrix $P$). First,

$$ \mu h^\star = -\frac{\varepsilon}{4} + \frac{1}{8} + \frac{\varepsilon}{4} - \frac{\varepsilon}{8} - \frac{\varepsilon^2}{4} - \frac{1}{8} + \frac{\varepsilon}{4} - \frac{\varepsilon}{8} + \frac{\varepsilon^2}{4} = 0. $$

It is also easy to check the first three rows of the equality $\rho^\star \mathbf{1} + h^\star = r + P h^\star$. For the fourth row, we have

$$ h^\star(4) + \frac{1}{2} + \frac{\varepsilon}{2} = \frac{1}{2} + \frac{1}{B} h^\star(1) + \left(1 - \frac{1}{B}\right) h^\star(4) $$

$$ \Longleftrightarrow \frac{1}{B} h^\star(4) = \frac{-\varepsilon}{2B} - \frac{\varepsilon}{2} $$

$$ \Longleftrightarrow h^\star(4) = \frac{\varepsilon}{2}(B+1). $$

Thus $\|h^\star\|_{\mathrm{span}} = \frac{1}{2} - \varepsilon/2 - \left(-(B+1)\frac{\varepsilon}{2}\right) = \frac{1}{2}(B\varepsilon + 1)$. If we set $B = \frac{2T}{\varepsilon} - \frac{1}{2}$, we have $\|h^\star\|_{\mathrm{span}} = T$. Also note that the calculation for $h^\star$ holds for any $\varepsilon$, so the optimal bias span of $\mathcal{M}_0$ is $[0, \frac{1}{2}, -\frac{1}{2}, 0]^\top$, and thus $\mathcal{M}_0$ has optimal bias span 1.

Finally, to distinguish between the two MDPs $\mathcal{M}_0$ and $\mathcal{M}_1$, we must be able to determine the next-state distribution of state 1, that is, to distinguish between the two hypotheses $Q_1 = \mathrm{Cat}(\frac{1}{2}, \frac{1}{2})$ and $Q_2 = \mathrm{Cat}(\frac{1}{2} + \varepsilon, \frac{1}{2} - \varepsilon)$. Given $n$ i.i.d. observations from the transition distribution of state 1, this is a binary hypothesis testing problem between the product distributions $Q_1^n$ and $Q_2^n$. By Le

Cam's bound [24], the testing failure probability is lower bounded by

$$\frac{1}{2}\left(1 - \|Q_1^n - Q_2^n\|_{\text{TV}}\right) \geq \frac{1}{2}\left(1 - \sqrt{\frac{1}{2}\text{D}_{\text{KL}}(Q_1^n|Q_2^n)}\right)$$

$$= \frac{1}{2}\left(1 - \sqrt{\frac{n}{2}\text{D}_{\text{KL}}(Q_1|Q_2)}\right),$$

where $\|Q_1^n - Q_2^n\|_{\text{TV}}$ and $\text{D}_{\text{KL}}(Q_1^n|Q_2^n)$ denote the total variation distance and Kullback–Leibler (KL) divergence between $Q_1^n$ and $Q_2^n$, respectively, and the last two (in)equalities follow from Pinsker's inequality and tensorization of KL divergence. By direct calculation, we have

$$\text{D}_{\text{KL}}(Q_1|Q_2) = \frac{1}{2}\log\frac{1}{1+2\varepsilon} + \frac{1}{2}\log\frac{1}{1-2\varepsilon}$$

$$\leq \frac{1}{2}\cdot\frac{-2\varepsilon}{1+2\varepsilon} + \frac{1}{2}\cdot\frac{2\varepsilon}{1-2\varepsilon} \qquad\qquad \log(1+x) \leq x, \forall x > -1$$

$$= \frac{4\varepsilon^2}{1-4\varepsilon^2}$$

$$\leq 8\varepsilon^2 \qquad\qquad\qquad\qquad\qquad \varepsilon \leq \frac{1}{4}.$$

Combining the last two equations, we see that the testing failure probability is at least $\frac{1}{2}\left(1 - \sqrt{4n\varepsilon^2}\right)$. Thus, if we set $\varepsilon = \frac{1}{4\sqrt{n}}$, the failure probability is at least $\frac{1}{4}$. $\qquad\square$

## B  Proofs for general MDPs

In this section, we provide the proofs for our main results in Section 4 for general MDPs. Again, we can assume that H + B is an integer, which only affects the sample complexity by a constant multiple < 2.

First we develop more notation which will be useful in the setting of general MDPs. Recall we defined, for any policy $\pi$, that $\mathcal{R}^\pi$ is the set of states which are recurrent in the Markov chain $P_\pi$, and $\mathcal{T}^\pi = \mathcal{S}\setminus\mathcal{R}^\pi$ is the set of transient states. We now present a standard decomposition of Markov chains [14, Appendix A]. For any policy $\pi$, possibly after reordering states so that the recurrent states appear first (and are grouped into disjoint irreducible closed sets), we can decompose

$$P_\pi = \begin{bmatrix} X_\pi & 0 \\ Y_\pi & Z_\pi \end{bmatrix} \tag{10}$$

such that $X_\pi$ are probabilities of transitions between states which are recurrent under $\pi$, $Y_\pi$ are probabilities of transitions from $\mathcal{T}^\pi$ into $\mathcal{R}^\pi$, and $Z_\pi$ are probabilities of transitions between states within $\mathcal{T}^\pi$. Furthermore, supposing there are $k$ irreducible closed blocks within $\mathcal{R}^\pi$, $X_\pi$ is block-diagonal of the form

$$X_\pi = \begin{bmatrix} X_{\pi,1} & 0 & \cdots & 0 \\ 0 & X_{\pi,2} & \cdots & 0 \\ \vdots & \vdots & \ddots & \vdots \\ 0 & 0 & \cdots & X_{\pi,k} \end{bmatrix}.$$

The limiting matrix of the Markov chain induced by policy $\pi$ is defined as the matrix

$$P_\pi^\infty = \text{C-}\lim_{T\to\infty} P_\pi^T = \lim_{T\to\infty}\frac{1}{T}\sum_{t=0}^{T-1} P_\pi^t.$$

$P_\pi^\infty$ is a stochastic matrix (all rows positive and sum to 1) since $\mathcal{S}$ is finite. We also have $P_\pi P_\pi^\infty = P_\pi^\infty = P_\pi^\infty P_\pi$. Additionally, $\rho^\pi = P_\pi^\infty r_\pi$. In terms of our decomposition, we have

$$P_\pi^\infty = \begin{bmatrix} X_\pi^\infty & 0 \\ Y_\pi^\infty & 0 \end{bmatrix} \tag{11}$$

where

$$X_\pi^\infty = \begin{bmatrix} X_{\pi,1}^\infty & 0 & \cdots & 0 \\ 0 & X_{\pi,2}^\infty & \cdots & 0 \\ \vdots & \vdots & \ddots & \vdots \\ 0 & 0 & \cdots & X_{\pi,k}^\infty \end{bmatrix},$$

each $X_{\pi,i}^\infty = \mathbf{1}x_{\pi,i}^\top$ for some stochastic row vector $x_{\pi,i}^\top$, and $Y_\pi^\infty = (I - Z_\pi)^{-1}Y_\pi X_\pi^\infty$. Also we have $(I - Z_\pi)^{-1} = \sum_{t=0}^\infty Z_\pi^t$, and $\sum_{t=0}^\infty Z_\pi^t Y_\pi = (I - Z_\pi)^{-1}Y_\pi$ has stochastic rows (each row is a probability distribution, that is all entries are positive and sum to 1).

With the same arrangement of states as within the above decomposition of $P_\pi$ (10), let

$$V_\gamma^\pi = \begin{bmatrix} \overline{V_\gamma^\pi} \\ \underline{V_\gamma^\pi} \end{bmatrix}$$

decompose $V_\gamma^\pi$ into recurrent and transient states, and generally we use this same notation for any vector $x \in \mathbb{R}^\mathcal{S}$: we let $\overline{x}$ list the values of $x_s$ for recurrent $x \in \mathcal{R}^\pi$, $\underline{x}$ contain $x_s$ for $s \in \mathcal{T}^\pi$, and we assume the entire $x$ has been rearranged so that $x = [\overline{x}\ \underline{x}]^\top$. Note that the rearrangement of states depends on the policy $\pi$ so this notation has potential for confusion if applied to objects relating to multiple policies at once, but the policy determining the rearrangement will always be clear from context in our arguments.

The main reason we decompose $P_\pi$ into recurrent and transient states is the following key observation.

**Lemma 17.** *For any policy $\pi$, if $s, s'$ are in the same recurrent block of the Markov chain with transition matrix $P_\pi$, then $\rho^\star(s) = \rho^\star(s')$.*

*Proof.* Define the history-dependent policy $\tilde\pi$ which follows $\pi$ until its history first contains $s'$, after which point it follows $\pi^\star$. Since $\rho^\star(s)$ is the optimal gain achievable starting at $s$ by following any history-dependent policy [14], we have $\rho^\star(s) \geq \rho^{\tilde\pi}(s) := \lim_{T\to\infty} \frac{1}{T}\mathbb{E}_s^{\tilde\pi} \sum_{t=0}^{T-1} R_t$ (where $\mathbb{E}_s^{\tilde\pi}$ is defined in the natural way from the distribution over trajectories $(S_0, A_0, \dots)$ where $A_t \sim \tilde\pi(S_0, A_0, \dots, S_t)$ and $S_{t+1} \sim P(\cdot \mid S_t, A_t)$). Let $T_{s'} = \inf\{t \geq 1 : S_t = s'\}$ be the hitting time of state $s'$ and let $\mathcal{F}_{T_{s'}}$ be the stopped $\sigma$-algebra (with respect to the filtration where for all nonnegative integers $t$, $\mathcal{F}_t$ is the $\sigma$-algebra generated by $S_0, A_0, \dots, S_t, A_t$). Then

$$\lim_{T\to\infty} \frac{1}{T}\mathbb{E}_s^{\tilde\pi} \sum_{t=0}^{T-1} R_t = \lim_{T\to\infty} \frac{1}{T}\mathbb{E}_s^{\tilde\pi}\left[ \mathbb{E}_s^{\tilde\pi}\left[ \sum_{t=0}^{T-1} R_t \middle| \mathcal{F}_{T_{s'}} \right] \right]$$

$$= \lim_{T\to\infty} \frac{1}{T}\mathbb{E}_s^{\tilde\pi}\left[ \sum_{t=0}^{T_{s'}-1} R_t + \mathbb{E}_s^{\tilde\pi}\left[ \sum_{t=T_{s'}}^{T-1} R_t \middle| \mathcal{F}_{T_{s'}} \right] \right]$$

$$= \lim_{T\to\infty} \frac{1}{T}\mathbb{E}_s^{\tilde\pi}\left[ \sum_{t=0}^{T_{s'}-1} R_t + g(T, T_{s'}) \right]$$

$$= \lim_{T\to\infty} \frac{1}{T}\mathbb{E}_s^{\pi}\left[ \sum_{t=0}^{T_{s'}-1} R_t + g(T, T_{s'}) \right]$$

$$\geq \lim_{T\to\infty} \frac{1}{T}\mathbb{E}_s^{\pi}\left[ g(T, T_{s'}) \right]$$

where $g(T, k) := \mathbb{E}_{s'}^{\pi^\star}\left[ \sum_{t=0}^{T-k-1} R_t \right]$, and we used the tower property, $\mathcal{F}_{T_{s'}}$-measurability of $\sum_{t=0}^{T_{s'}-1} R_t$, the strong Markov property, and the definition of $\tilde\pi$. Now note that $T_{s'} < \infty$ almost surely since $s$ and $s'$ are in the same recurrent block, and on the event $\{T_{s'} = k\}$ for any natural number $k$, we have that

$$\lim_{T\to\infty} \frac{1}{T}g(T, k) = \lim_{T\to\infty} \frac{1}{T}\mathbb{E}_{s'}^{\pi^\star}\left[ \sum_{t=0}^{T-k-1} R_t \right] = \rho^\star(s')$$

because we can bound

$$\frac{1}{T}\mathbb{E}_{s'}^{\pi^\star}\left[\sum_{t=0}^{T-1} R_t\right] - \frac{k}{T} \leq \frac{1}{T}\mathbb{E}_{s'}^{\pi^\star}\left[\sum_{t=0}^{T-k-1} R_t\right] \leq \frac{1}{T}\mathbb{E}_{s'}^{\pi^\star}\left[\sum_{t=0}^{T-1} R_t\right]$$

and both sides converge to $\rho^\star(s')$. Therefore $\frac{g(T,T_{s'})}{T}$ converges almost surely to the constant $\rho^\star(s')$, and also this random variable is bounded by 1, so by the dominated convergence theorem we have

$$\lim_{T\to\infty} \frac{1}{T}\mathbb{E}_s^\pi\left[g(T,T_{s'})\right] = \mathbb{E}_s^\pi\left[\lim_{T\to\infty} \frac{1}{T}g(T,T_{s'})\right] = \rho^\star(s').$$

Thus we have shown that $\rho^\star(s) \geq \rho^\star(s')$. Since $s$ and $s'$ were arbitrary states in the same recurrent block we also have $\rho^\star(s') \geq \rho^\star(s)$, and thus $\rho^\star(s) = \rho^\star(s')$ as desired. □

**Lemma 18.** *For any state $s$ which is transient under a policy $\pi$, if the MDP satisfies the bounded transient time assumption with parameter* B*, we have*

$$\left\|\sum_{t=0}^\infty e_s^\top Z_\pi^t\right\|_1 \leq \mathsf{B}.$$

*Proof.* Let $T = \inf\{t : S_t \in \mathcal{R}^\pi\}$. Notice that $\left\|e_s^\top Z_\pi^t\right\|_1 = \mathbb{P}_s^\pi(T > t)$. Therefore, we have

$$\begin{aligned}
\left\|\sum_{t=0}^\infty e_s^\top Z_\pi^t\right\|_1 &\leq \sum_{t=0}^\infty \left\|e_s^\top Z_\pi^t\right\|_1 \\
&= \sum_{t=0}^\infty \mathbb{P}_s^\pi(T > t) \\
&= \mathbb{E}_s^\pi[T] \\
&\leq \mathsf{B},
\end{aligned}$$

where we used a well-known formula for the expectation of nonnegative-integer-valued random variables, and the bounded transient time assumption. □

**Lemma 19.** *Let $s$ be a transient state under $P_\pi$. Then*

$$e_s^\top(I - \gamma P_\pi)^{-1} = \left[\underline{e_s}^\top \sum_{k=1}^\infty \gamma^k Z_\pi^{k-1} Y_\pi (I - \gamma X_\pi)^{-1} \quad \underline{e_s}^\top \sum_{t=0}^\infty \gamma^t Z_\pi^t\right].$$

*Proof.* Using the decomposition of $P_\pi$, we can calculate for any integer $t \geq 1$ that

$$P_\pi^t = \begin{bmatrix} X_\pi^t & 0 \\ \sum_{k=1}^t Z_\pi^{k-1} Y_\pi X_\pi^{t-k} & Z_\pi^t \end{bmatrix}.$$

Therefore, we have

$$\begin{aligned}
e_s^\top(I - \gamma P_\pi)^{-1} &= e_s^\top \sum_{t=0}^\infty \gamma^t P_\pi^t \\
&= \left[\underline{e_s}^\top \sum_{t=0}^\infty \gamma^t \sum_{k=1}^t Z_\pi^{k-1} Y_\pi X_\pi^{t-k} \quad \underline{e_s}^\top \sum_{t=0}^\infty \gamma^t Z_\pi^t\right] \\
&= \left[\underline{e_s}^\top \sum_{k=1}^\infty \sum_{t=k}^\infty \gamma^t Z_\pi^{k-1} Y_\pi X_\pi^{t-k} \quad \underline{e_s}^\top \sum_{t=0}^\infty \gamma^t Z_\pi^t\right] \\
&= \left[\underline{e_s}^\top \sum_{k=1}^\infty \gamma^k Z_\pi^{k-1} Y_\pi \sum_{t=k}^\infty \gamma^{t-k} X_\pi^{t-k} \quad \underline{e_s}^\top \sum_{t=0}^\infty \gamma^t Z_\pi^t\right] \\
&= \left[\underline{e_s}^\top \sum_{k=1}^\infty \gamma^k Z_\pi^{k-1} Y_\pi (I - \gamma X_\pi)^{-1} \quad \underline{e_s}^\top \sum_{t=0}^\infty \gamma^t Z_\pi^t\right].
\end{aligned}$$

Note that we are able to rearrange the order of the summation in the third equality because all summands are (elementwise) positive. □

### B.1 Proof of Theorem 6

Theorem 6, our result which helps reduce general average reward MDPs to discounted MDPs, is proven as a straightforward consequence of the following sequence of lemmas, some of which will also be needed for the proof of our discounted MDP sample complexity bound Theorem 7.

**Lemma 20.** *We have*

$$\left\| V_\gamma^{\pi^\star} - \frac{1}{1-\gamma}\rho^\star \right\|_\infty \leq \|h^\star\|_{\mathrm{span}}.$$

*Proof.* We begin by observing that $\pi^\star$ satisfies

$$\rho^\star + h^\star = r_{\pi^\star} + P_{\pi^\star}h^\star.$$

Therefore, it holds that

$$
\begin{aligned}
V_\gamma^{\pi^\star} &= (I - \gamma P_{\pi^\star})^{-1} r_{\pi^\star} \\
&= (I - \gamma P_{\pi^\star})^{-1}(\rho^\star + h^\star - P_{\pi^\star}h^\star) \\
&= (I - \gamma P_{\pi^\star})^{-1}\rho^\star + (I - \gamma P_{\pi^\star})^{-1}(I - P_{\pi^\star})h^\star.
\end{aligned}
$$

Since $P_{\pi^\star}\rho^\star = \rho^\star$, we can calculate that

$$(I - \gamma P_{\pi^\star})^{-1}\rho^\star = \sum_{t\geq 0}\gamma^t P_{\pi^\star}^t \rho^\star = \sum_{t\geq 0}\gamma^t \rho^\star = \frac{1}{1-\gamma}\rho^\star.$$

It also holds that

$$
\begin{aligned}
(I - \gamma P_{\pi^\star})^{-1}(I - P_{\pi^\star}) &= \sum_{t\geq 0}\gamma^t P_{\pi^\star}^t(I - P_{\pi^\star}) \\
&= \sum_{t\geq 0}\gamma^t P_{\pi^\star}^t - \sum_{t\geq 0}\gamma^t P_{\pi^\star}^{t+1} \\
&= P_{\pi^\star} + \sum_{t\geq 0}(\gamma^{t+1} - \gamma^t)P_{\pi^\star}^{t+1} \quad\quad (12)
\end{aligned}
$$

and $\sum_{t\geq 0}\gamma^{t+1} - \gamma^t = (\gamma - 1)\sum_{t\geq 0}\gamma^t = -1$. Therefore (12) is the difference of two stochastic matrices, and so it follows that

$$\left\| (I - \gamma P_{\pi^\star})^{-1}(I - P_{\pi^\star})h^\star \right\|_\infty \leq \|h^\star\|_{\mathrm{span}}.$$

$\square$

**Lemma 21.** *If $\pi_\gamma^\star$ is optimal for the discounted MDP $(P, r, \gamma)$ and $s$ is recurrent under $\pi_\gamma^\star$, then*

$$\left| V_\gamma^{\pi_\gamma^\star}(s) - \frac{1}{1-\gamma}\rho^\star(s) \right| \leq \|h^\star\|_{\mathrm{span}}$$

*and*

$$\left| V_\gamma^{\pi_\gamma^\star}(s) - \frac{1}{1-\gamma}\rho^{\pi_\gamma^\star}(s) \right| \leq 2\|h^\star\|_{\mathrm{span}}.$$

*These facts can be written as $\left\| \overline{V_\gamma^{\pi_\gamma^\star}} - \frac{1}{1-\gamma}\overline{\rho^\star} \right\|_\infty \leq \|h^\star\|_{\mathrm{span}}$ and $\left\| \overline{V_\gamma^{\pi_\gamma^\star}} - \frac{1}{1-\gamma}\overline{\rho^{\pi_\gamma^\star}} \right\|_\infty \leq 2\|h^\star\|_{\mathrm{span}}$ respectively.*

*Proof.* First note that if $s$ is recurrent for the Markov chain $P_{\pi_\gamma^\star}$, then all states in the support of $e_s^\top P_{\pi_\gamma^\star}$ are in the same recurrent block as state $s$, and $\rho^\star$ is constant (and equal to $\rho^\star(s)$) within this recurrent block by Lemma 17. The (unmodified) Bellman equation states that

$$\rho^\star(s) + h^\star(s) = \max_{a: P_{sa}\rho^\star = \rho^\star(s)} r_{sa} + P_{sa}h^\star.$$

Since we established that $e_s^\top P_{\pi_\gamma^\star} \rho^\star = \rho^\star(s)$, all actions $a$ in the support of $\pi_\gamma^\star(a \mid s)$ satisfy $P_{sa}\rho^\star = \rho^\star(s)$, and therefore

$$
\begin{aligned}
\rho^\star(s) + h^\star(s) &= \max_{a:P_{sa}\rho^\star=\rho^\star(s)} r_{sa} + P_{sa}h^\star \\
&\geq \sum_{a\in\mathcal{A}} \pi_\gamma^\star(a \mid s)\,(r_{sa} + P_{sa}h^\star) \\
&= e_s^\top \left( r_{\pi_\gamma^\star} + P_{\pi_\gamma^\star}h^\star \right).
\end{aligned}
$$

Since this holds for all $s \in \mathcal{R}^{\pi_\gamma^\star}$, we can rearrange to obtain that

$$
\overline{r_{\pi_\gamma^\star}} \leq \overline{\rho^\star} + \overline{h^\star} - \overline{P_{\pi_\gamma^\star}h^\star} = \overline{\rho^\star} + \overline{h^\star} - X_{\pi_\gamma^\star}\overline{h^\star}.
$$

Now we can follow an argument which is similar to that of [23, Lemma 2]. We have

$$
\begin{aligned}
\overline{V_\gamma^{\pi_\gamma^\star}} &= \overline{(I - \gamma P_{\pi_\gamma^\star})^{-1}r_{\pi_\gamma^\star}} \\
&= (I - X_{\pi_\gamma^\star})^{-1}\overline{r_{\pi_\gamma^\star}} \\
&\leq (I - X_{\pi_\gamma^\star})^{-1}\left( \overline{\rho^\star} + \overline{h^\star} - X_{\pi_\gamma^\star}\overline{h^\star} \right)
\end{aligned}
$$

using monotonicity of $(I - X_{\pi_\gamma^\star})^{-1}$ in the final inequality. Due to the observation above that for all $s \in \mathcal{R}^{\pi_\gamma^\star}$, all actions $a$ in the support of $\pi_\gamma^\star(a \mid s)$ satisfy $P_{sa}\rho^\star = \rho^\star(s)$, we have $X_{\pi_\gamma^\star}\overline{\rho^\star} = \overline{\rho^\star}$. Therefore we have

$$
(I - X_{\pi_\gamma^\star})^{-1}\overline{\rho^\star} = \sum_{t=0}^{\infty} \gamma^t X_{\pi_\gamma^\star}\overline{\rho^\star} = \sum_{t=0}^{\infty} \gamma^t\overline{\rho^\star} = \frac{1}{1-\gamma}\overline{\rho^\star}.
$$

For the second term, by using an argument which is completely analogous to that used in Lemma 20 we have $\left\| (I - X_{\pi_\gamma^\star})^{-1}\left( \overline{h^\star} - X_{\pi_\gamma^\star}\overline{h^\star} \right) \right\|_\infty \leq \|h^\star\|_{\text{span}}$. Combining these steps we obtain that

$$
\overline{V_\gamma^{\pi_\gamma^\star}} - \frac{1}{1-\gamma}\overline{\rho^\star} \leq \|h^\star\|_{\text{span}}\mathbf{1}.
$$

To obtain a lower bound, we can combine the optimality of $\pi_\gamma^\star$ for the $\gamma$-discounted problem with Lemma 20 to obtain the bound

$$
\overline{V_\gamma^{\pi_\gamma^\star}} - \frac{1}{1-\gamma}\overline{\rho^\star} \geq \overline{V_\gamma^{\pi^\star}} - \frac{1}{1-\gamma}\overline{\rho^\star} \geq \|h^\star\|_{\text{span}}\mathbf{1}.
$$

Therefore we can conclude that $\left\| \overline{V_\gamma^{\pi_\gamma^\star}} - \frac{1}{1-\gamma}\overline{\rho^\star} \right\|_\infty \leq \|h^\star\|_{\text{span}}$.

For the second bound in the lemma statement, we first note that, as observed in [20],

$$
P_{\pi_\gamma^\star}^\infty V_\gamma^{\pi_\gamma^\star} = P_{\pi_\gamma^\star}^\infty \sum_{t=0}^{\infty} \gamma^t P_{\pi_\gamma^\star}^t r_{\pi_\gamma^\star} = \sum_{t=0}^{\infty} \gamma^t P_{\pi_\gamma^\star}^\infty r_{\pi_\gamma^\star} = \frac{1}{1-\gamma}\rho^{\pi_\gamma^\star}.
$$

Also, as discussed previously, if $s \in \mathcal{R}^{\pi_\gamma^\star}$ then $e_s^\top P_{\pi_\gamma^\star}\rho^\star = \rho^\star(s)$, so then we also have $e_s^\top P_{\pi_\gamma^\star}^\infty \rho^\star = \rho^\star(s)$ (which can be seen directly from the definition of the limiting matrix $P_{\pi_\gamma^\star}^\infty$). Equivalently, $e_s^\top(I - P_{\pi_\gamma^\star}^\infty)\rho^\star = 0$. Using both of these two observations, we have

$$
\begin{aligned}
V_\gamma^{\pi_\gamma^\star}(s) - \frac{1}{1-\gamma}\rho^{\pi_\gamma^\star}(s) &= e_s^\top(I - P_{\pi_\gamma^\star}^\infty)V_\gamma^{\pi_\gamma^\star} \\
&= e_s^\top(I - P_{\pi_\gamma^\star}^\infty)(V_\gamma^{\pi_\gamma^\star} - \frac{1}{1-\gamma}\rho^\star) \\
&= \overline{e_s}^\top(I - X_{\pi_\gamma^\star}^\infty)(\overline{V_\gamma^{\pi_\gamma^\star}} - \frac{1}{1-\gamma}\overline{\rho^\star}).
\end{aligned}
$$

Therefore, we obtain

$$
\begin{aligned}
\left\| \overline{V_\gamma^{\pi_\gamma^\star}} - \frac{1}{1-\gamma}\overline{\rho^{\pi_\gamma^\star}} \right\|_\infty &\leq \left\| (I - X_{\pi_\gamma^\star}^\infty)(\overline{V_\gamma^{\pi_\gamma^\star}} - \frac{1}{1-\gamma}\overline{\rho^\star}) \right\|_\infty \\
&\leq \left\| \overline{V_\gamma^{\pi_\gamma^\star}} - \frac{1}{1-\gamma}\overline{\rho^\star} \right\|_{\mathrm{span}} \\
&\leq 2 \left\| \overline{V_\gamma^{\pi_\gamma^\star}} - \frac{1}{1-\gamma}\overline{\rho^\star} \right\|_\infty \\
&\leq 2 \left\| h^\star \right\|_{\mathrm{span}}
\end{aligned}
$$

using the first bound from the lemma statement in the final inequality. $\qquad\square$

**Lemma 22.** *We have*

$$
\left\| V_\gamma^{\pi_\gamma^\star} - \frac{1}{1-\gamma}\rho^\star \right\|_\infty \leq \mathsf{B} + \|h^\star\|_{\mathrm{span}}
$$

*and*

$$
\left\| V_\gamma^{\pi_\gamma^\star} - \frac{1}{1-\gamma}\rho^{\pi_\gamma^\star} \right\|_\infty \leq \mathsf{B} + 2\|h^\star\|_{\mathrm{span}}.
$$

*Proof.* Note that by combining with Lemma 21, it suffices to prove for any transient state $s \in \mathcal{T}^{\pi_\gamma^\star}$ that

$$
\left| V_\gamma^{\pi_\gamma^\star}(s) - \frac{1}{1-\gamma}\rho^\star(s) \right| \leq \mathsf{B} + \|h^\star\|_{\mathrm{span}}
$$

and

$$
\left| V_\gamma^{\pi_\gamma^\star}(s) - \frac{1}{1-\gamma}\rho^{\pi_\gamma^\star}(s) \right| \leq \mathsf{B} + 2\|h^\star\|_{\mathrm{span}}.
$$

Let $s$ be transient under $\pi_\gamma^\star$. Then starting by using Lemma 19, we can calculate

$$
\begin{aligned}
V_\gamma^{\pi_\gamma^\star}(s) &= e_s^\top (I - \gamma P_{\pi_\gamma^\star})^{-1} r_{\pi_\gamma^\star} \\
&= \sum_{t=0}^\infty \gamma^t \underline{e_s}^\top Z_{\pi_\gamma^\star}^t \underline{r_{\pi_\gamma^\star}} + \gamma \sum_{t=0}^\infty \gamma^t \underline{e_s}^\top Z_{\pi_\gamma^\star}^t Y_{\pi_\gamma^\star} (I - \gamma X_{\pi_\gamma^\star})^{-1} \overline{r_{\pi_\gamma^\star}} \\
&= \sum_{t=0}^\infty \gamma^t \underline{e_s}^\top Z_{\pi_\gamma^\star}^t \underline{r_{\pi_\gamma^\star}} + \gamma \sum_{t=0}^\infty \gamma^t \underline{e_s}^\top Z_{\pi_\gamma^\star}^t Y_{\pi_\gamma^\star} \overline{V_\gamma^{\pi_\gamma^\star}} \\
&\leq \sum_{t=0}^\infty \underline{e_s}^\top Z_{\pi_\gamma^\star}^t \underline{r_{\pi_\gamma^\star}} + \left( \sum_{t=0}^\infty \underline{e_s}^\top Z_{\pi_\gamma^\star}^t Y_{\pi_\gamma^\star} \right) \overline{V_\gamma^{\pi_\gamma^\star}}.
\end{aligned} \tag{13}
$$

By Lemma 18 we have that

$$
\sum_{t=0}^\infty \underline{e_s}^\top Z_{\pi_\gamma^\star}^t \underline{r_{\pi_\gamma^\star}} \leq \left\| \sum_{t=0}^\infty \underline{e_s}^\top Z_{\pi_\gamma^\star}^t \right\|_1 \left\| \underline{r_{\pi_\gamma^\star}} \right\|_\infty \leq \mathsf{B}.
$$

Now we can obtain the two bounds in the lemma statement by bounding the second term of (13) in two different ways. For the first bound in the lemma statement, we can use the first bound in Lemma

21 to calculate that

$$
\left(\sum_{t=0}^{\infty}\underline{e_s^\top Z_{\pi_\gamma^\star}^t Y_{\pi_\gamma^\star}}\right)\overline{V_\gamma^{\pi_\gamma^\star}} \leq \left(\sum_{t=0}^{\infty}\underline{e_s^\top Z_{\pi_\gamma^\star}^t Y_{\pi_\gamma^\star}}\right)\frac{1}{1-\gamma}\overline{\rho^\star} + \left(\sum_{t=0}^{\infty}\underline{e_s^\top Z_{\pi_\gamma^\star}^t Y_{\pi_\gamma^\star}}\right)\left\|\overline{V_\gamma^{\pi_\gamma^\star}} - \frac{1}{1-\gamma}\overline{\rho^\star}\right\|_\infty \mathbf{1}
$$

$$
= \left(\sum_{t=0}^{\infty}\underline{e_s^\top Z_{\pi_\gamma^\star}^t Y_{\pi_\gamma^\star}}\right)\frac{1}{1-\gamma}\overline{\rho^\star} + \left\|\overline{V_\gamma^{\pi_\gamma^\star}} - \frac{1}{1-\gamma}\overline{\rho^\star}\right\|_\infty
$$

$$
\leq \left(\sum_{t=0}^{\infty}\underline{e_s^\top Z_{\pi_\gamma^\star}^t Y_{\pi_\gamma^\star}}\right)\frac{1}{1-\gamma}\overline{\rho^\star} + \|h^\star\|_{\mathrm{span}}
$$

$$
= \left(\sum_{t=0}^{\infty}\underline{e_s^\top Z_{\pi_\gamma^\star}^t Y_{\pi_\gamma^\star}}\right)\frac{1}{1-\gamma}X_{\pi_\gamma^\star}^\infty\overline{\rho^\star} + \|h^\star\|_{\mathrm{span}}
$$

$$
= \left(\sum_{t=0}^{\infty}\underline{e_s^\top Z_{\pi_\gamma^\star}^t Y_{\pi_\gamma^\star}X_{\pi_\gamma^\star}^\infty}\right)\frac{1}{1-\gamma}\overline{\rho^\star} + \|h^\star\|_{\mathrm{span}}
$$

$$
= \underline{e_s^\top Y_{\pi_\gamma^\star}^\infty}\frac{1}{1-\gamma}\overline{\rho^\star} + \|h^\star\|_{\mathrm{span}}
$$

$$
= \frac{1}{1-\gamma}e_s^\top P_{\pi_\gamma^\star}^\infty\rho^\star + \|h^\star\|_{\mathrm{span}}
$$

$$
\leq \frac{1}{1-\gamma}\rho^\star(s) + \|h^\star\|_{\mathrm{span}}
$$

where we used the fact that $X_{\pi_\gamma^\star}^\infty\overline{\rho^\star} = \overline{\rho^\star}$ and then that $e_s^\top P_{\pi_\gamma^\star}^\infty\rho^\star \leq \rho^\star(s)$. This gives an upper bound of

$$
V_\gamma^{\pi_\gamma^\star} \leq \frac{1}{1-\gamma}\rho^\star(s) + \mathsf{B} + \|h^\star\|_{\mathrm{span}}\,.
$$

Combining with the lower bound

$$
V_\gamma^{\pi_\gamma^\star}(s) \geq V_\gamma^{\pi^\star}(s) \geq \frac{1}{1-\gamma}\rho^\star(s) - \|h^\star\|_{\mathrm{span}}\,,
$$

we obtain that

$$
\left\|V_\gamma^{\pi_\gamma^\star} - \frac{1}{1-\gamma}\rho^\star\right\|_\infty \leq \mathsf{B} + \|h^\star\|_{\mathrm{span}}
$$

which is the first bound in the lemma statement.

To obtain the second bound in the lemma statement, using the second bound from Lemma 21, we can calculate for the second term in (13) that

$$
\left(\sum_{t=0}^{\infty} \underline{e_s}^{\top} Z_{\pi_\gamma^\star}^t Y_{\pi_\gamma^\star}\right) \overline{V_\gamma^{\pi_\gamma^\star}} \leq \left(\sum_{t=0}^{\infty} \underline{e_s}^{\top} Z_{\pi_\gamma^\star}^t Y_{\pi_\gamma^\star}\right) \frac{1}{1-\gamma} \overline{\rho^{\pi_\gamma^\star}} + \left(\sum_{t=0}^{\infty} \underline{e_s}^{\top} Z_{\pi_\gamma^\star}^t Y_{\pi_\gamma^\star}\right) \left\|\overline{V_\gamma^{\pi_\gamma^\star}} - \frac{1}{1-\gamma}\overline{\rho^{\pi_\gamma^\star}}\right\|_\infty \mathbf{1}
$$

$$
= \left(\sum_{t=0}^{\infty} \underline{e_s}^{\top} Z_{\pi_\gamma^\star}^t Y_{\pi_\gamma^\star}\right) \frac{1}{1-\gamma} \overline{\rho^{\pi_\gamma^\star}} + \left\|\overline{V_\gamma^{\pi_\gamma^\star}} - \frac{1}{1-\gamma}\overline{\rho^{\pi_\gamma^\star}}\right\|_\infty
$$

$$
\leq \left(\sum_{t=0}^{\infty} \underline{e_s}^{\top} Z_{\pi_\gamma^\star}^t Y_{\pi_\gamma^\star}\right) \frac{1}{1-\gamma} \overline{\rho^{\pi_\gamma^\star}} + 2\left\|h^\star\right\|_{\mathrm{span}}
$$

$$
= \left(\sum_{t=0}^{\infty} \underline{e_s}^{\top} Z_{\pi_\gamma^\star}^t Y_{\pi_\gamma^\star}\right) \frac{1}{1-\gamma} \overline{P_{\pi_\gamma^\star}^\infty r_{\pi_\gamma^\star}} + 2\left\|h^\star\right\|_{\mathrm{span}}
$$

$$
= \left(\sum_{t=0}^{\infty} \underline{e_s}^{\top} Z_{\pi_\gamma^\star}^t Y_{\pi_\gamma^\star}\right) \frac{1}{1-\gamma} X_{\pi_\gamma^\star}^\infty \overline{r_{\pi_\gamma^\star}} + 2\left\|h^\star\right\|_{\mathrm{span}}
$$

$$
= \frac{1}{1-\gamma} \underline{e_s}^{\top} Y_{\pi_\gamma^\star}^\infty \overline{r_{\pi_\gamma^\star}} + 2\left\|h^\star\right\|_{\mathrm{span}}
$$

$$
= \frac{1}{1-\gamma} e_s^{\top} P_{\pi_\gamma^\star}^\infty r_{\pi_\gamma^\star} + 2\left\|h^\star\right\|_{\mathrm{span}}
$$

$$
= \frac{1}{1-\gamma} \rho^{\pi_\gamma^\star}(s) + 2\left\|h^\star\right\|_{\mathrm{span}}
$$

where in the second equality we used the fact that $\left(\sum_{t=0}^{\infty} \underline{e_s}^{\top} Z_{\pi_\gamma^\star}^t Y_{\pi_\gamma^\star}\right)$ is a probability distribution, and in the final steps we used the decomposition of $P_{\pi_\gamma^\star}^\infty$ and the fact that $\rho^{\pi_\gamma^\star} = P_{\pi_\gamma^\star}^\infty r_{\pi_\gamma^\star}$.

Therefore by combining these steps we obtain that

$$
V_\gamma^{\pi_\gamma^\star}(s) \leq \mathsf{B} + 2\left\|h^\star\right\|_{\mathrm{span}} + \frac{1}{1-\gamma} \rho^{\pi_\gamma^\star}(s).
$$

Combining with the lower bound

$$
V_\gamma^{\pi_\gamma^\star}(s) \geq V_\gamma^{\pi^\star}(s) \geq \frac{1}{1-\gamma} \rho^\star(s) - \left\|h^\star\right\|_{\mathrm{span}} \geq \frac{1}{1-\gamma} \rho^{\pi_\gamma^\star}(s) - \left\|h^\star\right\|_{\mathrm{span}},
$$

we obtain the desired bound

$$
\left|V_\gamma^{\pi_\gamma^\star}(s) - \frac{1}{1-\gamma} \rho^{\pi_\gamma^\star}(s)\right| \leq \mathsf{B} + 2\left\|h^\star\right\|_{\mathrm{span}}.
$$

$\square$

**Lemma 23.** *If $\pi$ satisfies $V_\gamma^\pi \geq V_\gamma^{\pi_\gamma^\star} - \delta\mathbf{1}$, then*

$$
\left\|V_\gamma^\pi - \frac{1}{1-\gamma} \rho^\pi\right\|_\infty \leq 3\mathsf{B} + 2\left\|h^\star\right\|_{\mathrm{span}} + \delta.
$$

*Proof.* Similar to the proof of Lemmas 21 and 22, we will first establish a bound for the states which are recurrent under $\pi$. Specifically, we will first show that if $s$ is recurrent under $\pi$ we have

$$
\left|V_\gamma^\pi(s) - \frac{1}{1-\gamma} \rho^\pi(s)\right| \leq 2\mathsf{B} + 2\left\|h^\star\right\|_{\mathrm{span}} + \delta. \tag{14}
$$

Letting $s \in \mathcal{R}^\pi$, following steps which are similar to the proof of the second part of Lemma 21, we have

$$
V_\gamma^\pi(s) - \frac{1}{1-\gamma} \rho^\pi(s) = e_s^{\top}(I - P_\pi^\infty) V_\gamma^\pi
$$

$$
= e_s^{\top}(I - P_\pi^\infty)(V_\gamma^\pi - \frac{1}{1-\gamma} \rho^\star)
$$

$$
= e_s^{\top}(I - P_\pi^\infty)(V_\gamma^{\pi_\gamma^\star} - \frac{1}{1-\gamma} \rho^\star) + e_s^{\top}(I - P_\pi^\infty)(V_\gamma^\pi - V_\gamma^{\pi_\gamma^\star})
$$

using the fact discussed in Lemma 21 that $e_s^\top(I - P_\pi^\infty)\rho^\star = 0$ since $s$ is recurrent under $\pi$. Then by triangle inequality, we obtain

$$
\begin{aligned}
\left|V_\gamma^\pi(s) - \frac{1}{1-\gamma}\rho^\pi(s)\right| &\leq \left|e_s^\top(I - P_\pi^\infty)(V_\gamma^{\pi_\gamma^\star} - \frac{1}{1-\gamma}\rho^\star)\right| + \left|e_s^\top(I - P_\pi^\infty)(V_\gamma^\pi - V_\gamma^{\pi_\gamma^\star})\right| \\
&\leq \left\|V_\gamma^{\pi_\gamma^\star} - \frac{1}{1-\gamma}\rho^\star\right\|_{\mathrm{span}} + \left\|V_\gamma^\pi - V_\gamma^{\pi_\gamma^\star}\right\|_{\mathrm{span}} \\
&\leq 2\left\|V_\gamma^{\pi_\gamma^\star} - \frac{1}{1-\gamma}\rho^\star\right\|_\infty + \delta \\
&\leq 2\mathsf{B} + 2\left\|h^\star\right\|_{\mathrm{span}} + \delta,
\end{aligned}
$$

where we used the facts that $\|\cdot\|_{\mathrm{span}} \leq 2\|\cdot\|_\infty$ and that $V_\gamma^{\pi_\gamma^\star} \geq V_\gamma^\pi \geq V_\gamma^{\pi_\gamma^\star} - \delta\mathbf{1}$.

Having established (14), we now extend to transient states using arguments similar to those for the second bound of Lemma 22. Let $s$ be transient under $\pi$. Then starting by using Lemma 19, we can calculate

$$
\begin{aligned}
V_\gamma^\pi(s) &= e_s^\top(I - \gamma P_\pi)^{-1}r_\pi \\
&= \sum_{t=0}^\infty \gamma^t \underline{e_s}^\top Z_\pi^t \underline{r_\pi} + \gamma \sum_{t=0}^\infty \gamma^t \underline{e_s}^\top Z_\pi^t Y_\pi (I - \gamma X_\pi)^{-1}\overline{r_\pi} \\
&= \sum_{t=0}^\infty \gamma^t \underline{e_s}^\top Z_\pi^t \underline{r_\pi} + \gamma \sum_{t=0}^\infty \gamma^t \underline{e_s}^\top Z_\pi^t Y_\pi \overline{V_\gamma^\pi} \\
&\leq \sum_{t=0}^\infty \underline{e_s}^\top Z_\pi^t \underline{r_\pi} + \left(\sum_{t=0}^\infty \underline{e_s}^\top Z_\pi^t Y_\pi\right)\overline{V_\gamma^\pi} \\
&\leq \left\|\sum_{t=0}^\infty \underline{e_s}^\top Z_\pi^t\right\|_1 \|\underline{r_\pi}\|_\infty + \left(\sum_{t=0}^\infty \underline{e_s}^\top Z_\pi^t Y_\pi\right)\overline{V_\gamma^\pi} \\
&\leq \mathsf{B} + \left(\sum_{t=0}^\infty \underline{e_s}^\top Z_\pi^t Y_\pi\right)\overline{V_\gamma^\pi} \qquad\qquad (15)
\end{aligned}
$$

using the bounded transient time assumption via Lemma 18 in the final step. Then we can calculate

$$
\begin{aligned}
\left(\sum_{t=0}^\infty \underline{e_s}^\top Z_\pi^t Y_\pi\right)\overline{V_\gamma^\pi} &\leq \left(\sum_{t=0}^\infty \underline{e_s}^\top Z_\pi^t Y_\pi\right)\frac{1}{1-\gamma}\overline{\rho^\pi} + \left(\sum_{t=0}^\infty \underline{e_s}^\top Z_\pi^t Y_\pi\right)\left\|\overline{V_\gamma^\pi} - \frac{1}{1-\gamma}\overline{\rho^\pi}\right\|_\infty \mathbf{1} \\
&= \left(\sum_{t=0}^\infty \underline{e_s}^\top Z_\pi^t Y_\pi\right)\frac{1}{1-\gamma}\overline{\rho^\pi} + \left\|\overline{V_\gamma^\pi} - \frac{1}{1-\gamma}\overline{\rho^\pi}\right\|_\infty \\
&\leq \left(\sum_{t=0}^\infty \underline{e_s}^\top Z_\pi^t Y_\pi\right)\frac{1}{1-\gamma}\overline{\rho^\pi} + 2\mathsf{B} + 2\|h^\star\|_{\mathrm{span}} + \delta \\
&= \left(\sum_{t=0}^\infty \underline{e_s}^\top Z_\pi^t Y_\pi\right)\frac{1}{1-\gamma}\overline{P_\pi^\infty r_\pi} + 2\mathsf{B} + 2\|h^\star\|_{\mathrm{span}} + \delta \\
&= \left(\sum_{t=0}^\infty \underline{e_s}^\top Z_\pi^t Y_\pi\right)\frac{1}{1-\gamma}X_\pi^\infty\overline{r_\pi} + 2\mathsf{B} + 2\|h^\star\|_{\mathrm{span}} + \delta \\
&= \frac{1}{1-\gamma}\underline{e_s}^\top Y_\pi^\infty\overline{r_\pi} + 2\mathsf{B} + 2\|h^\star\|_{\mathrm{span}} + \delta \\
&= \frac{1}{1-\gamma}e_s^\top P_\pi^\infty r_\pi + 2\mathsf{B} + 2\|h^\star\|_{\mathrm{span}} + \delta \\
&= \frac{1}{1-\gamma}\rho^\pi(s) + 2\mathsf{B} + 2\|h^\star\|_{\mathrm{span}} + \delta,
\end{aligned}
$$

where in the first equality we used the fact that $\left(\sum_{t=0}^{\infty} e_s^\top Z_\pi^t Y_\pi\right)$ is a probability distribution, in the second inequality we used the bound (14), and in the final steps we used the decomposition of $P_\pi^\infty$ and the fact that $\rho^\pi = P_\pi^\infty r_\pi$.

Therefore by combining this last bound with the bound (15), we have

$$V_\gamma^\pi(s) \leq 3\mathsf{B} + 2\left\|h^\star\right\|_{\mathrm{span}} + \delta + \frac{1}{1-\gamma}\rho^\pi(s).$$

Combining with the lower bound

$$V_\gamma^\pi(s) \geq V_\gamma^{\pi_\gamma^\star} - \delta \geq V_\gamma^{\pi^\star}(s) - \delta \geq \frac{1}{1-\gamma}\rho^\star(s) - \left\|h^\star\right\|_{\mathrm{span}} - \delta \geq \frac{1}{1-\gamma}\rho^\pi(s) - \left\|h^\star\right\|_{\mathrm{span}} - \delta,$$

we conclude that

$$\left|V_\gamma^\pi(s) - \frac{1}{1-\gamma}\rho^\pi(s)\right| \leq 3\mathsf{B} + 2\left\|h^\star\right\|_{\mathrm{span}} + \delta$$

as desired. $\qquad\square$

*Proof of Theorem 6.* Suppose $\pi$ is $\varepsilon_\gamma$-optimal for the discounted MDP $(P, r, \gamma)$. We can calculate that

$$
\begin{aligned}
\frac{1}{1-\gamma}\rho^\pi &\geq V_\gamma^\pi - (3\mathsf{B} + 2\left\|h^\star\right\|_{\mathrm{span}} + \varepsilon_\gamma) \\
&\geq V_\gamma^{\pi_\gamma^\star} - (3\mathsf{B} + 2\left\|h^\star\right\|_{\mathrm{span}} + 2\varepsilon_\gamma) \\
&\geq V_\gamma^{\pi^\star} - (3\mathsf{B} + 2\left\|h^\star\right\|_{\mathrm{span}} + 2\varepsilon_\gamma) \\
&\geq \frac{1}{1-\gamma}\rho^\star - (3\mathsf{B} + 3\left\|h^\star\right\|_{\mathrm{span}} + 2\varepsilon_\gamma),
\end{aligned}
$$

where in the first inequality we used Lemma 23, in the second inequality we used the fact that $\pi$ is $\varepsilon_\gamma$-optimal, in the third inequality we used the optimality of $\pi_\gamma^\star$ for the discounted MDP, and in the final inequality we used Lemma 20. Therefore by mulitplying both sides by $1 - \gamma$, we have that

$$\rho^\pi \geq \rho^\star - \frac{\varepsilon}{\mathsf{B} + \mathsf{H}}(3\mathsf{B} + 3\left\|h^\star\right\|_{\mathrm{span}} + 2\varepsilon_\gamma) \geq \rho^\star - \left(3\varepsilon + 2\frac{\varepsilon_\gamma}{\mathsf{B} + \mathsf{H}}\right)\varepsilon.$$

$\qquad\square$

## B.2 Proof of Theorem 7 (Discounted MDP Bounds)

In this section, we provide our main result on the sample complexity of general discounted MDPs.

Our proof relies on three lemmas that provide bounds on relevant variance parameters. The first lemma controls the variance for $\pi_\gamma^\star$ on recurrent states.

**Lemma 24.** *Letting $\pi_\gamma^\star$ be the optimal policy for the discounted MDP $(P, r, \gamma)$, if $\gamma \geq 1 - \frac{1}{\mathsf{B}+\mathsf{H}}$, we have*

$$\max_{s \in \mathcal{R}^{\pi_\gamma^\star}} \gamma \left|e_s^\top (I - \gamma P_{\pi_\gamma^\star})^{-1}\sqrt{\mathbb{V}_{P_{\pi_\gamma^\star}}\left[V_\gamma^{\pi_\gamma^\star}\right]}\right| \leq \sqrt{\frac{32}{5}\frac{\mathsf{B}+\mathsf{H}}{(1-\gamma)^2}}.$$

*Proof.* First, using the decomposition (10), we can calculate for any $s \in \mathcal{R}^{\pi_\gamma^\star}$ that

$$
\begin{aligned}
e_s^\top (I - \gamma P_{\pi_\gamma^\star})^{-1}\sqrt{\mathbb{V}_{P_{\pi_\gamma^\star}}\left[V_\gamma^{\pi_\gamma^\star}\right]} &= \overline{e_s}^\top (I - \gamma X_{\pi_\gamma^\star})^{-1}\sqrt{\mathbb{V}_{P_{\pi_\gamma^\star}}\left[V_\gamma^{\pi_\gamma^\star}\right]} \\
&= \overline{e_s}^\top (I - \gamma X_{\pi_\gamma^\star})^{-1}\sqrt{\mathbb{V}_{X_{\pi_\gamma^\star}}\left[V_\gamma^{\pi_\gamma^\star}\right]}.
\end{aligned}
$$

Also due to the decomposition, notice that set $\mathcal{R}^{\pi_\gamma^\star}$ is a closed set for the Markov chain with transition matrix $P_{\pi_\gamma^\star}$, and furthermore when restricting to the entries corresponding to this closed set we obtain the transition matrix $X_{\pi_\gamma^\star}$. Therefore we can apply Lemma 12 to this subchain to obtain that

$$\gamma \left\| (I - \gamma X_{\pi_\gamma^\star})^{-1} \sqrt{\mathbb{V}_{X_{\pi_\gamma^\star}} \left[ \overline{V_\gamma^{\pi_\gamma^\star}} \right]} \right\|_\infty \leq \sqrt{\frac{2}{1-\gamma}} \sqrt{\left\| \overline{\mathbb{V}^{\pi_\gamma^\star} \left[ \sum_{t=0}^\infty \gamma^t R_t \right]} \right\|_\infty}.$$

Abbreviating $L = \mathsf{B} + \mathsf{H}$, we can also then apply Lemma 13 to bound

$$\left\| \overline{\mathbb{V}^{\pi_\gamma^\star} \left[ \sum_{t=0}^\infty \gamma^t R_t \right]} \right\|_\infty \leq \frac{\left\| \overline{\mathbb{V}^{\pi_\gamma^\star} \left[ \sum_{t=0}^{L-1} \gamma^t R_t + \gamma^L V_\gamma^{\pi_\gamma^\star}(S_L) \right]} \right\|_\infty}{1 - \gamma^{2L}}.$$

We can repeat a similar argument as within Lemma 15 to bound this term. Fixing an initial state $s_0 \in \mathcal{R}^{\pi_\gamma^\star}$, the key observation is that $\rho^\star$ is constant on the recurrent block of $X_{\pi_\gamma^\star}$ containing $s_0$, and therefore any state trajectory $S_0 = s_0, S_1, S_2, \ldots$ under the transition matrix $P_{\pi_\gamma^\star}$ will have $\rho^\star(S_L) = \rho^\star(s_0)$. Therefore for this fixed $s_0$ we have

$$\begin{aligned}
\mathbb{V}_{s_0}^{\pi_\gamma^\star} \left[ \sum_{t=0}^{L-1} \gamma^t R_t + \gamma^L V_\gamma^{\pi_\gamma^\star}(S_L) \right] &= \mathbb{V}_{s_0}^{\pi_\gamma^\star} \left[ \sum_{t=0}^{L-1} \gamma^t R_t + \gamma^L \left( V_\gamma^{\pi_\gamma^\star}(S_L) - \frac{1}{1-\gamma} \rho^\star(s_0) \right) \right] \\
&\leq \mathbb{E}_{s_0}^{\pi_\gamma^\star} \left| \sum_{t=0}^{L-1} \gamma^t R_t + \gamma^L \left( V_\gamma^{\pi_\gamma^\star}(S_L) - \frac{1}{1-\gamma} \rho^\star(s_0) \right) \right|^2 \\
&\leq 2\mathbb{E}_{s_0}^{\pi_\gamma^\star} \left| \sum_{t=0}^{L-1} \gamma^t R_t \right|^2 + 2\mathbb{E}_{s_0}^{\pi_\gamma^\star} \left| \gamma^L \left( V_\gamma^{\pi_\gamma^\star}(S_L) - \frac{1}{1-\gamma} \rho^\star(s_0) \right) \right|^2 \\
&= 2\mathbb{E}_{s_0}^{\pi_\gamma^\star} \left| \sum_{t=0}^{L-1} \gamma^t R_t \right|^2 + 2\mathbb{E}_{s_0}^{\pi_\gamma^\star} \left| \gamma^L \left( V_\gamma^{\pi_\gamma^\star}(S_L) - \frac{1}{1-\gamma} \rho^\star(S_L) \right) \right|^2 \\
&\leq 2L^2 + 2 \sup_{s \in \mathcal{R}^{\pi_\gamma^\star}} \left( V_\gamma^{\pi_\gamma^\star}(s) - \frac{1}{1-\gamma} \rho^\star(s) \right)^2 \\
&\leq 2L^2 + 2\mathsf{H}^2 \\
&\leq 4L^2
\end{aligned}$$

where we used Lemma 21 in the penultimate inequality. Applying this argument to all $s_0 \in \mathcal{R}^{\pi_\gamma^\star}$ we obtain

$$\left\| \overline{\mathbb{V}^{\pi_\gamma^\star} \left[ \sum_{t=0}^{L-1} \gamma^t R_t + \gamma^L V_\gamma^{\pi_\gamma^\star}(S_L) \right]} \right\|_\infty \leq 4L^2.$$

Therefore by combining with our initial bounds we have that

$$
\max_{s \in \mathcal{R}^{\pi_\gamma^\star}} \gamma \left| e_s^\top (I - \gamma P_{\pi_\gamma^\star})^{-1} \sqrt{\mathbb{V}_{P_{\pi_\gamma^\star}}\left[V_\gamma^{\pi_\gamma^\star}\right]} \right| \leq \sqrt{\frac{2}{1-\gamma}} \sqrt{\left\| \mathbb{V}^{\pi_\gamma^\star}\left[\sum_{t=0}^{\infty} \gamma^t R_t\right] \right\|_\infty}
$$

$$
\leq \sqrt{\frac{2}{1-\gamma}} \sqrt{\frac{\left\| \mathbb{V}^{\pi_\gamma^\star}\left[\sum_{t=0}^{L-1} \gamma^t R_t + \gamma^L V_\gamma^{\pi_\gamma^\star}(S_L)\right] \right\|_\infty}{1-\gamma^{2L}}}
$$

$$
\leq \sqrt{\frac{2}{1-\gamma}} \sqrt{\frac{4L^2}{1-\gamma^{2L}}}
$$

$$
\leq \sqrt{\frac{2}{1-\gamma}} \sqrt{\frac{16L^2}{5L(1-\gamma)}}
$$

$$
\leq \sqrt{\frac{32}{5} \frac{L}{(1-\gamma)^2}},
$$

where in the penultimate inequality we used Lemma 14 to bound $\frac{1}{1-\gamma^{2L}} \leq \frac{5}{4}\frac{1}{(1-\gamma)L}$.  □

The next lemma controls the variance for $\widehat{\pi}_{\gamma,\mathrm{p}}^\star$ on recurrent states.

**Lemma 25.** *Letting $\widehat{\pi}_{\gamma,\mathrm{p}}^\star$ be the optimal policy for the discounted MDP $(\widehat{P}, \widetilde{r}, \gamma)$, if $\gamma \geq 1 - \frac{1}{\mathsf{B}+\mathsf{H}}$, we have*

$$
\max_{s \in \mathcal{R}^{\widehat{\pi}_{\gamma,\mathrm{p}}^\star}} \gamma \left| e_s^\top (I - \gamma P_{\widehat{\pi}_{\gamma,\mathrm{p}}^\star})^{-1} \sqrt{\mathbb{V}_{P_{\widehat{\pi}_{\gamma,\mathrm{p}}^\star}}\left[V_{\gamma,\mathrm{p}}^{\widehat{\pi}_{\gamma,\mathrm{p}}^\star}\right]} \right|
$$

$$
\leq \sqrt{29\frac{\mathsf{B}+\mathsf{H}}{(1-\gamma)^2}} + \sqrt{\frac{15}{\mathsf{B}+\mathsf{H}}} \frac{\left\| \widehat{V}_{\gamma,\mathrm{p}}^{\widehat{\pi}_{\gamma,\mathrm{p}}^\star} - V_\gamma^{\widehat{\pi}_{\gamma,\mathrm{p}}^\star} \right\|_\infty + \left\| \widehat{V}_{\gamma,\mathrm{p}}^{\pi_\gamma^\star} - V_\gamma^{\pi_\gamma^\star} \right\|_\infty}{1-\gamma}.
$$

*Proof.* Let $L = \mathsf{B} + \mathsf{H}$. By the same arguments as in the beginning of the proof of Lemma 24, we have

$$
\max_{s \in \mathcal{R}^{\widehat{\pi}_{\gamma,\mathrm{p}}^\star}} \gamma \left| e_s^\top (I - \gamma P_{\widehat{\pi}_{\gamma,\mathrm{p}}^\star})^{-1} \sqrt{\mathbb{V}_{P_{\widehat{\pi}_{\gamma,\mathrm{p}}^\star}}\left[V_{\gamma,\mathrm{p}}^{\widehat{\pi}_{\gamma,\mathrm{p}}^\star}\right]} \right| \leq \sqrt{\frac{2}{1-\gamma}} \sqrt{\left\| \mathbb{V}^{\widehat{\pi}_{\gamma,\mathrm{p}}^\star}\left[\sum_{t=0}^{\infty} \gamma^t \widetilde{R}_t\right] \right\|_\infty}
$$

$$
\leq \sqrt{\frac{2}{1-\gamma}} \sqrt{\frac{\left\| \mathbb{V}^{\widehat{\pi}_{\gamma,\mathrm{p}}^\star}\left[\sum_{t=0}^{L-1} \gamma^t \widetilde{R}_t + \gamma^L V_{\gamma,\mathrm{p}}^{\widehat{\pi}_{\gamma,\mathrm{p}}^\star}(S_L)\right] \right\|_\infty}{1-\gamma^{2L}}}
$$

so it again suffices to bound $\overline{\mathbb{V}^{\widehat{\pi}_{\gamma,\mathrm{p}}^\star}\left[\sum_{t=0}^{L-1} \gamma^t \widetilde{R}_t + \gamma^L V_{\gamma,\mathrm{p}}^{\widehat{\pi}_{\gamma,\mathrm{p}}^\star}(S_L)\right]}$. Fix $s_0 \in \mathcal{R}^{\widehat{\pi}_{\gamma,\mathrm{p}}^\star}$. Again, as observed in Lemma 24, $\rho^\star$ is constant on the recurrent block of $X_{\widehat{\pi}_{\gamma,\mathrm{p}}^\star}$ containing $s_0$, so we will have

$\rho^\star(S_L) = \rho^\star(s_0)$ with probability one. Therefore (mostly following the steps of Lemma 16)

$$
\mathbb{V}_{s_0}^{\widehat{\pi}_{\gamma,\mathrm{P}}^\star}\left[\sum_{t=0}^{L-1}\gamma^t\widetilde{R}_t + \gamma^L V_{\gamma,\mathrm{P}}^{\widehat{\pi}_{\gamma,\mathrm{P}}^\star}(S_L)\right]
$$

$$
= \mathbb{V}_{s_0}^{\widehat{\pi}_{\gamma,\mathrm{P}}^\star}\left[\sum_{t=0}^{L-1}\gamma^t\widetilde{R}_t + \gamma^L V_{\gamma,\mathrm{P}}^{\widehat{\pi}_{\gamma,\mathrm{P}}^\star}(S_L) - \gamma^L\frac{1}{1-\gamma}\rho^\star(s_0)\right]
$$

$$
\leq \mathbb{E}_{s_0}^{\widehat{\pi}_{\gamma,\mathrm{P}}^\star}\left(\sum_{t=0}^{L-1}\gamma^t\widetilde{R}_t + \gamma^L V_{\gamma,\mathrm{P}}^{\widehat{\pi}_{\gamma,\mathrm{P}}^\star}(S_L) - \gamma^L\frac{1}{1-\gamma}\rho^\star(s_0)\right)^2
$$

$$
= \mathbb{E}_{s_0}^{\widehat{\pi}_{\gamma,\mathrm{P}}^\star}\left(\sum_{t=0}^{L-1}\gamma^t\widetilde{R}_t + \gamma^L\left(V_{\gamma,\mathrm{P}}^{\widehat{\pi}_{\gamma,\mathrm{P}}^\star}(S_L) - V_\gamma^{\pi_\gamma^\star}(S_L)\right) + \gamma^L\left(V_\gamma^{\pi_\gamma^\star}(S_L) - \frac{1}{1-\gamma}\rho^\star(S_L)\right)\right)^2
$$

$$
\leq 3\mathbb{E}_{s_0}^{\widehat{\pi}_{\gamma,\mathrm{P}}^\star}\left(\sum_{t=0}^{L-1}\gamma^t\widetilde{R}_t\right)^2 + 3\gamma^{2L}\mathbb{E}_{s_0}^{\widehat{\pi}_{\gamma,\mathrm{P}}^\star}\left(V_{\gamma,\mathrm{P}}^{\widehat{\pi}_{\gamma,\mathrm{P}}^\star}(S_L) - V_\gamma^{\pi_\gamma^\star}(S_L)\right)^2
$$

$$
\quad + 3\gamma^{2L}\mathbb{E}_{s_0}^{\widehat{\pi}_{\gamma,\mathrm{P}}^\star}\left(V_\gamma^{\pi_\gamma^\star}(S_L) - \frac{1}{1-\gamma}\rho^\star(S_L)\right)^2
$$

$$
\leq 3\mathbb{E}_{s_0}^{\widehat{\pi}_{\gamma,\mathrm{P}}^\star}\left(\sum_{t=0}^{L-1}\gamma^t\widetilde{R}_t\right)^2 + 6\gamma^{2L}\mathbb{E}_{s_0}^{\widehat{\pi}_{\gamma,\mathrm{P}}^\star}\left(V_\gamma^{\widehat{\pi}_{\gamma,\mathrm{P}}^\star}(S_L) - V_\gamma^{\pi_\gamma^\star}(S_L)\right)^2 + 6\gamma^{2L}\left\|V_{\gamma,\mathrm{P}}^{\widehat{\pi}_{\gamma,\mathrm{P}}^\star} - V_\gamma^{\widehat{\pi}_{\gamma,\mathrm{P}}^\star}\right\|_\infty^2
$$

$$
\quad + 3\gamma^{2L}\mathbb{E}_{s_0}^{\widehat{\pi}_{\gamma,\mathrm{P}}^\star}\left(V_\gamma^{\pi_\gamma^\star}(S_L) - \frac{1}{1-\gamma}\rho^\star(S_L)\right)^2 \tag{16}
$$

using the inequalities $(a+b+c)^2 \leq 3a^2 + 3b^2 + 3c^2$ and $(a+b)^2 \leq 2a^2 + 2b^2$. Now we bound each term of (16) analogously to the steps of Lemma 16. For the first term of (16),

$$
3\mathbb{E}_{s_0}^{\widehat{\pi}_{\gamma,\mathrm{P}}^\star}\left(\sum_{t=0}^{L-1}\gamma^t\widetilde{R}_t\right)^2 \leq 3\left(L\|\widetilde{r}\|_\infty\right)^2 \leq 3L^2(\|r\|_\infty + \xi)^2 \leq 6L^2\left(1 + \left(\frac{(1-\gamma)\varepsilon}{6}\right)^2\right) \leq 6L^2\left(\frac{7}{6}\right)^2,
$$

where we had $\frac{(1-\gamma)\varepsilon}{6} \leq \frac{\varepsilon}{6L} \leq \frac{1}{6}$ because $\frac{1}{1-\gamma} \geq L$ and $\varepsilon \leq L$. For the second term of (16),

$$
6\gamma^{2L}\mathbb{E}_{s_0}^{\widehat{\pi}_{\gamma,\mathrm{P}}^\star}\left(V_\gamma^{\widehat{\pi}_{\gamma,\mathrm{P}}^\star}(S_L) - V_\gamma^{\pi_\gamma^\star}(S_L)\right)^2 \leq 6\left\|V_\gamma^{\widehat{\pi}_{\gamma,\mathrm{P}}^\star} - V_\gamma^{\pi_\gamma^\star}\right\|_\infty^2
$$

$$
\leq 6\left(\left\|\widehat{V}_{\gamma,\mathrm{P}}^{\widehat{\pi}_{\gamma,\mathrm{P}}^\star} - V_\gamma^{\widehat{\pi}_{\gamma,\mathrm{P}}^\star}\right\|_\infty + \left\|\widehat{V}_{\gamma,\mathrm{P}}^{\pi_\gamma^\star} - V_\gamma^{\pi_\gamma^\star}\right\|_\infty\right)^2
$$

where we used $(a+b)^2 \leq 2a^2 + 2b^2$ and the fact that $\left\|V_\gamma^{\widehat{\pi}_{\gamma,\mathrm{P}}^\star} - V_\gamma^{\pi_\gamma^\star}\right\|_\infty \leq \left\|\widehat{V}_{\gamma,\mathrm{P}}^{\widehat{\pi}_{\gamma,\mathrm{P}}^\star} - V_\gamma^{\widehat{\pi}_{\gamma,\mathrm{P}}^\star}\right\|_\infty + \left\|\widehat{V}_{\gamma,\mathrm{P}}^{\pi_\gamma^\star} - V_\gamma^{\pi_\gamma^\star}\right\|_\infty$ which was shown in Lemma 16. For the third term of (16),

$$
6\gamma^{2L}\left\|V_{\gamma,\mathrm{P}}^{\widehat{\pi}_{\gamma,\mathrm{P}}^\star} - V_\gamma^{\widehat{\pi}_{\gamma,\mathrm{P}}^\star}\right\|_\infty^2 \leq 6\left\|V_{\gamma,\mathrm{P}}^{\widehat{\pi}_{\gamma,\mathrm{P}}^\star} - V_\gamma^{\widehat{\pi}_{\gamma,\mathrm{P}}^\star}\right\|_\infty^2 \leq 6\left(\frac{\xi}{1-\gamma}\right)^2 = 6\left(\frac{\varepsilon}{6}\right)^2 \leq \frac{L^2}{6}
$$

where the fact that $\left\|V_{\gamma,\mathrm{P}}^{\widehat{\pi}_{\gamma,\mathrm{P}}^\star} - V_\gamma^{\widehat{\pi}_{\gamma,\mathrm{P}}^\star}\right\|_\infty \leq \frac{\xi}{1-\gamma}$ is identical to the arguments used in the proof of Lemma 10, and the final inequality is due to the assumption that $\varepsilon \leq L$. For the fourth term of (16),

$$
3\gamma^{2L}\mathbb{E}_{s_0}^{\widehat{\pi}_{\gamma,\mathrm{P}}^\star}\left(V_\gamma^{\pi_\gamma^\star}(S_L) - \frac{1}{1-\gamma}\rho^\star(S_L)\right)^2 \leq 3\left\|V_\gamma^{\pi_\gamma^\star} - \frac{1}{1-\gamma}\rho^\star\right\|_\infty^2 \leq 3L^2
$$

using Lemma 22 for the second inequality. Using all these bounds in (16), we obtain

$$
\mathbb{V}_{s_0}^{\widehat{\pi}_{\gamma,\mathrm{P}}^\star}\left[\sum_{t=0}^{L-1}\gamma^t\widetilde{R}_t + \gamma^L V_{\gamma,\mathrm{P}}^{\widehat{\pi}_{\gamma,\mathrm{P}}^\star}(S_L)\right] \leq \left(\frac{49}{6} + \frac{1}{6} + 3\right)L^2 + 6\left(\left\|\widehat{V}_{\gamma,\mathrm{P}}^{\widehat{\pi}_{\gamma,\mathrm{P}}^\star} - V_\gamma^{\widehat{\pi}_{\gamma,\mathrm{P}}^\star}\right\|_\infty + \left\|\widehat{V}_{\gamma,\mathrm{P}}^{\pi_\gamma^\star} - V_\gamma^{\pi_\gamma^\star}\right\|_\infty\right)^2
$$

and so (since this holds for arbitrary $s_0 \in \mathcal{R}^{\widehat{\pi}^\star_{\gamma,\mathrm{p}}}$), we have

$$\overline{\mathbb{V}^{\widehat{\pi}^\star_{\gamma,\mathrm{p}}}\left[\sum_{t=0}^{L-1}\gamma^t\widetilde{R}_t + \gamma^L V^{\widehat{\pi}^\star_{\gamma,\mathrm{p}}}_{\gamma,\mathrm{p}}(S_L)\right]} \leq \frac{68}{6}L^2 + 6\left(\left\|\widehat{V}^{\widehat{\pi}^\star_{\gamma,\mathrm{p}}}_{\gamma,\mathrm{p}} - V^{\widehat{\pi}^\star_{\gamma,\mathrm{p}}}_\gamma\right\|_\infty + \left\|\widehat{V}^{\pi^\star_\gamma}_{\gamma,\mathrm{p}} - V^{\pi^\star_\gamma}_\gamma\right\|_\infty\right)^2.$$

Therefore, combining with our initial arguments,

$$\max_{s\in\mathcal{R}^{\widehat{\pi}^\star_{\gamma,\mathrm{p}}}}\gamma\left|e_s^\top(I-\gamma P_{\widehat{\pi}^\star_{\gamma,\mathrm{p}}})^{-1}\sqrt{\mathbb{V}_{P_{\widehat{\pi}^\star_{\gamma,\mathrm{p}}}}\left[V^{\widehat{\pi}^\star_{\gamma,\mathrm{p}}}_{\gamma,\mathrm{p}}\right]}\right|$$

$$\leq \sqrt{\frac{2}{1-\gamma}}\sqrt{\frac{\left\|\overline{\mathbb{V}^{\widehat{\pi}^\star_{\gamma,\mathrm{p}}}\left[\sum_{t=0}^{L-1}\gamma^t\widetilde{R}_t + \gamma^L V^{\widehat{\pi}^\star_{\gamma,\mathrm{p}}}_{\gamma,\mathrm{p}}(S_L)\right]}\right\|_\infty}{1-\gamma^{2L}}}$$

$$\leq \sqrt{\frac{2}{1-\gamma}}\frac{\sqrt{\frac{68}{6}L^2 + 6\left(\left\|\widehat{V}^{\widehat{\pi}^\star_{\gamma,\mathrm{p}}}_{\gamma,\mathrm{p}} - V^{\widehat{\pi}^\star_{\gamma,\mathrm{p}}}_\gamma\right\|_\infty + \left\|\widehat{V}^{\pi^\star_\gamma}_{\gamma,\mathrm{p}} - V^{\pi^\star_\gamma}_\gamma\right\|_\infty\right)^2}}{\sqrt{1-\gamma^{2L}}}$$

$$\leq \sqrt{\frac{2}{1-\gamma}}\frac{\sqrt{\frac{68}{6}L^2} + \sqrt{6\left(\left\|\widehat{V}^{\widehat{\pi}^\star_{\gamma,\mathrm{p}}}_{\gamma,\mathrm{p}} - V^{\widehat{\pi}^\star_{\gamma,\mathrm{p}}}_\gamma\right\|_\infty + \left\|\widehat{V}^{\pi^\star_\gamma}_{\gamma,\mathrm{p}} - V^{\pi^\star_\gamma}_\gamma\right\|_\infty\right)^2}}{\sqrt{1-\gamma^{2L}}}$$

$$\leq \sqrt{\frac{2}{1-\gamma}}\frac{\sqrt{\frac{68}{6}L^2} + \sqrt{6\left(\left\|\widehat{V}^{\widehat{\pi}^\star_{\gamma,\mathrm{p}}}_{\gamma,\mathrm{p}} - V^{\widehat{\pi}^\star_{\gamma,\mathrm{p}}}_\gamma\right\|_\infty + \left\|\widehat{V}^{\pi^\star_\gamma}_{\gamma,\mathrm{p}} - V^{\pi^\star_\gamma}_\gamma\right\|_\infty\right)^2}}{\sqrt{\frac{4}{5}(1-\gamma)L}}$$

$$< \sqrt{29\frac{L}{(1-\gamma)^2}} + \sqrt{\frac{15}{L}}\frac{\left\|\widehat{V}^{\widehat{\pi}^\star_{\gamma,\mathrm{p}}}_{\gamma,\mathrm{p}} - V^{\widehat{\pi}^\star_{\gamma,\mathrm{p}}}_\gamma\right\|_\infty + \left\|\widehat{V}^{\pi^\star_\gamma}_{\gamma,\mathrm{p}} - V^{\pi^\star_\gamma}_\gamma\right\|_\infty}{1-\gamma},$$

where we used Lemma 14 to bound $\frac{1}{1-\gamma^{2L}} \leq \frac{5}{4}\frac{1}{(1-\gamma)L}$. $\qquad\square$

The next lemma controls the variance on all states.

**Lemma 26.** *Under the settings of Lemmas 24 and 25, we have*

$$\gamma\left\|(I-\gamma P_{\pi^\star_\gamma})^{-1}\sqrt{\mathbb{V}_{P_{\pi^\star_\gamma}}\left[V^{\pi^\star_\gamma}_\gamma\right]}\right\|_\infty \leq 4\sqrt{\frac{\mathsf{B}+\mathsf{H}}{(1-\gamma)^2}}$$

*and*

$$\gamma\left\|(I-\gamma P_{\widehat{\pi}^\star_{\gamma,\mathrm{p}}})^{-1}\sqrt{\mathbb{V}_{P_{\widehat{\pi}^\star_{\gamma,\mathrm{p}}}}\left[V^{\widehat{\pi}^\star_{\gamma,\mathrm{p}}}_{\gamma,\mathrm{p}}\right]}\right\|_\infty \leq 8\sqrt{\frac{\mathsf{B}+\mathsf{H}}{(1-\gamma)^2}} + \sqrt{\frac{15}{\mathsf{B}+\mathsf{H}}}\frac{\left\|\widehat{V}^{\widehat{\pi}^\star_{\gamma,\mathrm{p}}}_{\gamma,\mathrm{p}} - V^{\widehat{\pi}^\star_{\gamma,\mathrm{p}}}_\gamma\right\|_\infty + \left\|\widehat{V}^{\pi^\star_\gamma}_{\gamma,\mathrm{p}} - V^{\pi^\star_\gamma}_\gamma\right\|_\infty}{1-\gamma}.$$

*Proof.* First we establish the first bound in the lemma statement. As we have already bounded the entries corresponding to the recurrent states of $\pi^\star_\gamma$ by Lemma 24, it remains to bound the transient states. Let $s \in \mathcal{T}^{\pi^\star_\gamma}$ be an arbitrary transient state. Using Lemma 19, we have

$$e_s^\top\gamma(I-\gamma P_{\pi^\star_\gamma})^{-1}\sqrt{\mathbb{V}_{P_{\pi^\star_\gamma}}\left[V^{\pi^\star_\gamma}_\gamma\right]} = \gamma\underline{e_s}^\top\sum_{k=1}^\infty\gamma^k Z^{k-1}_{\pi^\star_\gamma}Y_{\pi^\star_\gamma}(I-\gamma X_{\pi^\star_\gamma})^{-1}\sqrt{\mathbb{V}_{P_{\pi^\star_\gamma}}\left[V^{\pi^\star_\gamma}_\gamma\right]}$$

$$+ \gamma\underline{e_s}^\top\sum_{t=0}^\infty\gamma^t Z^t_{\pi^\star_\gamma}\sqrt{\mathbb{V}_{P_{\pi^\star_\gamma}}\left[V^{\pi^\star_\gamma}_\gamma\right]}. \qquad (17)$$

Now we bound each of the terms in (17). For the first term, we can calculate

$$\gamma \underline{e_s}^\top \sum_{k=1}^\infty \gamma^k Z_{\pi_\gamma^\star}^{k-1} Y_{\pi_\gamma^\star} (I - \gamma X_{\pi_\gamma^\star})^{-1} \sqrt{\mathbb{V}_{P_{\pi_\gamma^\star}}\left[V_\gamma^{\pi_\gamma^\star}\right]}$$

$$\leq \gamma \underline{e_s}^\top \sum_{k=1}^\infty Z_{\pi_\gamma^\star}^{k-1} Y_{\pi_\gamma^\star} (I - \gamma X_{\pi_\gamma^\star})^{-1} \sqrt{\mathbb{V}_{P_{\pi_\gamma^\star}}\left[V_\gamma^{\pi_\gamma^\star}\right]}$$

$$\leq \left\| \underline{e_s}^\top \sum_{k=1}^\infty Z_{\pi_\gamma^\star}^{k-1} Y_{\pi_\gamma^\star} \right\|_1 \gamma \left\| (I - \gamma X_{\pi_\gamma^\star})^{-1} \sqrt{\mathbb{V}_{P_{\pi_\gamma^\star}}\left[V_\gamma^{\pi_\gamma^\star}\right]} \right\|_\infty$$

$$\leq \sqrt{\frac{32}{5} \frac{\mathsf{B} + \mathsf{H}}{(1-\gamma)^2}}$$

where we used the fact that $\underline{e_s}^\top \sum_{k=1}^\infty Z_{\pi_\gamma^\star}^{k-1} Y_{\pi_\gamma^\star}$ is a probability distribution and Lemma 24.

For the second term of (17), we have

$$\gamma \underline{e_s}^\top \sum_{t=0}^\infty \gamma^t Z_{\pi_\gamma^\star}^t \sqrt{\mathbb{V}_{P_{\pi_\gamma^\star}}\left[V_\gamma^{\pi_\gamma^\star}\right]} = \gamma \left\| \underline{e_s}^\top \sum_{t=0}^\infty \gamma^t Z_{\pi_\gamma^\star}^t \right\|_1 \sum_{t=0}^\infty \frac{\gamma^t \underline{e_s}^\top Z_{\pi_\gamma^\star}^t}{\left\| \underline{e_s}^\top \sum_{t=0}^\infty \gamma^t Z_{\pi_\gamma^\star}^t \right\|_1} \sqrt{\mathbb{V}_{P_{\pi_\gamma^\star}}\left[V_\gamma^{\pi_\gamma^\star}\right]}$$

$$\leq \gamma \left\| \underline{e_s}^\top \sum_{t=0}^\infty \gamma^t Z_{\pi_\gamma^\star}^t \right\|_1 \sqrt{\sum_{t=0}^\infty \frac{\gamma^t \underline{e_s}^\top Z_{\pi_\gamma^\star}^t}{\left\| \underline{e_s}^\top \sum_{t=0}^\infty \gamma^t Z_{\pi_\gamma^\star}^t \right\|_1} \mathbb{V}_{P_{\pi_\gamma^\star}}\left[V_\gamma^{\pi_\gamma^\star}\right]}$$

$$= \sqrt{\left\| \underline{e_s}^\top \sum_{t=0}^\infty \gamma^t Z_{\pi_\gamma^\star}^t \right\|_1} \sqrt{\gamma^2 \sum_{t=0}^\infty \gamma^t \underline{e_s}^\top Z_{\pi_\gamma^\star}^t \mathbb{V}_{P_{\pi_\gamma^\star}}\left[V_\gamma^{\pi_\gamma^\star}\right]} \qquad (18)$$

where we used Jensen's inequality since $x \mapsto \sqrt{x}$ is concave and $\frac{\sum_{t=0}^\infty \gamma^t \underline{e_s}^\top Z_{\pi_\gamma^\star}^t}{\left\| \underline{e_s}^\top \sum_{t=0}^\infty \gamma^t Z_{\pi_\gamma^\star}^t \right\|_1}$ is a probability distribution (all entries of this row vector are positive and they sum to 1 due to our normalization). Now we bound each factor in (18). Using Lemma 18, we have

$$\sqrt{\left\| \underline{e_s}^\top \sum_{t=0}^\infty \gamma^t Z_{\pi_\gamma^\star}^t \right\|_1} \leq \sqrt{\left\| \underline{e_s}^\top \sum_{t=0}^\infty Z_{\pi_\gamma^\star}^t \right\|_1} \leq \sqrt{\mathsf{B}}.$$

For the second factor in (18), we have

$$\sum_{t=0}^\infty \gamma^t \underline{e_s}^\top Z_{\pi_\gamma^\star}^t \mathbb{V}_{P_{\pi_\gamma^\star}}\left[V_\gamma^{\pi_\gamma^\star}\right] \leq \sum_{t=0}^\infty \gamma^t \underline{e_s}^\top Z_{\pi_\gamma^\star}^t \mathbb{V}_{P_{\pi_\gamma^\star}}\left[V_\gamma^{\pi_\gamma^\star}\right]$$

$$+ \underline{e_s}^\top \sum_{k=1}^\infty \gamma^k Z_{\pi_\gamma^\star}^{k-1} Y_{\pi_\gamma^\star} (I - \gamma X_{\pi_\gamma^\star})^{-1} \mathbb{V}_{P_{\pi_\gamma^\star}}\left[V_\gamma^{\pi_\gamma^\star}\right]$$

$$= e_s^\top (I - \gamma P_{\pi_\gamma^\star})^{-1} \mathbb{V}_{P_{\pi_\gamma^\star}}\left[V_\gamma^{\pi_\gamma^\star}\right]$$

where the equality step is due to Lemma 19. Now we can apply two steps which are used within Lemma 12 to obtain the desired bound on this term. Abbreviating $v = \mathbb{V}_{P_{\pi_\gamma^\star}}\left[V_\gamma^{\pi_\gamma^\star}\right]$, it is shown within Lemma 12 that

$$\gamma^2 \left\| (I - \gamma P_{\pi_\gamma^\star})^{-1} v \right\|_\infty \leq 2\gamma^2 \left\| (I - \gamma^2 P_{\pi_\gamma^\star})^{-1} v \right\|_\infty \leq 2 \left\| \mathbb{V}^{\pi_\gamma^\star}\left[\sum_{t=0}^\infty \gamma^t R_t\right] \right\|_\infty \leq \frac{2}{(1-\gamma)^2}$$

(where the final inequality is because the total discounted return is within $[0, \frac{1}{1-\gamma}]$). Therefore we can bound the second factor in (18) as

$$\sqrt{\gamma^2 \sum_{t=0}^\infty \gamma^t \underline{e_s}^\top Z_{\pi_\gamma^\star}^t \mathbb{V}_{P_{\pi_\gamma^\star}}\left[V_\gamma^{\pi_\gamma^\star}\right]} \leq \sqrt{\frac{2}{(1-\gamma)^2}} = \frac{\sqrt{2}}{1-\gamma}.$$

Combining all of these bounds back into (17), we have

$$e_s^\top \gamma (I - \gamma P_{\pi_\gamma^\star})^{-1} \sqrt{\mathbb{V}_{P_{\pi_\gamma^\star}} \left[ V_\gamma^{\pi_\gamma^\star} \right]} \leq \sqrt{\frac{32}{5} \frac{\mathsf{B} + \mathsf{H}}{(1 - \gamma)^2}} + \sqrt{\mathsf{B}} \frac{\sqrt{2}}{1 - \gamma}$$

$$< 4 \sqrt{\frac{\mathsf{B} + \mathsf{H}}{(1 - \gamma)^2}}.$$

Thus we have established the first inequality from the lemma statement.

For the second inequality, the argument is entirely analogous, except that we use Lemma 25 instead of Lemma 24, and also in the MDP with the perturbed reward $\widetilde{r}$ we have the bound

$$\left\| \mathbb{V}^{\pi_\gamma^\star} \left[ \sum_{t=0}^\infty \gamma^t R_t \right] \right\|_\infty \leq \left( \frac{\|\widetilde{r}\|_\infty}{1 - \gamma} \right)^2 \leq \left( \frac{\|r\|_\infty + \xi}{1 - \gamma} \right)^2$$

$$\leq \frac{1}{(1 - \gamma)^2} \left( 1 + \frac{(1 - \gamma)\varepsilon}{6} \right)^2 \leq \frac{1}{(1 - \gamma)^2} \left( \frac{7}{6} \right)^2,$$

where we used the fact that $\frac{(1-\gamma)\varepsilon}{6} \leq \frac{\varepsilon}{6(\mathsf{B}+\mathsf{H})} \leq \frac{1}{6}$ because $\frac{1}{1-\gamma} \geq \mathsf{B} + \mathsf{H}$ and $\varepsilon \leq \mathsf{B} + \mathsf{H}$. Thus we can obtain the bound

$$\gamma \left\| (I - \gamma P_{\widehat{\pi}_{\gamma,\mathrm{P}}^\star})^{-1} \sqrt{\mathbb{V}_{P_{\widehat{\pi}_{\gamma,\mathrm{P}}^\star}} \left[ V_{\gamma,\mathrm{P}}^{\widehat{\pi}_{\gamma,\mathrm{P}}^\star} \right]} \right\|_\infty$$

$$\leq \sqrt{29 \frac{\mathsf{B} + \mathsf{H}}{(1 - \gamma)^2}} + \sqrt{\frac{15}{\mathsf{B} + \mathsf{H}}} \frac{\left\| \widehat{V}_{\gamma,\mathrm{P}}^{\widehat{\pi}_{\gamma,\mathrm{P}}^\star} - V_\gamma^{\widehat{\pi}_{\gamma,\mathrm{P}}^\star} \right\|_\infty + \left\| \widehat{V}_{\gamma,\mathrm{P}}^{\pi_\gamma^\star} - V_\gamma^{\pi_\gamma^\star} \right\|_\infty}{1 - \gamma}$$

$$+ \sqrt{\mathsf{B}} \frac{7\sqrt{2}}{6(1 - \gamma)}$$

$$\leq 8 \sqrt{\frac{\mathsf{B} + \mathsf{H}}{(1 - \gamma)^2}} + \sqrt{\frac{15}{\mathsf{B} + \mathsf{H}}} \frac{\left\| \widehat{V}_{\gamma,\mathrm{P}}^{\widehat{\pi}_{\gamma,\mathrm{P}}^\star} - V_\gamma^{\widehat{\pi}_{\gamma,\mathrm{P}}^\star} \right\|_\infty + \left\| \widehat{V}_{\gamma,\mathrm{P}}^{\pi_\gamma^\star} - V_\gamma^{\pi_\gamma^\star} \right\|_\infty}{1 - \gamma}.$$

This completes the proof of the lemma. $\qquad \square$

We are now ready to prove Theorem 7 on the sample complexity of general discounted MDPs.

*Proof of Theorem 7.* To prove Theorem 7 we will combine our bounds of the variance parameters in Lemma 26 with Lemma 10. First, starting with (1) from Lemma 10 and combining with the first bound from Lemma 26, we have that there exist absolute constants $c_1, c_2$ such that for any $\delta \in (0, 1)$,

if $n \geq \frac{c_2}{1-\gamma} \log\left(\frac{SA}{(1-\gamma)\delta\varepsilon}\right)$, then with probability at least $1 - \delta$

$$\left\| \widehat{V}_{\gamma,\mathrm{P}}^{\pi_\gamma^\star} - V_\gamma^{\pi_\gamma^\star} \right\|_\infty \leq \gamma\sqrt{\frac{c_1 \log\left(\frac{SA}{(1-\gamma)\delta\varepsilon}\right)}{n}} \left\| (I - \gamma P_{\pi_\gamma^\star})^{-1} \sqrt{\mathbb{V}_{P_{\pi_\gamma^\star}}\left[V_\gamma^{\pi_\gamma^\star}\right]} \right\|_\infty$$

$$+ c_1 \gamma \frac{\log\left(\frac{SA}{(1-\gamma)\delta\varepsilon}\right)}{(1-\gamma)n} \left\| V_\gamma^{\pi_\gamma^\star} \right\|_\infty + \frac{\varepsilon}{6}$$

$$\leq \sqrt{\frac{c_1 \log\left(\frac{SA}{(1-\gamma)\delta\varepsilon}\right)}{n}} 4\sqrt{\frac{\mathsf{B}+\mathsf{H}}{(1-\gamma)^2}} + c_1\gamma \frac{\log\left(\frac{SA}{(1-\gamma)\delta\varepsilon}\right)}{(1-\gamma)n} \left\| V_\gamma^{\pi_\gamma^\star} \right\|_\infty + \frac{\varepsilon}{6}$$

$$\leq \sqrt{\frac{c_1 \log\left(\frac{SA}{(1-\gamma)\delta\varepsilon}\right)}{n}} 4\sqrt{\frac{\mathsf{B}+\mathsf{H}}{(1-\gamma)^2}} + c_1 \frac{\log\left(\frac{SA}{(1-\gamma)\delta\varepsilon}\right)}{(1-\gamma)^2 n} + \frac{\varepsilon}{6}$$

$$\leq \frac{\varepsilon}{6} + \frac{1}{16\cdot 6^2}\frac{\varepsilon^2}{\mathsf{B}+\mathsf{H}} + \frac{\varepsilon}{6}$$

$$\leq \frac{\varepsilon}{2},$$

where the penultimate inequality is under the assumption that $n \geq 16\cdot 6^2 c_1 \frac{\mathsf{B}+\mathsf{H}}{\varepsilon^2(1-\gamma)^2} \log\left(\frac{SA}{(1-\gamma)\delta\varepsilon}\right)$, and the final inequality makes use of the fact that $\varepsilon \leq \mathsf{B} + \mathsf{H}$.

Next, still using Lemma 10, under the same event, we also have

$$\left\| \widehat{V}_{\gamma,\mathrm{P}}^{\widehat{\pi}_{\gamma,\mathrm{P}}^\star} - V_\gamma^{\widehat{\pi}_{\gamma,\mathrm{P}}^\star} \right\|_\infty$$

$$\leq \gamma\sqrt{\frac{c_1 \log\left(\frac{SA}{(1-\gamma)\delta\varepsilon}\right)}{n}} \left\| (I - \gamma P_{\widehat{\pi}_{\gamma,\mathrm{P}}^\star})^{-1} \sqrt{\mathbb{V}_{P_{\widehat{\pi}_{\gamma,\mathrm{P}}^\star}}\left[V_{\gamma,\mathrm{P}}^{\widehat{\pi}_{\gamma,\mathrm{P}}^\star}\right]} \right\|_\infty$$

$$+ c_1\gamma \frac{\log\left(\frac{SA}{(1-\gamma)\delta\varepsilon}\right)}{(1-\gamma)n} \left\| V_{\gamma,\mathrm{P}}^{\widehat{\pi}_{\gamma,\mathrm{P}}^\star} \right\|_\infty + \frac{\varepsilon}{6}$$

$$\leq \sqrt{\frac{c_1 \log\left(\frac{SA}{(1-\gamma)\delta\varepsilon}\right)}{n}} \left( 8\sqrt{\frac{\mathsf{B}+\mathsf{H}}{(1-\gamma)^2}} + \sqrt{\frac{15}{\mathsf{B}+\mathsf{H}}} \frac{\left\| \widehat{V}_{\gamma,\mathrm{P}}^{\widehat{\pi}_{\gamma,\mathrm{P}}^\star} - V_\gamma^{\widehat{\pi}_{\gamma,\mathrm{P}}^\star} \right\|_\infty + \left\| \widehat{V}_{\gamma,\mathrm{P}}^{\pi_\gamma^\star} - V_\gamma^{\pi_\gamma^\star} \right\|_\infty}{1-\gamma} \right)$$

$$+ c_1\gamma \frac{\log\left(\frac{SA}{(1-\gamma)\delta\varepsilon}\right)}{(1-\gamma)n} \left\| V_{\gamma,\mathrm{P}}^{\widehat{\pi}_{\gamma,\mathrm{P}}^\star} \right\|_\infty + \frac{\varepsilon}{6}$$

$$\leq \sqrt{\frac{c_1 \log\left(\frac{SA}{(1-\gamma)\delta\varepsilon}\right)}{n}} \left( 8\sqrt{\frac{\mathsf{B}+\mathsf{H}}{(1-\gamma)^2}} + \sqrt{\frac{15}{\mathsf{B}+\mathsf{H}}} \frac{\left\| \widehat{V}_{\gamma,\mathrm{P}}^{\widehat{\pi}_{\gamma,\mathrm{P}}^\star} - V_\gamma^{\widehat{\pi}_{\gamma,\mathrm{P}}^\star} \right\|_\infty + (\mathsf{B}+\mathsf{H})/2}{1-\gamma} \right)$$

$$+ c_1 \frac{\log\left(\frac{SA}{(1-\gamma)\delta\varepsilon}\right)}{(1-\gamma)n} \frac{7}{6}\frac{1}{1-\gamma} + \frac{\varepsilon}{6}$$

using the second inequality from Lemma 26 for the second inequality, and then we use the fact that $\left\| V_{\gamma,\mathrm{P}}^{\widehat{\pi}_{\gamma,\mathrm{P}}^\star} \right\|_\infty \leq \frac{7}{6}\frac{1}{1-\gamma}$ which was argued in Lemma 26, as well as the fact from above that

$\left\|\widehat{V}_{\gamma,\mathrm{p}}^{\pi_\gamma^\star} - V_\gamma^{\pi_\gamma^\star}\right\|_\infty \le \varepsilon/2 \le (\mathsf{B}+\mathsf{H})/2$. After rearranging, we obtain that

$$\left(1 - \sqrt{\frac{c_1 \log\left(\frac{SA}{(1-\gamma)\delta\varepsilon}\right)}{n}} \sqrt{\frac{15}{\mathsf{B}+\mathsf{H}}} \frac{1}{1-\gamma}\right) \left\|\widehat{V}_{\gamma,\mathrm{p}}^{\widehat{\pi}_{\gamma,\mathrm{p}}^\star} - V_\gamma^{\widehat{\pi}_{\gamma,\mathrm{p}}^\star}\right\|_\infty$$

$$\le \sqrt{\frac{c_1 \log\left(\frac{SA}{(1-\gamma)\delta\varepsilon}\right)}{n}} \left(8\sqrt{\frac{\mathsf{B}+\mathsf{H}}{(1-\gamma)^2}} + \sqrt{\frac{15}{\mathsf{B}+\mathsf{H}}} \frac{(\mathsf{B}+\mathsf{H})/2}{1-\gamma}\right) + c_1 \frac{\log\left(\frac{SA}{(1-\gamma)\delta\varepsilon}\right)}{(1-\gamma)^2 n} \frac{7}{6} + \frac{\varepsilon}{6}$$

$$\le \sqrt{\frac{c_1 \log\left(\frac{SA}{(1-\gamma)\delta\varepsilon}\right)}{n}} 10\sqrt{\frac{\mathsf{B}+\mathsf{H}}{(1-\gamma)^2}} + c_1 \frac{\log\left(\frac{SA}{(1-\gamma)\delta\varepsilon}\right)}{(1-\gamma)^2 n} \frac{7}{6} + \frac{\varepsilon}{6}. \qquad (19)$$

If $n \ge 6^2 \cdot 10^2 c_1 \frac{\mathsf{B}+\mathsf{H}}{\varepsilon^2(1-\gamma)^2} \log\left(\frac{SA}{(1-\gamma)\delta\varepsilon}\right)$, then the RHS of (19) is bounded by

$$\frac{\varepsilon}{6} + \frac{7}{6} \frac{\varepsilon^2}{\mathsf{B}+\mathsf{H}} \frac{1}{6^2 \cdot 10^2} + \frac{\varepsilon}{6} \le \left(\frac{1}{6} + \frac{1}{6^2 \cdot 10^2} + \frac{1}{6}\right) \varepsilon \le 0.4\varepsilon$$

using the assumption that $\varepsilon \le \mathsf{B}+\mathsf{H}$. Under the same condition on $n$, we also have that

$$\left(1 - \sqrt{\frac{c_1 \log\left(\frac{SA}{(1-\gamma)\delta\varepsilon}\right)}{n}} \sqrt{\frac{15}{\mathsf{B}+\mathsf{H}}} \frac{1}{1-\gamma}\right) \left\|\widehat{V}_{\gamma,\mathrm{p}}^{\widehat{\pi}_{\gamma,\mathrm{p}}^\star} - V_\gamma^{\widehat{\pi}_{\gamma,\mathrm{p}}^\star}\right\|_\infty$$

$$\ge \left(1 - \sqrt{\frac{\varepsilon^2}{(\mathsf{B}+\mathsf{H})^2}} \sqrt{\frac{15}{6^2 \cdot 10^2}}\right) \left\|\widehat{V}_{\gamma,\mathrm{p}}^{\widehat{\pi}_{\gamma,\mathrm{p}}^\star} - V_\gamma^{\widehat{\pi}_{\gamma,\mathrm{p}}^\star}\right\|_\infty$$

$$\ge \left(1 - \sqrt{\frac{15}{6^2 \cdot 10^2}}\right) \left\|\widehat{V}_{\gamma,\mathrm{p}}^{\widehat{\pi}_{\gamma,\mathrm{p}}^\star} - V_\gamma^{\widehat{\pi}_{\gamma,\mathrm{p}}^\star}\right\|_\infty$$

$$\ge 0.9 \left\|\widehat{V}_{\gamma,\mathrm{p}}^{\widehat{\pi}_{\gamma,\mathrm{p}}^\star} - V_\gamma^{\widehat{\pi}_{\gamma,\mathrm{p}}^\star}\right\|_\infty$$

where again we used the assumption that $\varepsilon \le \mathsf{B}+\mathsf{H}$. Combining these two bounds with the inequality (19), we obtain that

$$0.9 \left\|\widehat{V}_{\gamma,\mathrm{p}}^{\widehat{\pi}_{\gamma,\mathrm{p}}^\star} - V_\gamma^{\widehat{\pi}_{\gamma,\mathrm{p}}^\star}\right\|_\infty \le 0.4\varepsilon$$

which implies that

$$\left\|\widehat{V}_{\gamma,\mathrm{p}}^{\widehat{\pi}_{\gamma,\mathrm{p}}^\star} - V_\gamma^{\widehat{\pi}_{\gamma,\mathrm{p}}^\star}\right\|_\infty \le \frac{0.4}{0.9}\varepsilon < \frac{\varepsilon}{2}.$$

Since we have established that $\left\|\widehat{V}_{\gamma,\mathrm{p}}^{\pi_\gamma^\star} - V_\gamma^{\pi_\gamma^\star}\right\|_\infty \le \frac{\varepsilon}{2}$ and that $\left\|\widehat{V}_{\gamma,\mathrm{p}}^{\widehat{\pi}_{\gamma,\mathrm{p}}^\star} - V_\gamma^{\widehat{\pi}_{\gamma,\mathrm{p}}^\star}\right\|_\infty \le \frac{\varepsilon}{2}$, since also $\widehat{V}_{\gamma,\mathrm{p}}^{\widehat{\pi}_{\gamma,\mathrm{p}}^\star} \ge \widehat{V}_{\gamma,\mathrm{p}}^{\pi_\gamma^\star}$, we can conclude that

$$V_\gamma^{\pi_\gamma^\star} - V_\gamma^{\widehat{\pi}_{\gamma,\mathrm{p}}^\star} \le \left\|\widehat{V}_{\gamma,\mathrm{p}}^{\pi_\gamma^\star} - V_\gamma^{\pi_\gamma^\star}\right\|_\infty \mathbf{1} + \left\|\widehat{V}_{\gamma,\mathrm{p}}^{\widehat{\pi}_{\gamma,\mathrm{p}}^\star} - V_\gamma^{\widehat{\pi}_{\gamma,\mathrm{p}}^\star}\right\|_\infty \mathbf{1} \le \varepsilon \mathbf{1},$$

that is that $\widehat{\pi}_{\gamma,\mathrm{p}}^\star$ is $\varepsilon$-optimal.

Finally, we check that all of our conditions on $n$ can be satisfied if

$$n \ge \max\left\{6^2 \cdot 10^2 c_1 \frac{\mathsf{B}+\mathsf{H}}{\varepsilon^2(1-\gamma)^2}, 6^2 \cdot 16 c_1 \frac{\mathsf{B}+\mathsf{H}}{\varepsilon^2(1-\gamma)^2}, \frac{c_2}{1-\gamma}\right\} \log\left(\frac{SA}{(1-\gamma)\delta\varepsilon}\right),$$

and since $\frac{1}{1-\gamma} \ge \mathsf{B}+\mathsf{H}$ and $\mathsf{B}+\mathsf{H} \ge \varepsilon$, we have $\frac{\mathsf{B}+\mathsf{H}}{\varepsilon^2(1-\gamma)^2} \ge \frac{(\mathsf{B}+\mathsf{H})^2}{\varepsilon^2(1-\gamma)} \ge \frac{1}{1-\gamma}$, so the above is guaranteed if we set $C_3 = \max\{6^2 \cdot 10^2 c_1, c_2\}$ and require $n \ge C_3 \frac{\mathsf{B}+\mathsf{H}}{\varepsilon^2(1-\gamma)^2} \log\left(\frac{SA}{(1-\gamma)\delta\varepsilon}\right)$. $\qquad \square$

### B.3 Proof of Theorem 8 (General Average-Reward MDP Bounds)

In this section, we prove our main result on the sample complexity of general average-reward MDPs.

*Proof of Theorem 8.* We can combine our bound for discounted MDPs, Theorem 7, with our reduction from average-reward MDPs to discounted MDPs, Theorem 6.

Using Theorem 7 with target accuracy $\mathsf{B} + \mathsf{H}$ and discount factor $\overline{\gamma} = 1 - \frac{\varepsilon}{12(\mathsf{B}+\mathsf{H})}$, we obtain a $(\mathsf{B} + \mathsf{H})$-optimal policy for the discounted MDP $(P, r, \overline{\gamma})$ with probability at least $1 - \delta$ as long as

$$
\begin{aligned}
n &\geq C_3 \frac{\mathsf{B} + \mathsf{H}}{(1 - \overline{\gamma})^2(\mathsf{B} + \mathsf{H})^2} \log\left(\frac{SA}{(1 - \overline{\gamma})\delta\varepsilon}\right) \\
&= 12^2 C_3 \frac{\mathsf{B} + \mathsf{H}}{(\mathsf{B} + \mathsf{H})^2} \frac{(\mathsf{B} + \mathsf{H})^2}{\varepsilon^2} \log\left(\frac{12(\mathsf{B} + \mathsf{H})}{\varepsilon} \frac{SA}{\delta\varepsilon}\right)
\end{aligned}
$$

which is satisfied when $n \geq C_4 \frac{\mathsf{B}+\mathsf{H}}{\varepsilon^2} \log\left(\frac{SA(\mathsf{B}+\mathsf{H})}{\delta\varepsilon}\right)$ for sufficiently large $C_4$.

Applying Theorem 6 (with error parameter $\frac{\varepsilon}{12}$), we obtain

$$
\rho^\star - \rho^{\widehat{\pi}^\star} \leq \left(3 + 2\frac{\mathsf{B} + \mathsf{H}}{\mathsf{B} + \mathsf{H}}\right) \frac{\varepsilon}{12} \leq \varepsilon \mathbf{1}
$$

as desired. $\qquad\square$

### B.4 Proof of Theorems 4 and 5 (Lower Bounds)

In this section, we prove our minimax lower bounds on the sample complexity of general average-reward MDPs (Theorem 4) and discounted MDPs (Theorem 5).

*Proof of Theorem 4.* First consider the MDP instances $\mathcal{M}_{a^\star}$ indexed by $a^\star \in \{1, \ldots, A\}$ shown in Figure 3. In all instances, states $2, 3$ and $4$ are absorbing states, and state $1$ is a transient state. State 1 has $A$ actions and is the only state with multiple actions. At state 1, taking action $a = 1$ will take the agent to state 4 deterministically; taking action 2 will take the agent back to state 1 with probability $P(1|1, 2) = 1 - \frac{1}{T}$, to state 2 with probability $P(2|1, 2)$, and to state 3 with probability $P(3|1, 2) = 1 - P(1|1, 2) - P(2|1, 2)$. The instances differ only in the values of $P(2|1, a)$ and $P(3|1, a)$, which are shown in Figure 3 along with the reward $R$ for each state-action pair.

For the MDP instance $\mathcal{M}_1$, the optimal policy is taking action $a = 1$ at state 1, leading to an average reward of $1/2$; taking any other action leads to a sub-optimal average reward of $\frac{1-2\varepsilon}{2}$. Similarly, for the instance $\mathcal{M}_{a^\star}$ with $a^\star \in \{2, \ldots, A\}$, the optimal action is $a = a^\star$ with average reward $\frac{1+2\varepsilon}{2}$, the action $a = 1$ has average reward $\frac{1}{2}$, and all other actions have average reward $\frac{1-2\varepsilon}{2}$. By direct calculation, we find that the span of the optimal policy is $\|h^\star\|_{\text{span}} = 0$ in all instances. Moreover, by taking any action $a \neq 1$, the agent will stay in state 1 for $B$ steps in expectation before transitioning to state 2 or 3, so the bounded transient time is satisfied with parameter $B$.

We next define $(A-1)S/4$ master MDPs $\overline{\mathcal{M}}_{s^\star, a^\star}$ indexed by $s^\star \in \{1, \ldots, S/4\}$ and $a^\star \in \{2, \ldots, A\}$ as follows. Each master MDP $\overline{\mathcal{M}}_{s^\star, a^\star}$ has $S/4$ copies of sub-MDPs such that the $s^\star$th sub-MDP is equal to $\mathcal{M}_{a^\star}$ and all other sub-MDPs are equal to $\mathcal{M}_1$. We rename the states so that the states of the $s$th sub-MDP has states $4s + 1, 4s + 2, 4s + 3, 4s + 4$ corresponding to states $1, 2, 3, 4$ of the instances shown in Figure 3. Note each of these master MDPs has $S$ states and $A$ actions, satisfies the bounded transient time property with parameter $B$, and has the span of the bias of its Blackwell optimal policy equal to 0. Note that for a given policy $\pi$ to be $\varepsilon/3$-average optimal in master MDP $\overline{\mathcal{M}}_{s^\star, a^\star}$, it must take action $a^\star$ in state $4s^\star + 1$ with probability at least $2/3$, and it must take action 1 in states $4s + 1$ for $s \in \{1, \ldots, S/4\} \setminus \{s^\star\}$ with probability at least $2/3$.

Thus, for an algorithm $\mathtt{Alg}$ to output an $\varepsilon/3$-average optimal policy $\pi$, it must identify the master MDP instance $\overline{\mathcal{M}}_{s^\star, a^\star}$ (equivalently, the values of $s^\star$ and $a^\star$), in the sense that there must be exactly one state $4s + 1$ where an action $a \neq 1$ is taken with probability $\geq 2/3$. Therefore it suffices to lower bound the failure probability of any algorithm $\mathtt{Alg}$ for this $(A - 1)S/4$-way testing problem. By

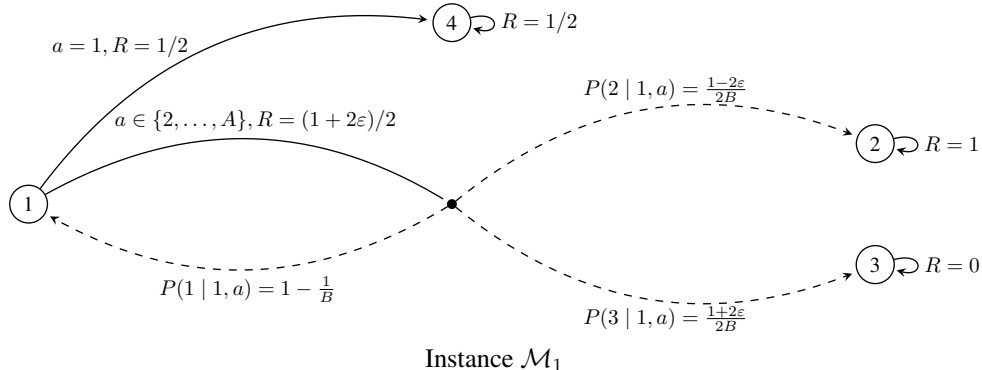

Instance $\mathcal{M}_1$

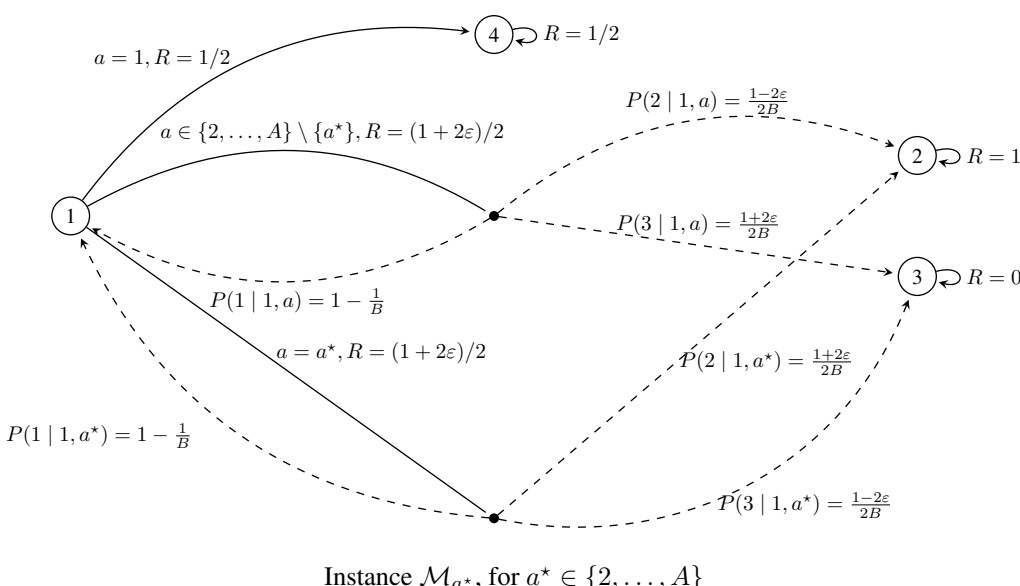

Instance $\mathcal{M}_{a^\star}$, for $a^\star \in \{2, \ldots, A\}$

Figure 3: MDP Instances Used in the Proof of Lower Bound in Theorem 4

construction, for any two distinct index pairs $(s_1^\star, a_1^\star)$ and $(s_2^\star, a_2^\star)$, the master MDPs $\overline{\mathcal{M}}_{s_1^\star, a_1^\star}$ and $\overline{\mathcal{M}}_{s_2^\star, a_2^\star}$ differ only in the state-action pairs $(4s_1^\star, a_1^\star)$ and $(4s_2^\star, a_2^\star)$, and we have

$$P_{\overline{\mathcal{M}}_{s_1^\star, a_1^\star}}(\cdot \mid 4s_1^\star, a_1^\star) = \text{Cat}\left(1 - \frac{1}{B}, \frac{1 - 2\varepsilon}{2B}, \frac{1 + 2\varepsilon}{2B}\right) =: Q_1,$$

$$P_{\overline{\mathcal{M}}_{s_2^\star, a_2^\star}}(\cdot \mid 4s_1^\star, a_1^\star) = \text{Cat}\left(1 - \frac{1}{B}, \frac{1 + 2\varepsilon}{2B}, \frac{1 - 2\varepsilon}{2B}\right) =: Q_2,$$

where $\text{Cat}(p_1, p_2, p_3)$ denotes the categorical distribution with event probabilities $p_i$'s (and vice versa for the distributions of the state action pair $(4s_2^\star, a_2^\star)$).

Now we use Fano's method [18] to lower bound this failure probability. Choose an index $J$ uniformly at random from the set $\mathcal{J} := \{1, \ldots, S/4\} \times \{2, \ldots, A\}$ and suppose that we draw $n$ iid samples $X = (X_1, \ldots, X_n)$ from the master MDP $\overline{\mathcal{M}}_J$; note that under the generative model, each random variable $X_i$ represents an $(S \times A)$-by-$S$ transition matrix with exactly one nonzero entry in each row. Letting $\text{I}(J; X)$ denote the mutual information between $J$ and $X$, Fano's inequality yields that the failure probability is lower bounded by

$$1 - \frac{\text{I}(J; X) + \log 2}{\log((A - 1)S/4)}.$$

We can calculate using the fact that the $P_i$'s are i.i.d., the chain rule of mutual information, and the form of the construction that

$$
\mathrm{I}(J; X) = n\mathrm{I}(J; X_1)
$$

$$
\leq n \max_{\substack{(s_1^\star, a_1^\star), (s_2^\star, a_2^\star) \in \mathcal{J}: \\ (s_1^\star, a_1^\star) \neq (s_2^\star, a_2^\star)}} \mathrm{D_{KL}}\left(P_{\overline{\mathcal{M}}_{s_1^\star, a_1^\star}} \,\Big|\, P_{\overline{\mathcal{M}}_{s_2^\star, a_2^\star}}\right)
$$

$$
= n\big(\mathrm{D_{KL}}(Q_1 \mid Q_2) + \mathrm{D_{KL}}(Q_2 \mid Q_1)\big).
$$

By direct calculation, we have

$$
\begin{aligned}
\mathrm{D_{KL}}(Q_1|Q_2) &= \frac{1-2\varepsilon}{2B} \log \frac{1-2\varepsilon}{1+2\varepsilon} + \frac{1+2\varepsilon}{2B} \log \frac{1+2\varepsilon}{1-2\varepsilon} \\
&\leq \frac{1-2\varepsilon}{2B} \cdot \frac{-4\varepsilon}{1+2\varepsilon} + \frac{1+2\varepsilon}{2B} \cdot \frac{4\varepsilon}{1-2\varepsilon} \qquad\qquad \log(1+x) \leq x, \forall x > -1 \\
&= \frac{16\varepsilon^2}{B(1+2\varepsilon)(1-2\varepsilon)} \\
&\leq \frac{32\varepsilon^2}{B} \qquad\qquad\qquad\qquad\qquad\qquad\qquad\qquad \varepsilon \leq \frac{1}{4}.
\end{aligned}
$$

Also note that $\mathrm{D_{KL}}(Q_2|Q_1) = \mathrm{D_{KL}}(Q_1|Q_2)$ in this case. Therefore the failure probability is at least

$$
1 - \frac{\mathrm{I}(J; P^n) + \log 2}{\log((A-1)S/4)} \geq 1 - \frac{n\frac{64\varepsilon^2}{B} + \log 2}{\log((A-1)S/4)}
$$

$$
\geq \frac{1}{2} - \frac{n\frac{64\varepsilon^2}{B}}{\log((A-1)S/4)},
$$

where in the second inequality we assumed $A$ and $S$ are at least a sufficiently large constant. For the above RHS to be smaller than $1/4$, we therefore require $n \geq \Omega(\frac{B\log(SA)}{\varepsilon^2})$. $\qquad\square$

*Proof of Theorem 5.* The desired DMDP lower bound follows from combining our AMDP lower bound Theorem 4 with the average-to-discount reduction in Theorem 6. $\qquad\square$

## B.5 Relationship between transient time and mixing time

**Lemma 27.** *In any uniformly mixing MDP, we have* $\mathsf{B} \leq 4\tau_{\mathrm{unif}}$.

*Proof.* Fix a deterministic stationary policy $\pi$. Notice that since all states in the support of the stationary distribution $\nu_\pi$ are recurrent, for any $s \in \mathcal{S}$ we have

$$
\begin{aligned}
\mathbb{P}_s^\pi\left(S_t \text{ is transient}\right) &= \sum_{s' \in \mathcal{T}^\pi} \mathbb{P}_s^\pi\left(S_t = s'\right) \\
&\leq \sum_{s' \in \mathcal{T}^\pi} \mathbb{P}_s^\pi\left(S_t = s'\right) + \sum_{s' \in \mathcal{R}^\pi} |\mathbb{P}_s^\pi\left(S_t = s'\right) - \nu^\pi(s')| \\
&= \sum_{s' \in \mathcal{S}} |\mathbb{P}_s^\pi\left(S_t = s'\right) - \nu^\pi(s')| \\
&\leq 2\max_{s \in \mathcal{S}} \frac{1}{2} \left\|e_s^\top P_\pi^t - \nu^\pi\right\|_1 \\
&\leq 2 \cdot 2^{-\lfloor t/\tau_{\mathrm{unif}} \rfloor}
\end{aligned}
$$

where the final inequality uses standard properties of mixing [11, Chapter 4]. Now define $T = \inf\{t : S_t \in \mathcal{R}^\pi\}$. Then, using a standard formula for the expectation of nonnegative-integer-values random

variables, we have for any $s \in \mathcal{S}$ that

$$
\begin{aligned}
\mathbb{E}_s^\pi [T] &= \sum_{t=0}^\infty \mathbb{P}_s^\pi (T > t) \\
&= \sum_{t=0}^\infty \mathbb{P}_s^\pi (S_t \text{ is transient}) \\
&\leq 2 \sum_{t=0}^\infty 2^{-\lfloor t/\tau_{\mathrm{unif}} \rfloor} \\
&= 2 \sum_{\ell=0}^\infty \tau_{\mathrm{unif}} 2^{-\ell} \\
&= 4\tau_{\mathrm{unif}}.
\end{aligned}
$$

Since this bound holds for all $s \in \mathcal{S}$ and all deterministic stationary policies $\pi$, we conclude that $\mathsf{B} \leq 4\tau_{\mathrm{unif}}$. $\qquad\square$

