# OpenReview forum: "Span-Based Optimal Sample Complexity for Weakly Communicating and General Average Reward MDPs"
_NeurIPS.cc/2024/Conference — NeurIPS 2024 oral_

### Official Review · Reviewer_hsZR · 2024-06-24

**Soundness:** 3
**Presentation:** 4
**Contribution:** 3
**Rating:** 7
**Confidence:** 3

**Summary:**

This paper presents a model-based reinforcement learning algorithm for tabular MDPs with the average reward criteria. The authors assume a generative model (each state-action pairs can be simulated) and studies learning algorithm that sample each state-action pairs n times.  In the weakly communicating setting, the authors provides the first algorithm that achieves the minimax lower-bound to find an \varepsilon-policy. The authors study also the more general case of multi-chain MDPs for which the authors introduce a parameter B (time to visite a recurrent-state) and propose both an analysis of a minimax lower bound and an algorithm that achieves this bound.

**Strengths:**

The paper is really well written. The related work section show that the paper is well documented, and the authors make a good use of the related work via the various references throughout the text. The choice of the examples and their use is nicely done. All in all, the paper is nice to ready.

The paper improves on related work and obtain an algorithm that matches the lower bound.

The paper is very precise in its definitions (contrary to many papers on the same subjects).

The paper studies learning algorithm for multi-chain MDPs, which is rarely done.

**Weaknesses:**

The algorithmic novelty of the paper seems limited. Only the analysis seems new.

The authors focus on a generative model (although this is probably unavoidable in the multichain case).

To me, the discussion on the lower bound is not complete.

**Questions:**

Are there any novelty in the algorithmic part or is it just the analysis that is new?

The use of a generative model drastically simplifies the analysis. Would any of these results translate to a navigating model?

The lower bound seem to imply that *all* state-action pairs have to be visited n times. I am surprized that the sampling of state-action pairs is not adaptive. Would it change anything?

**Limitations:**

NA.

---

> ### Author Rebuttal · Authors · 2024-08-07
>
> - Regarding algorithmic novelty, you are correct that several of our results are established with novel analyses of existing algorithms. We believe that this is a strength of our results, for several reasons. Simple algorithms are preferable in practice, and our analyses demonstrate that several of the simplest and most well-studied approaches, namely average-to-discounted reduction and the plug-in approach for solving DMDPs, can be used to achieve optimal sample complexity. We believe this is a more impactful result as opposed to a problem-tailored algorithm which is unlikely to be used in practice. Additionally, our results on the plug-in approach for solving DMDPs hold when any algorithm is used to solve the empirical MDP $\widehat{P}$, which is a stronger result than proving that only a particular algorithm works.
> - Additionally, we believe that our results for general/multichain MDPs represent algorithmic novelties, since conditions for the average-to-discounted reduction to work were not previously known for this setting (and the conditions are different compared to the weakly communicating setting; namely, a different effective horizon is required within the reduction). For example, even when the true transition matrix $P$ is known, our multichain average-to-discounted reduction Theorem 6 suggests new iterative algorithms for solving multichain MDPs: if we (approximately) solve $P$ as a DMDP with effective horizon $(\mathsf{B}+\mathsf{H})/\varepsilon$ by standard discounted value iteration methods, then we get an $\varepsilon$-gain-optimal policy. This is interesting to compare to usual (undiscounted/average-reward) value iteration, which to our knowledge does not have finite-time guarantees for general MDPs.
> - We agree that generative model access can be a strong assumption, but we note that it can be a building block for algorithms which work in more general settings. Sometimes this may be from an indirect theory-building perspective, and other times algorithms for the generative model can be directly used and combined with another procedure for exploring and obtaining samples in a navigating model. Also, as pointed out by reviewer i8Mw, in commonly studied uniformly mixing settings, the navigating model basically reduces to the generative model (with an additional mixing time complexity factor).
> - Regarding the lower bound, the sampling model is chosen to match that of the generative model, which assumes an equal number of samples from each state-action pair. We believe that an adaptive sampling model would not substantially change the lower bound. (We actually believe that our current lower bound construction could be adapted to this setting, for the following reasons: The construction for Theorem 4 is based on the difficulty of distinguishing amongst a set of $\Theta(SA)$ different MDPs. Each of these MDPs has a different “special” state-action pair $(s^\star,a^\star)$ which yields slightly larger expected average reward but are otherwise identical. Discerning the identity of $(s^\star,a^\star)$ by sampling adaptively from different state-action pairs is thus similar to a best-arm identification problem for stochastic multi-armed bandits, so we believe adaptive lower bound arguments from that setting could be used here.)

---

> > ### Comment · Reviewer_hsZR · 2024-08-12
> >
> > Thank you for these clarifications. I updated my score.

---

### Official Review · Reviewer_AFus · 2024-06-29

**Soundness:** 4
**Presentation:** 4
**Contribution:** 3
**Rating:** 7
**Confidence:** 3

**Summary:**

This work obtains the first minimax optimal sample complexity bound of weakly communicating and general average reward MDPs, without uniform mixing assumption, by introducing new transient time parameter and obtaining tighter minimax optimal sample complexity bound for discounted MDP.

**Strengths:**

This theoretical work provides comprehensive lit review. I did not check the proof, but the theoretical results look reasonable and strong based on my knowledge about discounted MDP theory. The presentation is clear.

**Weaknesses:**

A few points are to be clarified as shown in the questions below.

**Questions:**

(1) You may change the following citation which has been accepted by ICLR 2024.

[21] Shengbo Wang, Jose Blanchet, and Peter Glynn. Optimal Sample Complexity for Average Reward Markov Decision Processes, October 2023.

Some other citations like [13] lack location (ArXiv, conference, journal, etc.).

(2) In lines 152-153 about the definition of Blackwell-optimal 154 policy, do you mean for all $\gamma\in[\overline{\gamma},1)$ or $\gamma\in[\overline{\gamma},1]$? Does $V_{\gamma}^{\pi^*}\ge V_{\gamma}^{\pi}$ mean $V_{\gamma}^{\pi^*}(s)\ge V_{\gamma}^{\pi}(s), \forall s$?

(3) In lines 155-156, what does $P_{sa}\rho^*$ mean? Does $\rho^*(s)\ge P_{\pi}\rho^*$ mean $\rho^*(s)\ge (P_{\pi}\rho^*)(s):=\sum_{s\in\mathcal{S}}P_{\pi}(s,s')\rho^*(s')$? The meaning of $\rho^*(s)\ge {\rm a~vector}$ is not clear to me.

(4) In Theorem 2, the accuracy $\overline{\epsilon}=H$ for DMDP is not arbitrarily small. Why can the accuracy for AMDP be arbitrarily small $\epsilon$?

**Limitations:**

I agree with the following two statements made in the checklist:

(1) Limitations: "The conclusion (Section 5) mentions the main limitation, of the necessity of knowledge of H/B for the optimal average-reward complexity results to hold, and this point is elaborated upon in Section 3."

(2) Negative societal impact: "Our work is foundational research on the sample complexity of average-reward and discounted MDPs, and thus is not directly tied to any negative applications."

---

> ### Author Rebuttal · Authors · 2024-08-07
>
> 1. Thank you for making us aware of the updated citation. We will fix this and the other citations which lack locations.
> 2. Our definition of Blackwell-optimal policies is standard and matches that of Puterman [14, Chapter 10]. We must use $\gamma \in [\overline{\gamma}, 1)$ since the discount factor $\gamma$ must be strictly less than $1$ (the discounted value function is not defined for $\gamma = 1$). You are correct that the statement $V_\gamma^{\pi^\star} \geq V_\gamma^{\pi}$ means that $V_\gamma^{\pi^\star}(s) \geq V_\gamma^{\pi}(s)$ for all states $s$.
> 3. Thank you for catching this issue. $P_{sa} \rho^\star = \sum_{s’} P(s’\mid s,a) \rho^\star(s’)$, as we treat $P_{sa}$ as a row vector. You are correct that there should not be an index in the second definition, which should be written $\rho^\star \geq P_\pi \rho^\star$ (and understood to hold in an elementwise sense), in order for it to be equivalent to the first definition that $\rho^\star(s) \geq \max_{a \in \mathcal{A}} P_{sa} \rho^\star$. We will fix this typo.
> 4. We agree this is an interesting point. Mathematically, from the reduction from (weakly-communicating) average-reward MDPs to discounted MDPs [20, Proof of Theorem 1], we are guaranteed that if policy $\pi$ is $\overline{\varepsilon}$-optimal for DMDP (meaning $V^\pi_\gamma \geq V_\gamma^\star - \overline{\varepsilon} \mathbf{1}\$), then $\pi$ has gain at least $\rho^\star - C(1-\gamma)(\mathsf{H} + \overline{\varepsilon})$ for an absolute constant $C$. Note that $(1-\gamma)$ is the inverse of the effective horizon so this bound goes to $0$ as the effective horizon increases, which is exactly what is done within Theorem 2 as we set the effective horizon to be like $C’ \frac{\mathsf{H}}{\varepsilon}$ (where $\varepsilon$ is the target accuracy).

---

> > ### Comment · Reviewer_AFus · 2024-08-07
> > **Reviewer AFus is satisfied with the authors' response and will keep rating 7.**
> >
> > Reviewer AFus is satisfied with the authors' response and will keep rating 7.

---

### Official Review · Reviewer_i8Mw · 2024-07-03

**Soundness:** 3
**Presentation:** 3
**Contribution:** 3
**Rating:** 7
**Confidence:** 3

**Summary:**

This paper presents an algorithm with optimal sample complexity in general average reward MDPs.

**Strengths:**

The algorithm proposed is sample optimal for the general class of MDPs (possibly  multichain) that are much harder to learn than uni-chain or ergodic MDPs.
It also introduces a new parameter, the transient time that helps one assess  the sample complexity for multi-chain MDPs.

**Weaknesses:**

Although the following papers only appear recently on Arxiv ( i am not an author of either of them)
I think they should be mentioned because they answer some questions raised in the paper.

First XXBoone shows that regret bounds  using H are possible without prior knowledge of H, disproving the supposed conejcture
 in papers [5,4,25] cited in this submission. Actually, the same paper answers the point about the computational eficiency of the optimal algorithm.

Second, the results of the current submission should be compared to XXKauffman which seems to solve the same problem.

In the ergodic case, i agree that navigating and generative models are almost similar but in the general case, the generative model looks very strong.

**Questions:**

1. The authors discuss the fact that H cannot estimated while D can be. However they do not mention anything about B ?
My first guess would be that B cannot be estimated either because it looks discontinuous in the paramaters on the MDP.

2. Maybe I am mistaken but I did not see a proper definition of the transient time B, used in the statement of Theorems 4 and 5.

**Limitations:**

Again, the generative model looks like a strong assumptions.

The lack of numerical experiments is classical in this domain. It seems that MDPs of size 10 are already impossible to learn, which limits strongly the practical aspect of this type of algorithms. Can the authors comment on this?

---

> ### Author Rebuttal · Authors · 2024-08-07
>
> - We would like to call attention to our results for weakly-communicating MDPs, which we believe represent a major strength of our work as they resolve the longstanding problem of the sample complexity of average-reward MDPs (Theorem 2) with an interesting insight on the complexity of discounted MDPs (Theorem 1).
> - We note that both of the referenced preprints appeared on arXiv after the NeurIPS submission deadline, so they should not contribute negatively to the evaluation of our submission. Still, we are happy to discuss their relation to our work while attempting to maintain our own anonymity.
>   - The preprint by Kaufmann et al. is directly inspired by a preprint version of the present submission. They also provide evidence that the optimal bias span $\mathsf{H}$ is hard to estimate, although their lower bound instances require randomized rewards unlike our Theorem 3. As suggested by our paper starting at line 276, the diameter $D$ is estimable and upper bounds $\mathsf{H}$, so a diameter estimation procedure can be used to remove the need for knowledge of $\mathsf{H}$. Kaufmann et al. formalize this observation and combine an existing algorithm for the generative model which requires knowledge of $\mathsf{H}$ with a diameter estimation procedure from prior work to get a diameter-based sample complexity in the generative model setting, and they also combine a sample collection procedure for the navigating setting (also from prior work) to be able to apply the existing generative model algorithm to the navigating setting.
>   - The preprint by Boone et al. claims to obtain an optimal span-based regret bound in the online setting without prior knowledge of $\mathsf{H}$. We are unable to verify their result but we agree it would be highly surprising in light of past conjectures. We note that their result does not imply any sample complexity bounds for our setting, as unlike the episodic finite horizon setting, there is no known regret-to-PAC conversion. In fact, the mentioned paper by Kaufmann et al. provides discussion suggesting such a conversion is impossible, and it also claims to show that no online algorithm with or without knowledge of $\mathsf{H}$ can identify an $\varepsilon$-gain-optimal policy with the $\widetilde{O}(SA\mathsf{H}/\varepsilon^2)$ complexity achieved by our algorithm. Regarding simplicity and efficiency, while their algorithm is efficient, it is still very complicated and apparently does not achieve the optimal regret $\widetilde{O}(\sqrt{SAHT})$ until at least $T \geq \Omega(S^{40}A^{20}H^{10})$. Hence, we still find it surprising that (in our different setting), there exists an optimal algorithm, our Theorem 2, which achieves the optimal complexity $\widetilde{O}(SA \mathsf{H}/\varepsilon^2)$ for all $\varepsilon < 1$ and is highly simple.
> - We agree that generative model access can be a strong assumption, but we believe that it plays a fundamental theoretical role and its study can lead to algorithms for more general settings. In particular, even when the MDP is not ergodic, algorithms for the navigating setting can reduce to the generative model, which is the approach taken in the mentioned paper by Kaufmann et al., who reduce to the model-based algorithm that we use. Additionally, as discussed starting at line 49, we believe the generative model is particularly natural for studying general/multichain MDPs.
> - The definition of the bounded transient time parameter $\mathsf{B}$ appears in Section 2 (Problem setup and preliminaries), line 176.
> Regarding the estimation of the bounded transient time parameter $\mathsf{B}$, we believe that, as you suggest, this parameter may be difficult to estimate. While we believe your point about the discontinuity of $\mathsf{B}$ is correct, we believe that the discontinuity may not actually be the main obstacle. While generally $\widehat{P} \to P$ does not imply $\mathsf{B}(\widehat{P}) \to \mathsf{B}(P)$, the natural sampling model also ensures that the support of $\widehat{P}$ is contained within that of $P$ (and eventually their supports must be equal), and with this additional constraint (on the sequence of empirical transition matrices) we should have $\mathsf{B}(\widehat{P}) \to \mathsf{B}(P)$. However, we are unsure how to compute the function $\mathsf{B}(\widehat{P})$ without enumerating exponentially many policies. Consequently we believe $\mathsf{B}$ can only be tractably bounded for small MDPs or when there is some prior knowledge/structure in the MDP.
> - Regarding your comment that MDPs of size 10 are impossible to learn, we are not sure which measure of size you refer to. For MDPs with $S \cdot A = 10$, our algorithms would be highly practical. For MDPs with a number of states on the order of $2^{10}$, we agree that tabular algorithms such as ours would not be practical, and instead function approximation would be needed. We hope that analogous to episodic MDPs, our study of the average-reward tabular setting can lead towards algorithms using function approximation methods.

---

### Official Review · Reviewer_6ECs · 2024-07-15

**Soundness:** 4
**Presentation:** 4
**Contribution:** 4
**Rating:** 10
**Confidence:** 4

**Summary:**

The paper resolves the open problem of designing an algorithm for the generative tabular average reward setting for weakly communicating MDPs that achieves optimal span-dependent sample complexity with known span. This is done by an original observation that is concerned with discounted MDPs: Existing sample complexity bounds for the discounted setting are refined and the result is obtained from this refinement by reducing the average reward setting to the discounted setting, just like it was done in previous works. A second result is to give the first sample complexity results for general MDPs; with matching lower and upper bounds.

**Strengths:**

Solving a major open problem based on an interesting insight: This is a breakthrough paper.

**Weaknesses:**

None

**Questions:**

n.a.

**Limitations:**

n.a.

---

> ### Author Rebuttal · Authors · 2024-08-07
>
> Thank you for your positive review.

---

### Author Rebuttal · Authors · 2024-08-07

We thank all reviewers for their time and positive feedback. We will respond to each reviewer directly via individual rebuttals.

---

### Decision · Program_Chairs · 2024-09-25

**Decision:**

Accept (oral)

**Comment:**

Reviewers unanimously agree that this is a good paper resolving an important question. The presentation is also clear. Some reviewers believe this is an award-quality paper.